# BENCHMARKING WORLD-MODEL LEARNING

## ABSTRACT

Model-learning agents should gather information to learn world models that support many downstream tasks and inferences, such as predicting unobserved states, estimating near- and far-term consequences of actions, planning action sequences, and detecting changes in dynamics. Current methods for learning and evaluating world models diverge from this goal: training and evaluation are anchored to next-frame prediction, and success is scored by reward maximization in the same environment. We propose *WorldTest*, a protocol to evaluate model-learning agents that separates reward-free interaction from a scored test phase in a different but related environment. WorldTest is open-ended—models should support many different tasks unknown ahead of time—and agnostic to model representation, allowing comparison across approaches. We instantiated WorldTest with *AutumnBench*, a suite of 43 interactive grid-world environments and 129 tasks across three families: masked-frame prediction, planning, and predicting changes to the causal dynamics. We compared 517 human participants and three frontier models on AutumnBench. We found that humans outperform the models, and scaling compute improves performance only in some environments but not others. WorldTest provides a novel template—reward-free exploration, derived tests, and behavior-based scoring— to evaluate what agents learn about environment dynamics, and AutumnBench exposes significant headroom in world-model learning.

## 1 INTRODUCTION

Consider someone who cooks regularly in their own kitchen. Over time, they build an internal model of the workspace—where tools live and how appliances behave. That model supports various everyday capabilities. For example, it enables the person to: (1) Predict how long the hidden contents of a covered pot will take to finish cooking, based on steam intensity and elapsed time, (2) Recognize and adapt to changes when traveling and cooking in a short-term rental, where the kitchen is different (e.g., knives in a different drawer), and (3) Plan a sequence of actions to complete a set of recipes. Cognitive science refers to this flexible, predictive, and counterfactual understanding as *world model* (Weisberg & Gopnik, 2013), a core substrate of human intelligence. Many researchers argue that learning such models is pivotal for the next step in AI progress (LeCun & Courant, 2022).

**Benchmarking World-Model Learning with Environment-Level Queries**   A core challenge in world-model research is evaluation. Classic reinforcement learning (RL) benchmarks assess agents through task-specific rewards (Brockman et al., 2016). While agents may implicitly build a representation of the environment structure as they pursue rewards—for example, learning that one burner is more efficient because using it leads to faster cooking and thus higher returns—RL benchmarks do not directly test if the agent learns this structure. To evaluate world models rather than task-specific policies, we need a way to explicitly probe what an agent has inferred about the environment's underlying dynamics.

We call such probes *environment-level queries*: questions whose answers depend on properties of the *entire* environment dynamics, rather than a trajectory of an agent's interaction with the environment. Examples include: (1) inferring what will happen behind an occlusion, (2) detecting changes in the environment's dynamics, and (3) determining if a state is reachable from another. To assess agent's world-model learning capabilities, we should present the query explicitly and treat the agent's response as a direct measure of what aspects of the environment's dynamics it has learned.

Existing efforts to evaluate world-model learning, including RL-based benchmarks, only partially assess such capabilities. Cognitive-science-style benchmarks such as Bongard problems and the ARC

challenge (Depeweg et al., 2018; Chollet, 2019) assess the ability to infer hidden rules from static examples, which are fundamentally environment-level queries about the underlying non-interactive environment. However, they do not test an agent's ability to learn structure through exploration. Conversely, interactive benchmarks such as DiscoveryWorld (Jansen et al., 2024) and CLEVRER variants (Yi et al., 2020b) allow exploration but constrain answers to symbolic logics, code programs, or natural-language descriptions. Symbolic formats restrict the agent's input–output interface and hinder fair comparison across different agent types, including humans. Natural-language answers are informal and impractical for automated evaluation.

These limitations motivate the central question of this work: *What is an evaluation framework for assessing world-model learning in interactive environments that supports environment-level queries without imposing constraints on the agent's output representations?*

**Our Approach**   We introduce *WorldTest*, a behavior-based world-model evaluation framework for probing environment-level queries. It consists of two phases: interaction and test. In the *interaction phase*, the agent interacts with the base environment *without* external rewards. In the *test phase*, the framework transforms the base environment into a *derived challenge environment with an explicit objective* that differs from—but remains correlated with—the base environment. WorldTest evaluates the agent based on its ability to achieve this objective in the challenge environment.

WorldTest offers three advantages over existing methods: (1) WorldTest scores only an agent's behavior in a challenge environment, without inspecting or constraining its internal representations, which minimally captures action-conditioned dynamics without committing to specific formats such as natural-language, enabling fair and automatic evaluation. (2) The interaction phase is goal-free, i.e., the agent does not optimize any specific reward function. (3) The test phase measures how well the agent uses its learned world model to solve a new, related challenge environment. Unlike RL-based approaches that restrict rewards in the original environment, this framework evaluates world-learning by introducing novel challenges. It can cover a broad range of environment-level queries about the environment dynamics that were previously impossible under RL-based benchmarks.

We further instantiate WorldTest by *AutumnBench*, a benchmark consisting of 43 grid-world environments and three types of challenges for each environment. The three challenges mirror the capabilities in our cooking example:

- Masked-frame prediction (MFP), where the agent infers unobserved parts of a final observation given a masked trajectory, analogous to predicting cooking completion time from partially observable information.

- Change detection (CD), where the agent identifies a change in the dynamics of the environment and reports the earliest timestep where the observation differs from the expected observation in the original environment, similar to recognizing changes in a different kitchen.

- Planning, where the agent acts to achieve a goal state, similar to sequencing recipe steps.

We provide 129 tasks across all environments that cover essential world-model skills, including prediction, planning, and counterfactual reasoning. These tasks are not exhaustive, and we designed AutumnBench to be extensible. Example extensions include new grid worlds with non-Newtonian physics or tests for tool reuse and analogical reasoning. For example, an agent might be tested to see if it can recognize that a wine bottle can substitute for a rolling pin when one is unavailable. Finally, we validate AutumnBench with an empirical study involving 517 humans and three state-of-the-art AI models, showing that it reveals the gap between AI and humans in world-model learning.

**Contributions**   We present the following contributions:

- We propose *WorldTest*, the first theoretical world-model learning assessment framework that is interaction-centric and supports environment-level queries about the learned world model.

- We release *AutumnBench*, an instantiation of WorldTest with 43 interactive environments and three challenge families, resulting in 129 tasks. AutumnBench is designed following the desiderata for novel games outlined in Ying et al. (2025), is easily extensible, and supports automated evaluation for both human and AI models to accelerate progress in world-model learning.

- We evaluate 517 human participants and three state-of-the-art reasoning models (Anthropic Claude, OpenAI o3, and Google Gemini 2.5 Pro) on AutumnBench. We analyze their interaction trajectories and performance on challenge tasks, revealing substantial headroom for the reasoning models.

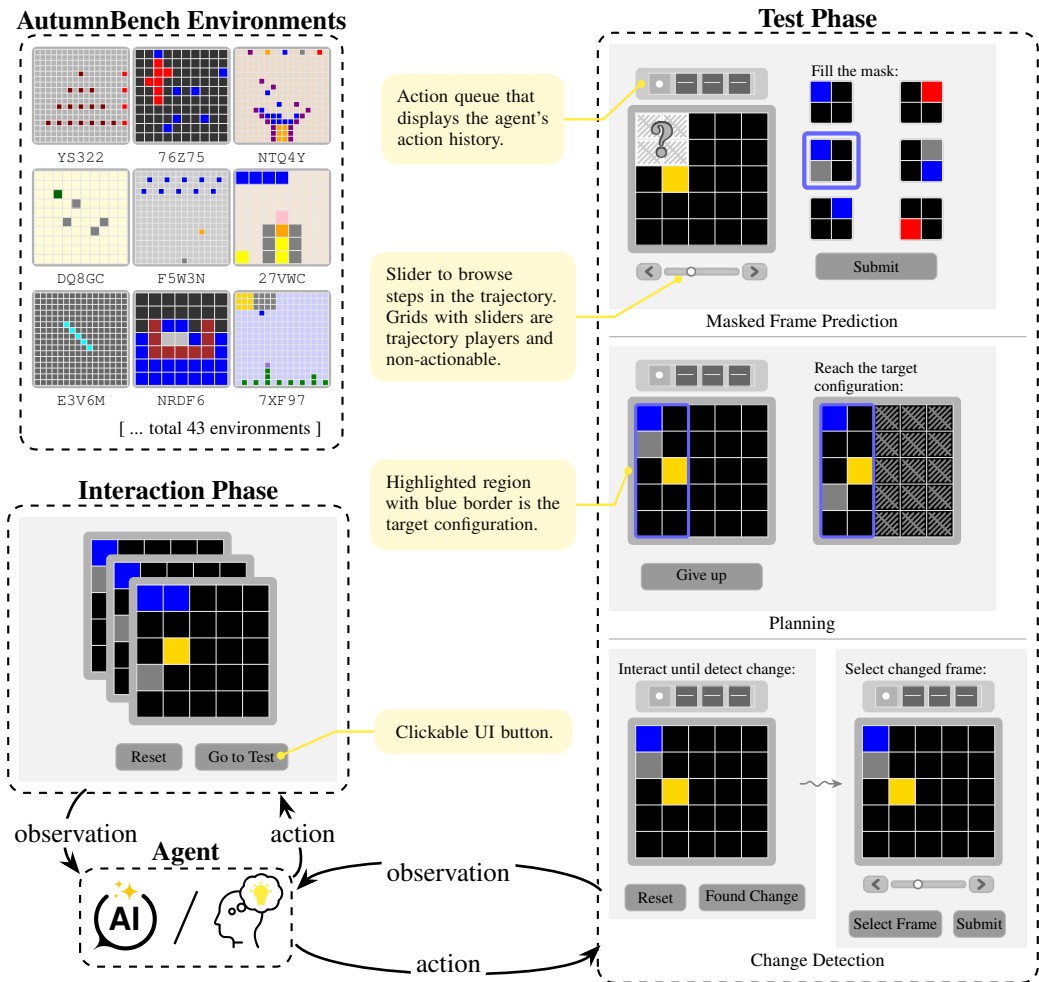

Figure 1: Overview of the WorldTest framework and the AutumnBench instantiation. Agents first interact with an environment without external rewards to build a world model, then are evaluated on a derived challenge. Top-left box shows the 9 example AutumnBench environments. Yellow notes in the middle explain the key UI elements in the human interface of AutumnBench.

In the remainder of the paper, Section 2 reviews related work around addressing the core challenge of environment-level queries, Section 3 provides background on the Autumn language, which is used for defining the environments, and the Partially Observed Markov Decision Process (POMDP) formulation of the environments in WorldTest and AutumnBench. Section 4 describes the core WorldTest framework and AutumnBench. Section 5 presents evaluations and analysis of the baseline agents on AutumnBench. Finally, Section 6 concludes and discusses future work.

## 2 RELATED WORKS

We categorize approaches to evaluating world-model learning into four non-exclusive bins, detailed in Table 1. Each represents a different and incomplete way of posing environment-level queries that probe what an agent has learned. Specifically, non-interactive benchmarks test rule-learning from static examples without time-varying dynamics; representation-based approaches probe environment-level properties but constrain models' outputs to fixed formats; gym-like benchmarks measure task-success rather than the quality of learned models; and unsupervised RL benchmarks decouple exploration from evaluation but restrict both to the same environment. We explain each bin below:

- *Non-interactive benchmarks* test whether agents can infer underlying rules from examples and generalize to novel test cases (Chollet, 2019; Depeweg et al., 2018; Barrett et al., 2018; Kim et al., 2023; Zhang et al., 2020; Yi et al., 2020a). While they assess various forms of environment-level reasoning: analogy-based concept induction from positive/negative sets (Depeweg et al., 2018),

Table 1: Comparison of benchmarks in the literature. MDP refers to Markov Decision Process, DET-POMDP refers to Deterministic-Partially Observable Markov Decision Process, and POMDP refers to Partially Observable Markov Decision Process. POMDP is the most general type of environment.

| Benchmark | Environment | Not Representation-Based | Not Gym-like | Modified Test Environment |
|---|---|---|---|---|
| VBench Huang et al. (2024) | Static | ✗ | ✓ | ✓ |
| SVIB Kim et al. (2023) | Static | ✗ | ✓ | ✓ |
| CLEVRER Yi et al. (2020a) | Static | ✗ | ✓ | ✓ |
| ACRE Zhang et al. (2020) | Static | ✓ | ✓ | ✓ |
| RAVEN Zhang et al. (2019) | Static | ✓ | ✓ | ✓ |
| PGM Barrett et al. (2018) | Static | ✓ | ✓ | ✓ |
| BONGARD-LOGO Depeweg et al. (2018) | Static | ✓ | ✓ | ✓ |
| ARC-AGI Chollet (2019) | Static | ✓ | ✓ | ✓ |
| PUZZLES Estermann et al. (2024) | MDP | ✓ | ✗ | ✓ |
| Procgen Cobbe et al. (2019) | DET-POMDP | ✓ | ✗ | ✓ |
| DiscoveryWorld Jansen et al. (2024) | POMDP | ✗ | ✓ | ✓ |
| Alchemy Wang et al. (2021) | POMDP | ✓ | ✗ | ✗ |
| CausalWorld Ahmed et al. (2021) | POMDP | ✓ | ✗ | ✓ |
| PHYRE Bakhtin et al. (2019) | POMDP | ✓ | ✗ | ✓ |
| NetHack Küttler et al. (2023) | POMDP | ✓ | ✗ | ✗ |
| MiniHack Samvelyan et al. (2021) | POMDP | ✓ | ✗ | ✗ |
| Atari Bellemare et al. (2013) | POMDP | ✓ | ✗ | ✗ |
| URLB Laskin et al. (2021) | POMDP | ✓ | ✓ | ✗ |
| **AutumnBench (Ours)** | **POMDP** | ✓ | ✓ | ✓ |

causal reasoning (Zhang et al., 2020; Yi et al., 2020a), and rule induction from examples (Chollet, 2019), they cannot evaluate learning through interaction with dynamic environments.

- *Representation-based approaches* probe environment-level properties by requiring models to use specific predefined formats for the outputs of the query—next-frame prediction, programs, or causal graphs—then measuring performance via format-specific proxies: pixel-level reconstruction error (Kamada & Ichimura, 2024; Hu et al., 2024), LLM-based evaluation (Jansen et al., 2024), or predicate prediction accuracy (Yi et al., 2020b; Girdhar & Ramanan, 2020; Ahmed et al., 2021). While this enables testing particular aspects of world understanding, reliance on potentially inadequate proxies prevents faithfully evaluating learned world models and benchmarking against human capabilities.

- *Gym-like benchmarks* provide decision-making environments with explicit objectives like rewards (Brockman et al., 2016; Bellemare et al., 2013; Cobbe et al., 2019; Küttler et al., 2023; Samvelyan et al., 2021; Ahmed et al., 2021; Wang et al., 2021; Tunyasuvunakool et al., 2020). These measure how well agents accomplish tasks rather than assessing the world models they construct. High performance may arise from memorized policies rather than a generalizable understanding of the environment structure.

- *Unsupervised RL benchmarks* separate learning from evaluation through two-phase protocols (Laskin et al., 2021): agents first explore without objectives, then face downstream tasks. However, both phases occur in the same environment, limiting evaluation to properties observable through action sequences and their rewards. This precludes testing the agent's understanding of structural properties, such as regional connectivity or counterfactual environmental variations, that it cannot reason about through trajectories.

## 3 BACKGROUND

This section provides background on the POMDP formulation of the environments in AutumnBench and Autumn, as well as the language used to implement these POMDP environments.

**Partially Observable Markov Decision Processes (POMDPs).** We formulate environments as (reward-free) POMDPs. A POMDP is a tuple $\langle \mathcal{S}, \mathcal{A}, \mathcal{O}, \mathcal{T}, \Omega \rangle$, where $\mathcal{S}$ is the (hidden) state space, $\mathcal{A}$ is the action space, $\mathcal{O}$ is the observation space, $\mathcal{T} : \mathcal{S} \times \mathcal{A} \to \Delta(\mathcal{S})$ is the transition function that maps state-action pairs to a distribution over the next states , and $\Omega : \mathcal{S} \times \mathcal{A} \to \Delta(\mathcal{O})$ is the observation function that maps from state-action pairs to a distribution over observations. Agents interact with the environment by choosing actions and receiving observations at each step.

**The Autumn Language.** We implement the POMDP environments in AutumnBench using the Autumn domain-specific language (DSL). Autumn is a functional reactive language for specifying

causal interactions in 2D grids, introduced in Das et al. (2023). We chose Autumn as the language for AutumnBench because it allows for succinct and expressive specification of the environments, is easy to extend to implement downstream challenges, and supports both a text-based Gym-like interface for evaluating AI agents and a browser-based graphical user interface for evaluating human agents. We provide a detailed description of the Autumn language in Appendix A.

## 4 THEORETICAL FRAMEWORK AND BENCHMARK

We introduce *WorldTest*, a behavior-based framework that evaluates an agent's ability to infer and use models of the environment. We then present an instantiation of WorldTest, *AutumnBench*, as a suite of 43 environments written in the Autumn language, each with three types of challenges. The following sections describe the WorldTest framework and its realization in AutumnBench.

### 4.1 WORLDTEST FRAMEWORK

We first outline the intuition of the framework and then provide a formal definition.

**Intuition.** We design WorldTest to evaluate world-model learning without constraining either the agent's internal representation or the agent's reward. To this end, WorldTest follows a two-phase protocol that assesses the agent purely through its behavior:

1. **Interaction phase**: The agent interacts autonomously with the environment, selecting actions without objectives or external rewards. During this phase, the agent can reset the environment to its initial state as many times as needed to facilitate hypothesis testing and systematic exploration. The phase ends when the agent elects to proceed to the next.

2. **Test phase**: The evaluation protocol presents a challenge environment derived from the original by modifying one or more POMDP components, such as states, transitions, or actions. These environments add explicit rewards or goals, unlike the earlier reward-free phase. They test whether an agent's learned world model can generalize across different aspects of understanding the environment. The agent must act in the challenge environment to achieve the stated objective.

We give the formal definition of this framework below.

**Formal Definition.** Let $\mathcal{M} = \langle S, A, O, T, \Omega \rangle$ denote the reward-free base environment whose dynamics are unknown to the agent. Let $\Xi$ be a task parameter space with associated distribution $P_\Xi$. A WorldTest evaluation protocol consists of $P_\Xi$ and a deterministic function

$$\tau : (\mathcal{M}, \xi) \mapsto (\mathcal{M}', R, H)$$

where $\xi \sim P_\Xi$ is a task parameter, $\mathcal{M}' = \langle S', A', O', T', \Omega' \rangle$ is a derived challenge environment, $R : (O' \times A)^H \to \mathbb{R}$ is an objective function, and $H \in \mathbb{N}$ is the evaluation horizon. The challenge environment $\mathcal{M}'$ differs from $\mathcal{M}$ through modifications to the state space, dynamics, observations, or the addition of rewards. Different definitions of $\tau$ correspond to different task types (for example, masked frame prediction, planning, and change detection in AutumnBench).

We describe a concrete run of the protocol by detailing the actions of both the protocol and the agent at each step. Given an environment $\mathcal{M}$ and a protocol $(\tau, P_\Xi)$, the evaluation proceeds as follows:

**Step 1. Disclose task type.** The protocol discloses the task type and structure by describing the transformation function $\tau$, which reveals the challenge action space $A'$, the challenge observation space $O'$, and the form of the reward function $R$ (e.g., binary success or continuous rewards with penalties). The protocol does **not** disclose the task parameters $\Xi$, the modified dynamics $(T', \Omega')$, the explicit reward targets, or any actual observations from $\mathcal{M}'$.

**Step 2. Interact with the environment.** The agent explores $\mathcal{M}$ without external rewards, interacting freely with the environment. At any time, the agent may reset the environment to its initial state or proceed to the test phase. During this phase, the agent collects an interaction history and constructs an internal model $\widehat{\mathcal{M}}$ that generalizes to the challenge action and observation spaces $A'$ and $O'$ for the given $\tau$.

**Step 3. Instantiate the protocol.** When the agent decides to go to the test, the protocol samples task parameters $\xi \sim P_\Xi$ and computes $(\mathcal{M}', R, H) = \tau(\mathcal{M}, \xi)$.

**Step 4. Produce a policy from the internal model.** Using its internal model $\widehat{\mathcal{M}}$, and knowledge of $\tau$, the agent returns a policy $\pi : (O')^* \to \Delta(A')$.

**Step 5. Execute the policy in the challenge environment.** The agent runs $\pi$ in $\mathcal{M}'$ for $H$ steps, obtaining a history $h' = (o'_0, a'_0, \ldots, o'_H, a'_H)$.

**Step 6. Score.** The protocol computes the score $R(h')$.

The interaction phase imposes no time limit, but agents only receive scores if they proceed to and complete the test phase. While, in theory, an agent might interact indefinitely, WorldTest conditions evaluation on agents eventually moving on to testing. In practice, however, finite resources prevent indefinite exploration.

### 4.2 AUTUMNBENCH: DESIGN AND IMPLEMENTATION

In this section, we present an instantiation of the WorldTest framework, namely, AutumnBench. We instantiate WorldTest by implementing three $\tau$ functions for three task-types: masked frame prediction ($\tau_{\mathrm{MFP}}$), planning ($\tau_{\mathrm{PL}}$), and change detection ($\tau_{\mathrm{CD}}$). AutumnBench consists of 129 *problems*. Each *problem* in AutumnBench has two parts: (1) a base environment with no rewards that the agent interacts with in the interaction phase, and (2) a related challenge environment for the test phase.

We describe each of those parts in more detail below, beginning with the environments used in the interaction phase and followed by the related challenge tasks for the test phase:

**Interaction-Phase Environments.** Each AutumnBench environment is a grid world composed of objects, represented as collections of pixels, along with their dynamics. We define these dynamics programmatically in AutumnBench environments using the Autumn DSL (Das et al., 2023).

The grid sizes of these environments range from $3 \times 3$ to $25 \times 25$, with most at $16 \times 16$. Each environment includes about five or fewer object types and 1 to 12 colors, with 19 of the 43 environments being stochastic. Table B.1 summarizes the complexity metrics across all environments. Our environments meet the three *novel game* desiderata from Ying et al. (2025) for evaluating world model learning: they are structurally novel, intuitive to humans, and diverse in both world dynamics and learning mechanisms. Figure 1 illustrates sample environments. AutumnBench includes Atari-like games (Bellemare et al., 2013), simulations of real-world phenomena like plant growth and sandcastle construction, and strategic games like Nim (Wikipedia, 2025). Appendix B provides further details on the different classes of environments in AutumnBench.

**Test-Phase Challenges.** We selected the test-phase challenges to address various aspects of existing world model evaluations. Specifically, AutumnBench implements the test phase of WorldTest by assessing the agent through the following three types of challenges:

- **Masked Frame Prediction (MFP)**: The agent observes a trajectory with partially masked frames and predicts the missing content in the final frame by selecting from six options, only one of which matches the ground truth. The agent receives a score of 1 for a correct selection and 0 otherwise.

- **Change Detection (CD)**: The agent interacts with a modified version of the base environment in which a rule changes dynamically. It must identify the timestep when the change is triggered. Agents receive full scores for detecting the change at the correct time; late detection incurs penalties, and selecting a timestep before the change results in a score of zero.

- **Planning**: The protocol gives the agent a goal that specifies a target state for a subgrid, and the agent is required to generate a sequence of actions to reach that target state. The agent is scored 1 or 0 based on whether or not it reaches the target state.

In AutumnBench, we implement Step 1 of the WorldTest protocol in Section 4.1 using interface-specific means: interactive tutorials for humans and text descriptions for reasoning models, further described in Section 5. However, the agents do not learn the specific task parameters $\xi$ at this stage, such as which frames are masked, when the change occurs, or which goal states are designated.

We detail the formulations of each challenge and environment in Appendix C.

## 5 BASELINE EVALUATIONS AND ANALYSIS

We evaluate humans and reasoning models on AutumnBench using the two-phase WorldTest framework. Human participants are recruited via Prolific (Prolific, 2025). For model baselines, we use Claude 4 Sonnet, Gemini 2.5 Pro, and o3. Section 5.1 details our evaluation setup for both. We additionally include an Autumn-simulator baseline agent with access to the ground-truth Autumn programs; details are provided in Section E.2.1.

We analyze agents' performance, exploration strategies during the interaction phase, and the impact of computational resources across AutumnBench tasks in Section 5.2. We then discuss these results, as well as their implications in Section 5.3.

### 5.1 EVALUATION SETUP

In this section, we describe how we tested agents on AutumnBench, including participant selection and filtering criteria, as well as the implementation details of the interface for each agent.

#### 5.1.1 HUMAN EVALUATION

We recruited 517 English-speaking participants via Prolific. To ensure the quality of our sample, we included only individuals who were not color blind and who successfully passed both attention and comprehension checks (Muszyński, 2023). We repeated each problem 20 times and distributed them uniformly among participants.

Having recruited our sample as described above, we aimed to measure the performance of a single baseline agent representing the average human who engages seriously with the task rather than to evaluate individual variation. As crowd-sourced responses exhibited high variability despite screening measures (Reid et al., 2022; Douglas et al., 2023), often reflecting differences in effort and engagement rather than cognitive capability, we constructed this agent by taking the 80th percentile score per problem across the 20 human attempts. Throughout our analysis, "human" performance refers to this single 80th-percentile aggregate agent.

We implemented AutumnBench using a web-based graphical user interface (GUI) for human participants. We provide each participant with a tutorial that teaches them how to navigate the GUI for their assigned task type. The tutorial describes the $\tau$ being used, i.e., $\tau_{\text{MFP}}$, $\tau_{\text{Planning}}$, or $\tau_{\text{CD}}$, which corresponds to Step 1 of the WorldTest protocol in Section 4.1. Figure 1 shows a rendered grid and GUI buttons for `reset` and `go-to-test` actions during the interaction phase.

The GUI updates the grid at a fixed environment-dependent frame rate of 3–8 frames per second (FPS), creating real-time experiences that mimic natural exploration. Participants engage using the directional arrow keys or by clicking grid cells at each timestep. If they do not act within the timestep, the interface automatically updates the grid state using `no-op`. The interface for the masked frame prediction, planning, and change detection tasks includes interactive features shown in Figure 1. Section D.2.1 gives the GUI implementation details for each task type.

#### 5.1.2 REASONING-MODEL EVALUATION

We evaluated three frontier reasoning models—Claude 4 Sonnet, Gemini 2.5 Pro, and o3—on AutumnBench. We gave one AutumnBench problem to each model at a time. Due to cost constraints, we evaluated each model's performance based on a single trajectory completion per problem.

At each timestep, we provide each model with complete interaction histories, current grid state, available actions, and a description of the task-type (i.e., which $\tau$ function is being applied), corresponding to Step 1 of the WorldTest protocol in Section 4.1. We represent grid states as two-dimensional arrays of color strings, see Figure 2.

While humans have an implicit time limit for taking actions, due to the fixed frame rate, reasoning models are allowed to step through the environment without a time limit. The model can still choose not to take any action in the environment through a `no-op`. Similarly to humans, during the interaction phase, the

```
[["blue",  "black", "black", "black", "black"],
 ["grey",  "black", "black", "black", "black"],
 ["black", "yellow","black", "black", "black"],
 ["black", "black", "black", "black", "black"],
 ["black", "black", "black", "black", "black"],
 ["black", "black", "black", "black", "black"]]
```

Figure 2: Textual representation of the top grid in the "interaction phase" of Figure 1.

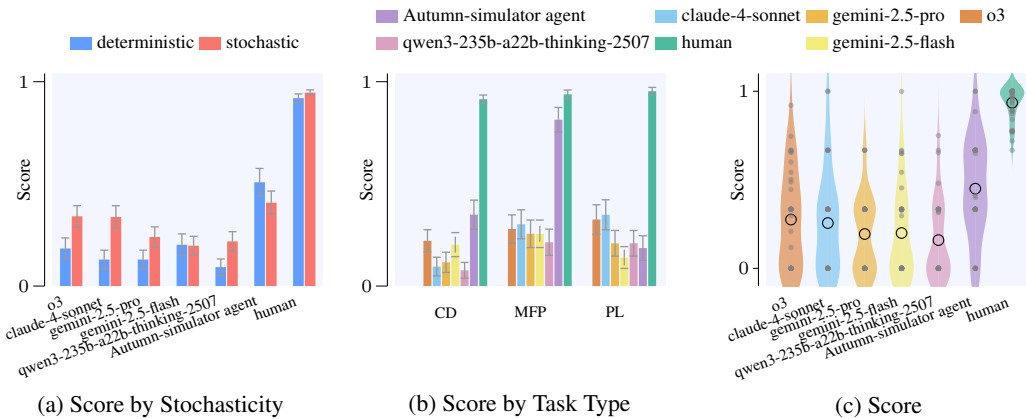

(a) Score by Stochasticity     (b) Score by Task Type     (c) Score

Figure 3: Aggregate scores over all AutumnBench tasks. Left: Reasoning models perform better in stochastic environments than in deterministic ones; humans perform consistently across both. Right: Humans outperform reasoning models across all task types: CD, MFP, and planning (PL).

models can select the `go-to-test` action to transition to the test phase and the `reset` action to reset the grid to its initial state. Section D.2.2 describes the implementation details for each task type.

## 5.2 RESULTS

We analyze agent behavior on AutumnBench in three areas: task performance, exploration patterns, and world-model learning over time. We first compare agent scores across environments and tasks. Next, we examine how the agents' exploration strategies affect task performance. Finally, we analyze the emergence of focused behavior as agents build their world models.

### 5.2.1 OVERALL PERFORMANCE

In this section, we present our comparative performance results and examine the factors that enable reasoning models to perform better. Humans outperformed all reasoning models across all environments, as shown in Figure 3b.

Table E.2 shows detailed scores for each agent and task type, and Figure 3c shows environment-wise score distributions.

To understand what factors contribute to the performance of reasoning models, we first examined how environmental stochasticity affects different agents. Humans maintained nearly identical performance in both deterministic and stochastic environments. Reasoning models, however, performed significantly better in stochastic settings than in deterministic settings, as shown in Figure 3a.

Next, we examined the impact of computational cost on model performance. We identified two environment sets based on scaling behavior: SetA, with 25 environments, where performance improved monotonically with cost, and SetB, with 18 environments, where additional compute provided no benefit, as shown in Figure 4. We detail these two sets in Appendix D. Within specific task types, increased cost improved performance in 37% of masked frame prediction and planning environments, but only 33% for change detection. In some environments, av-

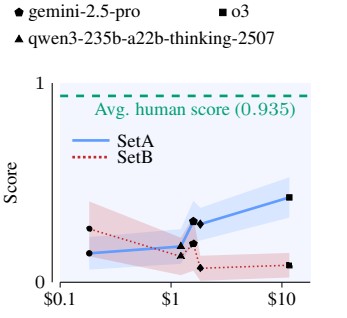

Figure 4: Score vs. cost per problem across environments.

erage performance consistently improved with additional resources across all task types, while others showed no improvement, regardless of task type, highlighting the need for better agents. Appendix E provides environment-specific analyses.

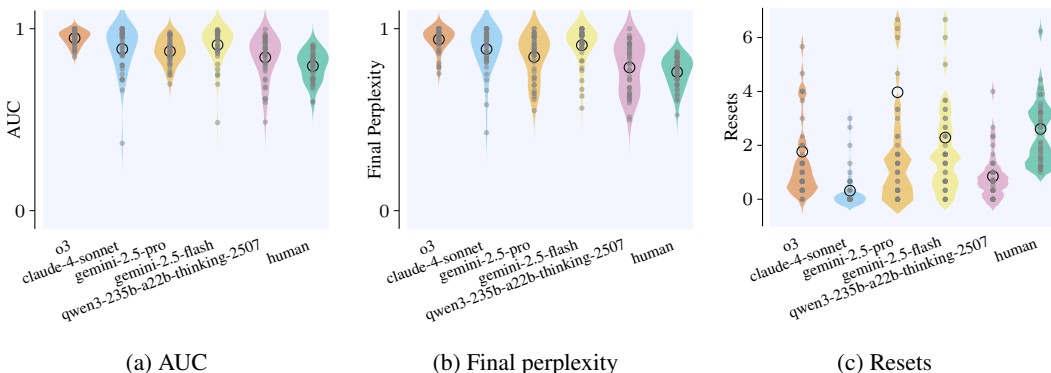

(a) AUC       (b) Final perplexity       (c) Resets

Figure 6: Environment-specific performance and behavioral metrics. (a) Task scores across environments, (b) Area Under the Curve (AUC) for normalized perplexities, (c) Final perplexity values, and (d) Reset frequency distributions across different environments.

### 5.2.2 Exploration Patterns

We analyze agents' actions during exploration using two complementary methods. First, we analyze the distribution of unique actions by counting each distinct click position and directional action separately. For example, clicking on positions $(2, 3)$ and $(5, 7)$ counts as two unique click actions. This approach allows us to account for environment-specific variations, such as differences in static object positions and grid sizes within AutumnBench. We then calculate the fraction of unique actions in each category—clicks, directional actions, `reset`, and `no-op`—relative to the total number of unique actions performed by the agent, as shown in Figure 5. Humans use roughly equal proportions of `resets`

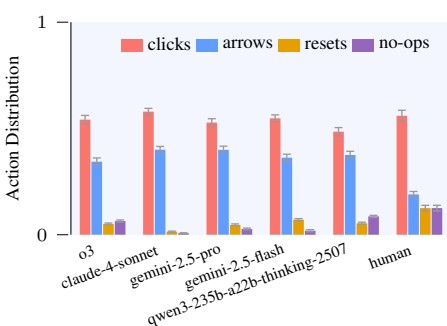

Figure 5: Action type distribution averaged over AutumnBench problems v.s. agents.

and `no-ops` at 12.5% each, while reasoning models focus on clicks and directional actions. All reasoning models use less than 7% of their actions for `resets` and `no-ops` combined. Claude uses 98.6% for clicks and arrows. Among reasoning models, o3 uses the highest fraction for `resets` and `no-ops` at 11.5%, while Claude uses the lowest at 2.1%.

Second, we examine agents' reset frequencies. Figure 6c shows humans reset more often than models, which reset less often and with greater variance.

Both analyses suggest that reasoning models do not treat resets as special actions, unlike humans. Table E.3 summarizes average actions per environment.

### 5.2.3 World-Model Learning

We quantify how agents acquire world models by measuring how their actions become more focused over time. Specifically, we use *normalized perplexity*, which we define in Definition D.1, to quantify how predictable an agent's actions are while exploring. High perplexity suggests actions are random, while low perplexity suggests more targeted behavior.

We analyze normalized perplexity across the entire interaction sequence using two metrics. The Area Under the Curve (AUC) of normalized perplexity, as defined in Definition D.2, captures the overall learning trajectory. Agents that quickly develop focused exploration strategies show lower AUC values. Final perplexity measures how targeted agents become by the end of exploration. Both metrics are robust to brief fluctuations and enable fair comparison across environments and agents.

Humans show lower values on both measures, as seen in Figures 6a and 6b, suggesting more effective world model learning. Lower AUC values indicate that they rapidly transition from random clicks and keypresses to more targeted actions. Their final normalized perplexity values are consistently

lower than those of reasoning models, suggesting more deterministic and purposeful behavior by the end of the exploration. Figure 6 visualizes these trends across environments.

### 5.3 DISCUSSION

Our results reveal differences in how humans and reasoning models approach world-model learning in two key ways: experimental design and belief updating. These differences reflect limitations in metacognitive capabilities—the ability of an agent to monitor and update its own learning process, including deciding what information to seek and when to revise beliefs based on new evidence.

**Experimental design.** We conjecture that humans use resets as experimental tools to test hypotheses about environmental dynamics. Reasoning models also form and test hypotheses, but their reasoning traces suggest a narrow view of what counts as informative actions, as illustrated in Listing E.1. They prioritize keypresses and clicks, failing to recognize that `resets` and `no-ops` can be equally valuable for hypothesis testing and generating informative observations. This limitation manifests quantitatively: humans use resets in 12.5% of their actions, as seen in Figure 5, while reasoning models use them in fewer than 7%, with Claude using only 2.1%. This limited view of action informativeness leads reasoning models to miss opportunities for more effective exploration.

**Belief updating.** Reasoning models often fail to update their understanding when faced with contradictory evidence, especially in masked frame prediction tasks (see Listing E.1). Even when they recognize that test-phase observations contradict the rules learned during interaction, they tend to rely on those original rules in their predictions.

Collectively, these results demonstrate that humans outperform current reasoning models due to limitations at multiple levels of inference. Rather than simply requiring better priors over world models, which most current training approaches focus on, our findings suggest that achieving human-level performance may require advances in metacognitive capabilities—including strategic experimental design, uncertainty quantification, and flexible belief updating—beyond improvements in inductive reasoning and memory alone.

## 6 CONCLUSION

In this work, we present WorldTest, a behavior-based framework that separates a goal-free interaction phase from a scored test phase. Unlike prior approaches that constrain agent representations to specific formats or evaluate only through reward optimization, WorldTest uses environment-level queries to assess whether agents have learned transferable world models. By scoring only behavior in derived challenges, WorldTest evaluates how agents apply their learned understanding to new tasks while enabling fair comparison across different agent types, including humans.

Instantiating WorldTest, AutumnBench provides 43 environments and three derived challenge families, resulting in 129 tasks. These challenges mirror the key capabilities of world models: masked-frame prediction tests forecasting of latent dynamics, change detection evaluates recognition of changes to environmental dynamics, and planning assesses goal-directed action sequencing. This unified framework enables cross-comparison across different agents and tasks.

Our experiments with 517 human participants and five frontier reasoning models on AutumnBench reveal substantial gaps between human and AI world-model-learning capabilities. Humans outperform these models across all environments and task types, achieving near-optimal scores while the models frequently fail. We also observed that humans reset the environment more often than the models and achieved lower perplexity over their trajectories, suggesting a possible link between humans' use of resets as an exploration or hypothesis-testing mechanism and their ability to refine internal models of the world during interaction better than reasoning models. These findings indicate that current frontier models lack the flexible, predictive understanding that characterizes human-like world models.

While AutumnBench demonstrates WorldTest in a grid-world setting, the WorldTest framework we presented is a much broader framework that can be instantiated across diverse domains: Future work may instantiate WorldTest in physics-rich environments, robotics domains, multi-agent systems, and any general domain with a rich dynamics.

## REPRODUCIBILITY STATEMENT

We provide a detailed description of the benchmark's design, scoring methodology, and theoretical framework in Section 4 to facilitate the reproduction of our experiments and results by others. In Section 5.1, we describe the evaluation setup, agent interfaces, and procedures. The appendix contains the exact prompt template used for the reasoning-model agents. Our supplementary materials include: (1) Autumn source programs for all base environments and their derived challenges, and (2) Task specifications and answer keys where applicable. (3) The code implementing AutumnBench for reasoning models. To protect the benchmark's integrity and prevent data leakage, we keep a subset of AutumnBench problems private but provide full specifications for all reported experiments and for the publicly released problems.

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

# Appendices

## A  THE AUTUMN LANGUAGE

We implement the POMDP environments in AutumnBench using the Autumn domain-specific language (DSL). Autumn is a functional reactive language for specifying causal interactions in 2D grids, introduced in Das et al. (2023). We chose Autumn as the base language for AutumnBench because it allows succinct and expressive specification of environments, is easily extensible to implement downstream challenges, and supports both a text-based Gym-like interface for evaluating AI agents and a browser-based graphical user interface for evaluating human participants.

An Autumn program has the following parts:

- *Environment setup* defines the grid size ($n \times n$) and background color.
- *Object type definitions* specify each object type by listing its shape (as relative 2D coordinates and colors) and its internal fields. These fields record state variables unique to each object instance.
- *Object instance definitions* define the initial state of each object instance and the default function for each object instance under the `no-op` action.
- *Event handlers* establish the rest of the full transition function $T$. In particular, an on clause pairs a predicate $p(s, a)$ with an intervention $i$, so that if $p(s, a)$ holds, the state transitions to $s' = i(s)$.

An Autumn program defines a POMDP. Specifically, the global variables in the program correspond to the hidden state. The agent observes only the grid's color matrix ($n \times n$), where every cell occupied by the highest-z-order object instance is visible; all other state components remain latent. The action space of an Autumn environment are: 1. the `no-op` action, 2. four directional actions—up, down, left, right—corresponding to the four arrow keys on a typical keyboard, and 3. a family of click actions click($x, y$), corresponding to clicking on the cell at position ($x, y$) by a pointer device.

**Example Autumn Environment.**  To illustrate the Autumn language, we present a treasure-hunting environment in which an agent searches for hidden treasures using a noisy distance sensor. Listing A.1 shows the complete Autumn program implementing this environment, and Figure A.1 shows a rendered trajectory sampled from it. At the start, the environment reveals only one treasure—the gold grid cell in the middle of the grid. The agent controls a sensor probe, the grey grid cell, and moves it using four directional actions: up, down, left, and right. The top blue horizontal bar displays the probe's noisy estimate of its distance to the nearest treasure. If the agent clicks a hidden treasure, the environment reveals it.

```
1   (= GRID_SIZE 5)  (= BACKGROUND "black")      ← Environment setup
2
3   (object Proximity (dist)
4     (map (-->) (i) (Cell i 0 (if (<= i dist) "blue" "black" )))
5       (range 0 GRID_SIZE)))
6   (object Treasure (revealed) (Cell 0 0 (if revealed "gold" "black")))    ← Object definitions
7   (object Agent (Cell 0 0 "grey"))
8
9   (= t1 (initnext (Treasure true (randomPosition 0 1 GRID_SIZE GRID_SIZE)) (prev t1)))
10  (= t2 (initnext (Treasure true (randomPosition 0 1 GRID_SIZE GRID_SIZE)) (prev t2)))
11  (= t3 (initnext (Treasure false (randomPosition 0 1 GRID_SIZE GRID_SIZE)) (prev t3)))    Object instances;
12  (= agent (initnext (Agent (Position 0 1)) (prev agent)))                                 they define the initial state
13  (= tdist (fn (treasure) (sqdist (.. agent origin) (.. treasure origin))))                and the default transitions
14  (= proximity (initnext (Proximity 2 (Position 0 0)) (updateObj proximity "dist"          with the no-op action
15        (+ (min (min (tdist t1) (tdist t2)) (tdist t3))
16        (- (uniformChoice (range 0 3)) 1)))))
17
18  (on up (= agent (moveUp agent)))
19  (on down (= agent (moveDown agent)))
20  (on left (= agent (moveLeft agent)))
21  (on right (= agent (moveRight agent)))                                    ← Event handlers ("on" clauses)
22  (on (clicked t1) (= t1 (updateObj t1 "revealed" true)))
23  (on (clicked t2) (= t2 (updateObj t2 "revealed" true)))
24  (on (clicked t3) (= t3 (updateObj t3 "revealed" true)))
```

Listing A.1: A treasure-hunting environment example in Autumn.

Figure A.1: Renderings of a partial trajectory of the Autumn environment example from Listing A.1, following the sequence of actions down, down, down and click(1, 4).

# B AUTUMNBENCH ENVIRONMENT DETAILS

This section provides detailed statistics and categorization of the AutumnBench environments. AutumnBench comprises 43 environments spanning six broad categories: 1. Physical simulations that follow intuitive, middle-school-level physics principles. 2. Emergent systems where complex patterns arise from simple rules. 3. Multi-agent dynamics that require reasoning about other agents' goals and interactions. 4. Abstract logic environments that follow well-defined but non-physical rules. 5. Game-inspired scenarios abstracted from popular games. 6. Tool use environments that require understanding object affordances and performing multi-step reasoning.

Table B.1 reports the number of object types, #Obj; latent variables, #Latent; program length measured in lines of code; the number of event handlers, #On Clauses; stochasticity; the number of colors, #Colors; and grid size.

# C AUTUMNBENCH CHALLENGE FORMULATION

This section provides the formal definitions of the three challenge types in AutumnBench: masked frame prediction, change detection, and planning. For each challenge type, we define the task structure, scoring function, and empirical difficulty measurements using a random agent baseline.

To characterize task difficulty, we evaluate a random agent that selects uniformly among six action types: click, up, down, left, right, and no-op. When the agent selects click, it uniformly samples a grid position. We approximate the random agent's performance using approximately 50 samples per environment with a 1000-step timeout. For masked frame prediction tasks, the random agent selects uniformly over six answer options, yielding a chance probability of 1/6. Table E.6 reports per-environment random agent baseline results.

## C.1 MASKED FRAME PREDICTION

We define each masked frame prediction task by a tuple of a fixed action sequence $\mathbf{a} = (a_1, a_2, \ldots, a_T) \in \mathcal{A}^T$ and binary masks $\mathcal{M}_{1:T}$ with $\mathcal{M}_t \subseteq [W] \times [H]$ for $t = 1, \ldots, T$ over the observable grid. Executing $\mathbf{a}$ in the environment produces the observation trajectory $\tau = (o_0, o_1, \ldots, o_T)$.

Masked-frame-prediction asks the agent to predict the colors in the masked region of the final observation, $o_T|_{\mathcal{M}}$, choosing from the candidates $K$ $\mathcal{C}_{\text{choices}} = \{c_1, c_2, \ldots, c_K\}$, where each $c_i \in \mathcal{C}^{|\mathcal{M}|}$ and $\mathcal{C}$ denotes the set of possible grid cell colors.

We construct each masked frame prediction task so that, for any realization of $\tau$, exactly one candidate $c^* \in \mathcal{C}_{\text{choices}}$ matches the ground truth, $c^* = o_T|_{\mathcal{M}}$. This guarantees a unique correct answer even when the underlying POMDP dynamics is stochastic.

We use a simple binary accuracy metric: score $= \mathbb{1}[\text{correct}]$.

## C.2 CHANGE DETECTION

We define each change detection task with an original environment $\mathcal{M}$ and a changed environment $\mathcal{M}'$, both written in the Autumn language. We write their transition and observation functions as $(\mathcal{T}, \Omega)$ and $(\mathcal{T}', \Omega')$. A boolean guard $g(s, a)$ in an Autumn event handler gates the change in $\mathcal{M}'$; $g(s, a)$ determines when $\mathcal{T}'$ deviates from $\mathcal{T}$.

Table B.1: Program statistics for all AutumnBench environments.

| Env | #Obj | #Latent | Length | #On Clauses | Stochastic | #Colors | Grid Size |
|---|---|---|---|---|---|---|---|
| VZ2Q4 | 1 | 4 | 18 | 3 | Yes | 12 | 20 |
| S2KT7 | 2 | 0 | 10 | 3 | Yes | 2 | 16 |
| 27VWC | 4 | 4 | 19 | 4 | No | 10 | 7 |
| KFQYT | 2 | 5 | 21 | 8 | No | 6 | 11 |
| B58F3 | 3 | 2 | 22 | 2 | Yes | 5 | 13 |
| NRDF6 | 3 | 1 | 25 | 1 | No | 3 | 7 |
| 6JKKA | 5 | 8 | 54 | 6 | No | 6 | 13 |
| K8MTQ | 2 | 2 | 32 | 3 | No | 3 | 8 |
| T5F9B | 5 | 6 | 28 | 6 | No | 8 | 7 |
| N59TE | 1 | 4 | 21 | 3 | Yes | 5 | 10 |
| 76Z75 | 1 | 1 | 8 | 1 | Yes | 2 | 10 |
| NF5VZ | 3 | 1 | 18 | 3 | Yes | 3 | 9 |
| 236VK | 3 | 1 | 18 | 6 | Yes | 4 | 20 |
| DQ8GC | 1 | 1 | 14 | 6 | No | 2 | 16 |
| QM9XB | 3 | 3 | 24 | 7 | No | 5 | 16 |
| 6JVMF | 1 | 0 | 7 | 1 | Yes | 1 | 16 |
| 27JBD | 2 | 2 | 17 | 3 | No | 4 | 16 |
| VQJH6 | 2 | 2 | 24 | 9 | No | 5 | 17 |
| QQM74 | 2 | 3 | 16 | 6 | No | 2 | 21 |
| 7XF97 | 4 | 2 | 24 | 8 | No | 5 | 16 |
| BT3GB | 3 | 2 | 18 | 4 | No | 4 | 16 |
| BT2KZ | 1 | 2 | 34 | 1 | Yes | 5 | 25 |
| B8AKZ | 2 | 3 | 15 | 3 | No | 5 | 9 |
| 9F8AJ | 4 | 3 | 34 | 6 | No | 8 | 24 |
| 7WWW9 | 1 | 1 | 22 | 8 | No | 2 | 16 |
| N2NTD | 5 | 5 | 34 | 10 | No | 6 | 12 |
| BY2Q7 | 4 | 4 | 52 | 2 | Yes | 12 | 12 |
| 4T8TR | 2 | 3 | 8 | 1 | Yes | 3 | 10 |
| YS322 | 2 | 1 | 27 | 8 | No | 3 | 17 |
| WHGHP | 4 | 4 | 42 | 6 | Yes | 6 | 10 |
| EAHCW | 1 | 3 | 17 | 9 | No | 5 | 16 |
| 83WKQ | 1 | 0 | 5 | 1 | Yes | 1 | 16 |
| 3J4Z7 | 1 | 4 | 17 | 1 | Yes | 4 | 5 |
| QDHS3 | 4 | 6 | 141 | 36 | No | 6 | 24 |
| VA6FQ | 3 | 3 | 23 | 5 | No | 5 | 10 |
| 7VKTD | 3 | 2 | 57 | 5 | Yes | 3 | 20 |
| XHGKQ | 2 | 1 | 24 | 4 | Yes | 3 | 16 |
| F5W3N | 4 | 2 | 35 | 14 | Yes | 5 | 16 |
| JXQAW | 1 | 3 | 38 | 2 | No | 4 | 5 |
| 4N7BB | 1 | 2 | 38 | 5 | Yes | 9 | 3 |
| NTQ4Y | 4 | 2 | 34 | 9 | No | 5 | 16 |
| DGG2C | 2 | 2 | 18 | 6 | No | 2 | 17 |
| 4CKC2 | 3 | 4 | 36 | 2 | No | 6 | 24 |

We fix an initial state distribution and an action sequence $\mathbf{a} = (a_1, \ldots, a_T)$. Executing $\mathbf{a}$ in $\mathcal{M}'$ produces a realized observation prefix $o'_{0:T}$. We define the defect time $t^*$ as the first step where this realized prefix leaves the support of $\mathcal{M}$:

$$t^* = \min\Big\{ t \geq 1 : \underbrace{\mathbb{P}_{\mathcal{M}}(o_{0:t} = o'_{0:t} \mid a_{1:t})}_{=0} \text{ and } \underbrace{\mathbb{P}_{\mathcal{M}}(o_{0:t-1} = o'_{0:t-1} \mid a_{1:t-1})}_{>0} \Big\}.$$

Equivalently, $t^*$ is the earliest time when $o'_{0:t}$ has zero probability under $\mathcal{M}$ while $o'_{0:t-1}$ still has positive probability under $\mathcal{M}$. We require the guard to trigger exactly at $t^*$: $g(s'_{t^*-1}, a_{t^*}) = \text{true}$ and $g(s'_{t-1}, a_t) = \text{false}$ for all $t < t^*$.

The agent outputs a time index $t$ that marks the detected change.

$$\text{score}(t) = \begin{cases} 0 & \text{if } t < (t^* - 1), \\ 1 & \text{if } t \in \{t^* - 1, \ t^*\}, \\ 1.377 \cdot f_{t^*}(t) - 1.178 & \text{otherwise.} \end{cases}$$

We set $f_{t^*}(t) = \frac{1}{1 - \frac{t}{t^*} \cdot e^{-\frac{t}{t^*}}}$; $t^*$ denotes the defect time. This rule gives full credit for exact or one-step-early detection and smoothly penalizes late detections.

Across our 43 change detection environments, the random agent triggers changes with mean probability 0.80 in an average of 295 steps. Twenty-four environments trigger changes with probability 1.0, while 5 environments require 900 or more steps before changes occur, and 2 environments never trigger changes within the 1000-step timeout.

## C.3 PLANNING

We define each planning task by a goal specification $(\mathcal{S}, g)$, where $\mathcal{S} \subseteq [W] \times [H]$ indexes a subgrid, and $g \in \mathcal{C}^{|\mathcal{S}|}$ specifies the target colors on $\mathcal{S}$. The agent must produce an action sequence $\mathbf{a}$ that, when executed, produces a final observation $o_T$ that matches the goal on the subgrid, that is, $o_T|_{\mathcal{S}} = g$. We use a simple binary success metric: $\text{score} = \mathbb{1}[o_T|_{\mathcal{S}} = g]$.

Across our 43 planning environments, the random agent reaches the goal with mean probability 0.399 in an average of 679 steps. The random agent reaches the goal with probability 1.0 in nine environments, while 13 environments remain unsolved at the 1000-step timeout.

## D    IMPLEMENTATION DETAILS

This section provides detailed definitions of the behavioral metrics and interface implementations used in our experiments. We first define normalized perplexity and the associated area under the curve (AUC), which quantify how agents' actions become more focused over time. We then describe the task-specific interface implementations for both the graphical user interface used by human participants and the text-based interface used by reasoning models.

### D.1    NORMALIZED PERPLEXITY

**Definition D.1 (Normalized Perplexity)** *Let $p(a)$ be the empirical action distribution over an active alphabet of size $K$ within a sliding window. Given entropy $H(p) = -\sum_a p(a) \log_2 p(a)$ and perplexity $P = 2^{H(p)}$, normalized perplexity is:*

$$Perplexity_{norm} = \frac{P - 1}{K - 1} \in [0, 1]$$

*where $K$ is the number of unique actions observed in the window, including directional keys and click positions.*

**Definition D.2 (Area Under the Curve (AUC) of Normalized Perplexity)** *The AUC of normalized perplexity is computed as:*

$$AUC = \int_0^1 Perplexity_{norm}(x)\, dx$$

*where $x \in [0, 1]$ is the normalized interaction position.*

**Definition D.3 (Final Normalized Perplexity)** *The final normalized perplexity is the value of Perplexity$_{norm}$ evaluated at the last window of the interaction sequence.*

### D.2 TASK INTERFACE IMPLEMENTATION

We implemented distinct interfaces for human participants and reasoning models. Human participants interact with environments through a graphical user interface (GUI), while reasoning models use a text-based interface. The following sections detail the implementation of each interface type across the three task families.

#### D.2.1 GRAPHICAL USER INTERFACE

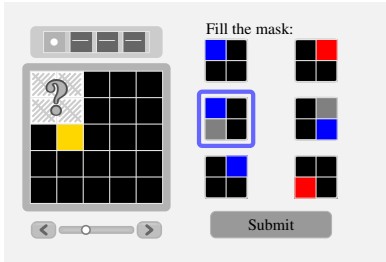

Figure D.1: Example of masked frame prediction task on the AutumnBench GUI showing the slider, action viewer, six choice options, and a `Submit` button.

For the masked frame prediction task, we implemented a slider to visualize the partially masked sequence of observations, as shown in Figure D.1. The GUI displays six answer choices from which participants select their prediction. Participants submit their selection using a `Submit` button.

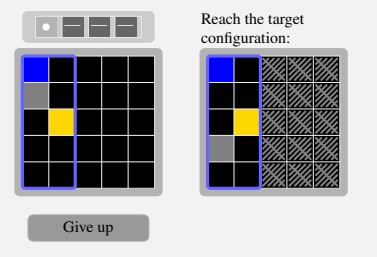

Figure D.2: Example of planning task on the AutumnBench GUI that shows a target region by masking out cells that are not in the target.

For the planning task, we display the goal state on a static grid alongside the interactive grid, with non-target regions shaded out, as shown in Figure D.2.

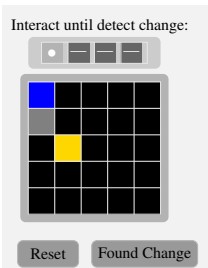
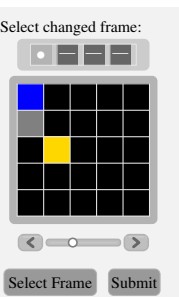

Figure D.3: Change detection on the AutumnBench GUI. (a) Initial change detection phase with `Reset` and `Found change` buttons. (b) Frame selection phase with the action viewer, slider, and `Submit` button.

```
972   You are a helpful assistant currently operating as a curious agent exploring an environment
973   ↪   that consists of a grid containing cells which can take colors.
      You will be given observations and available actions to choose from at each step.
974   Your task is to interact with the environment efficiently and effectively and try to
975   ↪   understand the underlying rules of the environment.

976   Here is a description of the actions:
977
      - `click x y` - Click on the cell at the location (x, y) on the grid.
978   - `left` - Press the left arrow key.
979   - `right` - Press the right arrow key.
      - `up` - Press the up arrow key.
980   - `down` - Press the down arrow key.
981   - `noop` - Do nothing and continue to the next step.
      - `quit` - Quit the environment.
982   - `step` - Step through a sequence one frame at a time.
983   - `go-to-test` - Go to the test phase.
      - `reset` - Reset the environment to the initial state.
984
985   Additional actions will be described whenever available.
986
      Follow exactly the format when producing the action. So if the action is to click on a cell at
987   ↪   location (1, 2), you should provide the action as <action>click 1 2</action>.
```

Listing D.1: System prompt for all reasoning models

For the change detection task, we provide a `Found change` button below the interactive grid, as shown in Figure D.3. Clicking this button transitions participants to the selection phase, where they use a slider to review observations from the test phase and identify the earliest changed frame. Participants submit their selection using a `Submit` button.

### D.2.2   TEXT-BASED INTERFACE

All reasoning models receive the same text-based prompts, with the system prompt shown in Listing D.1.

For the masked frame prediction task, we add two actions in the test phase: `step` and `rewind`, which allow the model to navigate through the partially masked observations. We provide six answer options as matrices of color strings, which the model selects using the `choose_option_i` action.

For the planning task, we provide the target configuration in the same format as the grid observation at every timestep of the test phase.

For the change detection task, we provide a `Found the change` action during the test phase. Once the model selects this action, we present `choose_grid_i` actions enumerating the observations from the test phase. After the model chooses a grid, we display it in the subsequent prompt, allowing the model to confirm its selection using the `submit` action.

Example prompts for all three task types are provided in Listings D.2 to D.4.

## E   ADDITIONAL RESULTS

This section provides comprehensive supplementary analyses to support and extend the main results. We organize the material as follows: Appendix E.1 formalizes the three task types through the lens of next-state prediction models; Appendix E.2 introduces both an Autumn-simulator agent and a random agent to establish performance bounds; Appendix E.3 presents detailed per-environment scores and examines how performance scales with computational cost and varies across prompts; Appendix E.4 analyzes agent action distributions and correlations between perplexity metrics and task performance; and finally Appendix E.5 provides representative agent traces to illustrate qualitative reasoning behavior.

A key finding is that despite complete access to ground-truth environment transitions, the simulator agent significantly underperforms humans on all three task types: masked-frame prediction, change detection, and planning. This gap highlights the need for better inference under aleatoric uncertainty and long-horizon planning.

```
[
    {
      "role": "user",
      "content": "Welcome, you are now in the interactive phase, where you can interact with
    ↪    the grid using the available actions.\nDuring the interactive phase your goal is
    ↪    to act in the environment to understand the underlying rules of the environment.
    ↪    \nUnderstand the environment and the dynamics of the environment well. Once you
    ↪    have understood the environment, you can select 'go-to-test' to go to the test
    ↪    phase.\nAfter the interactive phase you will be asked to use this knowledge about
    ↪    the environment to answer some questions about it.\n\n# Task Description:\nIn the
    ↪    test phase, you will step through frames from a trajectory in this same
    ↪    environment you interacted with (use the `step` action to step through the
    ↪    trajectory). Each frame is structured as a json object with the following
    ↪    fields:\n\"render\": the grid observed,\n\"video_location\": timestep at which the
    ↪    frame was observed,\n\"action_took\": action taken at this
    ↪    timestep,\n\"is_finished\": whether the episode is finished.\nYou will step
    ↪    through the trajectory one frame at a time. Towards the end of the trajectory,
    ↪    parts of the grid will be masked (where the masked locations are marked as `mask`)
    ↪    and you will be given a set of choices to fill in the masked region at the final
    ↪    timestep. You need to choose option that fits the masked region at the final
    ↪    timestep.\n\n\nHere is the initial state of the grid: \n[[\"black\", \"black\",
    ↪    \"red\", ...], ...]\n\nThe following actions are available at this step:
    ↪    left,\nright,\nup,\ndown,\nclick [0-9]
    ↪    [0-9],\nnoop,\nquit,\ngo-to-test\nreset\nThink step by step about the next action
    ↪    that should be taken. Remember, you are exploring the environment and trying to
    ↪    understand the underlying rules. Reflect on your action and self evaluate any
    ↪    potential issues before selecting the action. Output your final choice of action
    ↪    within a <action> tag.\nAdditionally, you can modify the contents of the
    ↪    scratchpad to use as memory since you can only observe the most recent
    ↪    states.\nPlease include the additions to the scratchpad withing <scratchpad_add>
    ↪    tags and deletions withing <scratchpad_del> tags. Output your choice of action
    ↪    within a <action> tag."
    },
    ...,
    {
      "role": "user",
      "content": "The interaction phase is now over. You will now step through frames from a
    ↪    trajectory in this same environment you interacted with (use the `step` action to
    ↪    step through the trajectory). Each frame is structured as a json object with the
    ↪    following fields:\n\"render\": the grid observed,\n\"video_location\": timestep at
    ↪    which the frame was observed,\n\"action_took\": action taken at this
    ↪    timestep,\n\"is_finished\": whether the episode is finished.\nYou will step
    ↪    through the trajectory one frame at a time. Towards the end of the trajectory,
    ↪    parts of the grid will be masked (where the masked locations are marked as `mask`)
    ↪    and you will be given a set of choices to fill in the masked region at the final
    ↪    timestep. You need to choose option that fits the masked region at the final
    ↪    timestep. You can also use the `rewind` action to go back to the previous
    ↪    frame.\n{\"video_location\": \"0/21\", \"render\": [[\"black\", \"black\", \"red\",
    ↪    \"black\", ...], ...], \"action_took\": \"start\", \"is_finished\": false}\n\nThe
    ↪    following actions are available at this step: step\nThink step by step about the
    ↪    next action that should be taken. Remember, you are exploring the environment and
    ↪    trying to understand the underlying rules. Reflect on your action and self
    ↪    evaluate any potential issues before selecting the action. Output your final
    ↪    choice of action within a <action> tag.\nAdditionally, you can modify the contents
    ↪    of the scratchpad to use as memory since you can only observe the most recent
    ↪    states.\nPlease include the additions to the scratchpad withing <scratchpad_add>
    ↪    tags and deletions withing <scratchpad_del> tags. Output your choice of action
    ↪    within a <action> tag."
    }
]
```

Listing D.2: Prompt for masked frame prediction task

```
1080    [
1081        {
1082            "role": "user",
1083            "content": "Welcome, you are now in the interactive phase, where you can interact
                ↪    with the grid using the available actions.\nDuring the interactive phase your
                ↪    goal is to act in the environment to understand the underlying rules of the
                ↪    environment. You can reset the environment to it's initial state at any
                ↪    time.\nUnderstand the environment and the dynamics of the environment well.
                ↪    Once you have understood the environment, you can select 'go-to-test' to go to
                ↪    the test phase.\nAfter the interactive phase you will be asked to use this
                ↪    knowledge about the environment to answer some questions about it.\n\n# Task
                ↪    Description:\nIn the test phase, you will interact with a changed version of
                ↪    the environment – where one of the dynamics rules has been changed. \nYour
                ↪    goal is to use you understanding of the environemnt from the interaction phase
                ↪    to detect the change. The environment will start in a normal state and then at
                ↪    some point, the environment will transition to a defective state. \nAs soon as
                ↪    you detect the change, you have to select 'I found the change!' action to go
                ↪    to the next phase, wherein you have to choose exactly which frame the change
                ↪    occurred, then submit it. You may choose as many times as you want to see the
                ↪    frames. You will be penalized if you click 'I found the change!' before the
                ↪    change is detected. \n\n\nHere is the initial state of the grid: \n[[\"black\",
                ↪    \"black\", \"black\", ...], ...]\n\nThe following actions are available at
                ↪    this step: left,\nright,\nup,\ndown,\nclick [0-15]
                ↪    [0-15],\nnoop,\nquit,\ngo-to-test,\nreset\nThink step by step about the next
                ↪    action that should be taken. Remember, you are exploring the environment and
                ↪    trying to understand the underlying rules. Reflect on your action and self
                ↪    evaluate any potential issues before selecting the action. Output your final
                ↪    choice of action within a <action> tag.\nAdditionally, you can modify the
                ↪    contents of the scratchpad to use as memory since you can only observe the
                ↪    most recent states.\nPlease include the additions to the scratchpad withing
                ↪    <scratchpad_add> tags and deletions withing <scratchpad_del> tags. Output your
                ↪    choice of action within a <action> tag."
1102        },
            ...,
1103        {
1104            "role": "user",
            "content": "You are now in the test phase.You will now interact with a changed
                ↪    version of the environment – where one of the dynamics rules has been changed.
                ↪    Your goal is to use you understanding of the environemnt from the interaction
                ↪    phase to detect the change. The environment will start in a normal state and
                ↪    then at some point, the environment will transition to a defective state. As
                ↪    soon as you detect the change, you have to select 'I found the change!' action
                ↪    to go to the next phase, wherein you have to choose exactly which frame the
                ↪    change occurred, then submit it. You may choose as many times as you want to
                ↪    see the frames. You will be penalized if you click 'I found the change!'
                ↪    before the change is detected. Here is the initial frame: [[\"black\",
                ↪    \"black\", \"black\", ...], ...]\n\nThe following actions are available at
                ↪    this step: left,\nright,\nup,\ndown,\nclick [0-15] [0-15],\nnoop,\nquit,\nI
                ↪    found the change!\nreset,\nquit\nThink step by step about the next action that
                ↪    should be taken. Remember, you are exploring the environment and trying to
                ↪    understand the underlying rules. Reflect on your action and self evaluate any
                ↪    potential issues before selecting the action. Output your final choice of
                ↪    action within a <action> tag.\nAdditionally, you can modify the contents of
                ↪    the scratchpad to use as memory since you can only observe the most recent
                ↪    states.\nPlease include the additions to the scratchpad withing
                ↪    <scratchpad_add> tags and deletions withing <scratchpad_del> tags. Output your
                ↪    choice of action within a <action> tag."
1118        },
            ...
1119        {
1120            "role": "user",
1121            "content": "You are now in the test phase.You will now interact with a changed
                ↪    version of the environment – where one of the dynamics rules has been changed.
                ↪    Your goal is to use you understanding of the environemnt from the interaction
                ↪    phase to detect the change. The environment will start in a normal state and
                ↪    then at some point, the environment will transition to a defective state. As
                ↪    soon as you detect the change, you have to select 'I found the change!' action
                ↪    to go to the next phase, wherein you have to choose exactly which frame the
                ↪    change occurred, then submit it. You may choose as many times as you want to
                ↪    see the frames. You will be penalized if you click 'I found the change!'
                ↪    before the change is detected. Here is the initial frame: [[\"black\",
                ↪    \"black\", \"black\", ...], ...]\n\nThe following actions are available at
                ↪    this step: choose_frame_0,\nchoose_frame_1,\nchoose_frame_2,\nSubmit
                ↪    choice,\nquit\nThink step by step about the next action that should be taken.
                ↪    Remember, you are exploring the environment and trying to understand the
                ↪    underlying rules. Reflect on your action and self evaluate any potential
                ↪    issues before selecting the action. Output your final choice of action within
                ↪    a <action> tag.\nAdditionally, you can modify the contents of the scratchpad
                ↪    to use as memory since you can only observe the most recent states.\nPlease
                ↪    include the additions to the scratchpad withing <scratchpad_add> tags and
                ↪    deletions withing <scratchpad_del> tags. Output your choice of action within a
                ↪    <action> tag."
        }
    ]
```

Listing D.3: Prompts for change detection task

```
[
    {
        "role": "user",
        "content": "Welcome, you are now in the interactive phase, where you can interact
        ↪   with the grid using the available actions.\nDuring the interactive phase your
        ↪   goal is to act in the environment to understand the underlying rules of the
        ↪   environment. \nUnderstand the environment and the dynamics of the environment
        ↪   well. Once you have understood the environment, you can select 'go-to-test' to
        ↪   go to the test phase.\nAfter the interactive phase you will be asked to use
        ↪   this knowledge about the environment to answer some questions about it.\n\n#
        ↪   Task Description:\nIn the test phase, you will be given a goal state and a
        ↪   highlight mask of the same size as the grid where 1 indicates the region to be
        ↪   reached and 0 indicates the region to be ignored. \nYour aim is to solve a
        ↪   planning task in the environment you interacted by reaching the goal state in
        ↪   the highlighted region.\nNote that you can no longer reset the environment, so
        ↪   plan carefully. You will be given the same environment as you interacted with
        ↪   in the interaction phase, you need to interact with it to reach the goal state
        ↪   in the highlighted region.\nYour grid will be checked against the goal state
        ↪   and the highlight mask at every timestep. If you reach the goal state in the
        ↪   highlighted region, you will be given a reward. You may choose to quit at any
        ↪   time if you are stuck.\n\n\nHere is the initial state of the grid:
        ↪   \n[[\"black\", \"black\", \"black\", ...], ...]\n\nThe following actions are
        ↪   available at this step: left,\nright,\nup,\ndown,\nclick [0-15]
        ↪   [0-15],\nnoop,\nquit,\ngo-to-test,\nreset\nThink step by step about the next
        ↪   action that should be taken. Remember, you are exploring the environment and
        ↪   trying to understand the underlying rules. Reflect on your action and self
        ↪   evaluate any potential issues before selecting the action. Output your final
        ↪   choice of action within a <action> tag.\nAdditionally, you can modify the
        ↪   contents of the scratchpad to use as memory since you can only observe the
        ↪   most recent states.\nPlease include the additions to the scratchpad withing
        ↪   <scratchpad_add> tags and deletions withing <scratchpad_del> tags. Output your
        ↪   choice of action within a <action> tag."
    },
    ...,
    {
        "role": "user",
        "content": "The interaction phase is over, you have entered the test phase. You
        ↪   will now be given a goal state and a highlight mask of the same size as the
        ↪   grid where 1 indicates the region to be reached and 0 indicates the region to
        ↪   be ignored. \n            Your aim is to solve a planning task in the
        ↪   environment you interacted by reaching the goal state in the highlighted
        ↪   region.\n            Note that you can no longer reset the environment, so
        ↪   plan carefully. You will be given the same environment as you interacted with
        ↪   in the interaction phase, you need to interact with it to reach the goal state
        ↪   in the highlighted region.\n            Your grid will be checked against the
        ↪   goal state and the highlight mask at every timestep. If you reach the goal
        ↪   state in the highlighted region, you will be given a reward. You may choose to
        ↪   quit at any time if you are stuck.\n            The initial grid is:\n
        ↪   {\"render\": [[\"black\", \"black\", \"black\", \"black\", ...] ...],
        ↪   \"highlight_mask\": [[0, 1, 1, ...], ...]}\n\nThe following actions are
        ↪   available at this step: left,\nright,\nup,\ndown,\nclick [0-15]
        ↪   [0-15],\nnoop,\nquit,\nquit\nThink step by step about the next action that
        ↪   should be taken. Remember, you are exploring the environment and trying to
        ↪   understand the underlying rules. Reflect on your action and self evaluate any
        ↪   potential issues before selecting the action. Output your final choice of
        ↪   action within a <action> tag.\nAdditionally, you can modify the contents of
        ↪   the scratchpad to use as memory since you can only observe the most recent
        ↪   states.\nPlease include the additions to the scratchpad withing
        ↪   <scratchpad_add> tags and deletions withing <scratchpad_del> tags. Output your
        ↪   choice of action within a <action> tag."
    }
]
```

Listing D.4: Prompts for planning task

### E.1 THEORETICAL FRAMEWORK FOR NEXT-STATE PREDICTION MODELS

In this section, we formalize the three AutumnBench task types in terms of a learned next-state prediction model. This framework clarifies how world-model approaches would tackle each task and lays the groundwork for our Autumn-simulator agent baseline (Section E.2.1).

#### E.1.1 NEXT-STATE PREDICTION MODEL

Let a next-state prediction model define $\hat{p}(x_{t+1} \mid x_{1:t}, a_{1:t})$.

**Masked Frame Prediction (MFP).** Select

$$\hat{c} \in \arg \max_{c \in \{1, \ldots, C\}} p\Big(x_{T+1} = x^{(c)} \mid x_1, a_{1:T}, x_{\bar{m}, 2:T+1}\Big),$$

with

$$p\Big(x_{T+1} = x^{(c)} \mid x_1, a_{1:T}, x_{\bar{m}, 2:T+1}\Big) := \int \mathbb{I}\Big[x_{T+1} = x^{(c)}\Big] \, p(x_{2:T+1} \mid x_1, a_{1:T}, x_{\bar{m}, 2:T+1}) \, dx_{2:T+1},$$

and

$$p(x_{2:T+1} \mid x_1, a_{1:T}, x_{\bar{m}, 2:T+1}) \propto \mathbb{I}[x_{2:T+1} \text{ matches } x_{\bar{m}, 2:T+1}] \, \hat{p}(x_{2:T+1} \mid x_1, a_{1:T}).$$

A Monte-Carlo estimator (dropping the normalizer $Z$) is

$$\hat{c} \approx \arg \max_c \sum_{i=1}^{N} \mathbb{I}\Big(x^{(c)}, x_{T+1}^{(i)}\Big) \cdot \mathbb{I}\Big(x_{2:T+1}^{(i)} \text{ matches } x_{\bar{m}, 2:T+1}\Big), \quad x_{2:T+1}^{(i)} \sim \hat{p}(\cdot \mid x_1, a_{1:T}).$$

I.e., we sample $N$ rollouts from $\hat{p}$, keep those consistent with the masks, and pick the candidate that appears most often at $T+1$.

**Change Detection (CD).** Detect the first defect frame when model surprise exceeds an entropy-scaled threshold:

$$t^\star := \min \Big\{ t : \, -\log \hat{p}(x_t \mid x_{1:t-1}, a_{1:t-1}) > \kappa \, \mathbb{H}[\hat{p}(\cdot \mid x_{1:t-1}, a_{1:t-1})] \Big\}.$$

Here, $\kappa > 0$ is tuned on held-out data and $\mathbb{H}[p] = \mathbb{E}_p[-\log p]$.

**Planning (PL).** Given $x_t$, action space $\mathcal{A}$, and a goal specified by a partial observation, use online planning with selective replanning (e.g., MCTS over a learned model):

- Maintain/reuse a search tree rooted at the current $x_t$.

- Replan when uncertainty is high (e.g., surprise, small value margins) or a budget triggers.

- Run $N$ UCT simulations with a goal-based reward; act by highest-visit action; advance the subtree after stepping.

### E.2 NON-LLM BASELINE AGENTS

To establish performance bounds, we evaluate two baseline agents: an oracle simulator with perfect environment access (Section E.2.1) and a purely random agent (Section E.2.2). These baselines help contextualize reasoning model performance by showing what is achievable with perfect world knowledge versus chance-level behavior.

#### E.2.1 AUTUMN-SIMULATOR AGENT

The Autumn-simulator agent has direct access to the ground-truth Autumn program simulator for each environment, making it comparable in spirit to next-frame-prediction models such as Dreamer-v3 (Hafner et al., 2023). Despite this privileged access, the agent must still contend with stochasticity and long planning horizons. The agent observes environments through the same visual interface and task APIs as the reasoning models, ensuring a fair comparison.

**Task-specific solvers.**

- **Masked-frame prediction**: Roll out the simulator along the given test actions to the final timestep; choose the option whose masked region best matches the ground-truth final grid.

- **Change detection**: Simulate the original and test-phase dynamics under the same random action sequence and report the earliest timestep at which the observed grids differ.

- **Planning**: Perform BFS over the action space using the ground-truth simulator until a state satisfying the highlighted goal region is reached; return the first (shortest) solution found.

To mitigate stochasticity, we repeat each procedure 10,000 times and aggregate as above; for planning we cap search depth and breadth at 10.

### E.2.2 RANDOM AGENT BASELINE

To establish a lower performance bound, we evaluate a random agent that selects uniformly among six action types: click, up, down, left, right, and no-op. When the agent selects *click*, it samples a grid position uniformly at random. Table E.6 (presented later for flow reasons) reports the probability that this random policy triggers the defect within 1000 timesteps for change detection, and the probability of reaching the goal within 1000 timesteps for planning. These probabilities quantify task difficulty from a pure exploration perspective: environments with very low random success rates require structured reasoning rather than exploration alone.

### E.3 DETAILED PERFORMANCE ANALYSIS

This subsection presents comprehensive per-environment performance results (Section E.3.1), examines how reasoning models scale with computational resources (Section E.3.3), and assesses robustness to prompt variations (Section E.3.2).

### E.3.1 PER-ENVIRONMENT SCORES

Table E.2 presents average scores for all agents across the three task types (change detection, masked frame prediction, and planning) for each of the 43 AutumnBench environments. These granular results reveal substantial heterogeneity in task difficulty and agent capabilities: some environments (e.g., 7WWW9 , 76Z75 ) are solved reliably across multiple task types, while others (e.g., 6JKKA , K8MTQ ) remain challenging for all agents.

### E.3.2 PROMPT ROBUSTNESS ANALYSIS

Table E.1: Performance of reasoning models under different prompts on a subset of AutumnBench problems

| Model | new_system_prompt | new_system_prompt_refined | prompt_with_tutorial | original |
|---|---|---|---|---|
| claude-4-sonnet | $0.1605 \pm 0.3389$ | $0.0630 \pm 0.2142$ | $0.1418 \pm 0.3442$ | $\mathbf{0.2609} \pm 0.4490$ |
| gemini-2.5-pro | $0.1880 \pm 0.3853$ | $\mathbf{0.1917} \pm 0.3890$ | $0.1667 \pm 0.3807$ | $0.1667 \pm 0.3807$ |
| o3 | $0.1539 \pm 0.3537$ | $0.1764 \pm 0.3834$ | $\mathbf{0.2214} \pm 0.3982$ | $0.1769 \pm 0.3846$ |

To assess the sensitivity of our results to prompt design, we evaluated three alternative prompt formulations alongside the original prompt on a subset of AutumnBench problems. The prompts and complete interaction logs are available in the dataset release (Anonymous, 2025). We tested: (1) new_system_prompt, which provides more explicit task instructions; (2) new_system_prompt_refined, an enhanced version generated using Claude's Prompt Improver tool with example interaction traces[1]; and (3) prompt_with_tutorial, which includes worked examples from the human tutorial.

Table E.1 shows that the optimal prompt varies by model: Gemini-2.5-Pro achieves highest performance with new_system_prompt_refined, Claude-4-Sonnet with the original prompt,

---

[1] https://platform.claude.com/docs/en/build-with-claude/prompt-engineering/prompt-improver

and o3 with `prompt_with_tutorial`. However, performance differences across prompts remain relatively modest (typically $\Delta < 0.1$), suggesting that model capabilities rather than prompt engineering primarily determine performance. The evaluation subset comprises all three task types from eight environments: `S2KT7` , `27VWC` , `KFQYT` , `B8AKZ` , `4CKC2` , `4T8TR` , `QDHS3` , and `NTQ4Y` .

### E.3.3 Performance versus Computational Cost

A central question in AI evaluation is whether increased computational resources translate to better performance. To investigate this, we evaluated two additional models with different cost profiles: Qwen3-235b-a22b-thinking-2507 and Gemini 2.5 Flash. We partition the 43 AutumnBench environments into two categories based on whether average agent performance (computed by ranking agents by their known computational cost and checking for monotonic improvement): *cost-increasing* environments where performance strictly increases with computational budget, and *non-increasing* environments where performance plateaus or decreases despite additional resources.

**Overall findings.** Performance improves with higher computational cost in 25 of 43 environments (58%), but remains flat or decreases in 18 environments (42%). Critically, *no* environment achieves perfect performance (score = 1.0) with the least expensive model, indicating that some minimum capability threshold is necessary. However, the failure of 42% of environments to benefit from additional compute suggests fundamental reasoning limitations that cannot be overcome by scaling resources alone.

**Task-specific patterns.** Masked frame prediction shows improvements in 16 environments (37.2%), including `S2KT7` , `T5F9B` , `N59TE` , `NF5VZ` , `236VK` , `6JVMF` , `27JBD` , `BT2KZ` , `4CKC2` , `N2NTD` , `YS322` , `83WKQ` , `VA6FQ` , `XHGKQ` , `4N7BB` , and `DGG2C` . The remaining 62.7% show no improvement regardless of additional resources.

Change detection exhibits the least benefit from additional computation, with improvements in only 14 environments (32.6%): `NRDF6` , `N59TE` , `NF5VZ` , `236VK` , `27JBD` , `QQM74` , `BT2KZ` , `4CKC2` , `N2NTD` , `EAHCW` , `3J4Z7` , `7VKTD` , `VZ2Q4` , and `JXQAW` .

Planning demonstrates improvements in 16 environments (37.2%): `S2KT7` , `KFQYT` , `B58F3` , `27JBD` , `VQJH6` , `QQM74` , `9F8AJ` , `7WWW9` , `YS322` , `EAHCW` , `83WKQ` , `3J4Z7` , `7VKTD` , `F5W3N` , `JXQAW` , and `4N7BB` .

**Environment characteristics.** Certain environments (`27JBD` , `QQM74` , `7VKTD` , `JXQAW` ) consistently benefit from additional resources across multiple tasks, suggesting they require sophisticated reasoning that scales with compute. Conversely, environments like `27VWC` , `6JKKA` , `K8MTQ` , and `76Z75` show no improvement in any task, indicating either very low or very high baseline difficulty that additional compute cannot address.

### E.4 Behavioral Analysis

Beyond aggregate performance metrics, we analyze how agents *behave* during interaction by examining their action distributions (Section E.4.1) and the relationship between internal perplexity dynamics and task success (Section E.4.2).

### E.4.1 Action-Distribution Analysis

Understanding *how* agents explore environments provides insight into their reasoning strategies. Table E.3 reports the average number of unique actions each agent performs during the exploration phase, broken down by action type. We count spatial actions distinctly: clicking on positions $(2, 3)$ and $(5, 7)$ counts as two unique click actions, while repeatedly pressing the directional key *up* counts as a single unique directional action. This metric accounts for environment-specific variations such as differences in grid sizes and object configurations.

Humans consistently use more unique click actions than reasoning models, particularly in complex environments like `YS322` (75.7 unique clicks vs. 7.0–23.3 for models) and `EAHCW` (55.0 vs. 5.3–15.7). This suggests humans employ more thorough spatial exploration. In contrast, reasoning models tend to use directional actions more uniformly across environments (typically 2–4 unique

Table E.2: Average scores by environment across three task types and agents. Here Claude refers to Claude-4-Sonnet, Gemini refers to Gemini-2.5-Pro, Gemini-Flash refers to Gemini 2.5 Flash, and Qwen refers to Qwen3-235b-a22b-thinking-2507.

| Env | Change Detection | | | | | | Masked Frame Prediction | | | | | | Planning | | | | | |
|---|---|---|---|---|---|---|---|---|---|---|---|---|---|---|---|---|---|---|
| | Claude | Gemini-Pro | Human | o3 | Gemini-Flash | Qwen | Claude | Gemini-Pro | Human | o3 | Gemini-Flash | Qwen | Claude | Gemini-Pro | Human | o3 | Gemini-Flash | Qwen |
| S2KT7 | 0.0 | 0.0 | 0.86 | 0.0 | 0.0 | 0.0 | 1.0 | 1.0 | 1.0 | 1.0 | 0.0 | 1.0 | 1.0 | 0.0 | 1.0 | 1.0 | 1.0 | 1.0 |
| 27VWC | 0.0 | 0.0 | 0.99 | 0.0 | 0.0 | 0.0 | 0.0 | 0.0 | 0.20 | 0.0 | 0.0 | 1.0 | 0.0 | 0.0 | 0.66 | 0.0 | 0.0 | 0.0 |
| KFQYT | 0.0 | 0.0 | 0.93 | 0.0 | 0.0 | 0.0 | 0.0 | 1.0 | 1.0 | 1.0 | 1.0 | 0.0 | 0.0 | 0.0 | 1.0 | 0.0 | 0.0 | 0.0 |
| B58F3 | 0.0 | 0.0 | 0.99 | 0.0 | 0.0 | 0.0 | 1.0 | 0.0 | 1.0 | 1.0 | 1.0 | 0.0 | 1.0 | 0.0 | 1.0 | 1.0 | 0.0 | 0.0 |
| NRDF6 | 1.0 | 0.0 | 0.99 | 0.62 | 0.36 | 0.0 | 1.0 | 0.0 | 1.0 | 0.0 | 0.0 | 0.0 | 0.0 | 1.0 | 1.0 | 0.0 | 1.0 | 0.0 |
| 6JKKA | 0.0 | 0.0 | 1.0 | 0.0 | 0.0 | 0.0 | 0.0 | 0.0 | 0.38 | 0.0 | 0.0 | 0.0 | 0.0 | 0.0 | 1.0 | 0.0 | 0.0 | 0.0 |
| K8MTQ | 0.0 | 0.0 | 1.0 | 0.0 | 0.0 | 0.0 | 0.0 | 0.0 | 0.52 | 0.0 | 0.0 | 0.0 | 0.0 | 0.0 | 1.0 | 0.0 | 0.0 | 0.0 |
| T5F9B | 0.0 | 0.0 | 0.15 | 0.0 | 0.0 | 0.0 | 0.0 | 0.0 | 0.73 | 0.0 | 0.0 | 0.0 | 0.0 | 0.0 | 1.0 | 1.0 | 0.0 | 0.0 |
| N59TE | 0.0 | 0.0 | 1.0 | 0.80 | 0.0 | 0.0 | 0.0 | 0.0 | 0.98 | 0.0 | 0.0 | 0.0 | 1.0 | 1.0 | 1.0 | 1.0 | 0.0 | 0.0 |
| 76Z75 | 0.0 | 0.0 | 0.81 | 0.0 | 0.0 | 0.0 | 1.0 | 1.0 | 1.0 | 1.0 | 1.0 | 1.0 | 1.0 | 0.0 | 1.0 | 0.0 | 1.0 | 1.0 |
| NF5VZ | 0.0 | 1.0 | 0.88 | 0.0 | 0.0 | 0.0 | 0.0 | 1.0 | 0.68 | 0.0 | 1.0 | 0.0 | 0.0 | 0.0 | 0.0 | 0.0 | 0.0 | 0.0 |
| 236VK | 1.0 | 0.0 | 1.0 | 0.46 | 0.0 | 0.0 | 0.0 | 0.0 | 0.0 | 0.0 | 0.0 | 1.0 | 0.0 | 0.0 | 1.0 | 1.0 | 0.0 | 0.0 |
| DQ8GC | 0.0 | 1.0 | 1.0 | 0.0 | 0.95 | 0.95 | 1.0 | 0.0 | 0.0 | 0.0 | 1.0 | 0.0 | 0.0 | 0.0 | 1.0 | 0.0 | 0.0 | 0.0 |
| QM9XB | 0.0 | 0.0 | 1.0 | 0.0 | 0.0 | 0.0 | 0.0 | 0.0 | 0.98 | 0.0 | 0.0 | 0.0 | 0.0 | 0.0 | 1.0 | 0.0 | 0.0 | 0.0 |
| 6JVMF | 0.0 | 0.0 | 0.98 | 0.0 | 0.0 | 0.0 | 0.0 | 0.0 | 1.0 | 0.0 | 0.0 | 0.0 | 1.0 | 0.0 | 1.0 | 0.0 | 0.0 | 0.0 |
| 27JBD | 0.0 | 1.0 | 1.0 | 0.23 | 0.0 | 0.96 | 1.0 | 1.0 | 1.0 | 1.0 | 0.0 | 0.0 | 0.0 | 0.0 | 0.33 | 1.0 | 0.0 | 1.0 |
| VQJH6 | 0.0 | 0.0 | 1.0 | 0.0 | 0.0 | 0.0 | 1.0 | 0.0 | 1.0 | 0.0 | 0.0 | 0.0 | 1.0 | 0.0 | 1.0 | 0.0 | 0.0 | 0.0 |
| QQM74 | 0.0 | 0.0 | 0.99 | 0.99 | 0.0 | 0.0 | 1.0 | 0.0 | 1.0 | 1.0 | 0.0 | 0.0 | 1.0 | 0.0 | 0.66 | 0.0 | 0.0 | 1.0 |
| 7XF97 | 0.0 | 0.0 | 0.64 | 0.0 | 0.0 | 0.0 | 0.0 | 0.0 | 0.60 | 0.0 | 0.0 | 0.0 | 0.0 | 0.0 | 0.66 | 0.0 | 0.0 | 0.0 |
| BT3GB | 0.0 | 0.0 | 1.0 | 0.0 | 0.98 | 0.0 | 0.0 | 0.0 | 0.98 | 0.0 | 0.0 | 0.0 | 0.0 | 0.0 | 1.0 | 0.0 | 0.0 | 0.0 |
| BT2KZ | 1.0 | 0.0 | 1.0 | 0.62 | 0.0 | 0.0 | 0.0 | 0.0 | 1.0 | 0.0 | 0.0 | 1.0 | 1.0 | 1.0 | 1.0 | 1.0 | 0.0 | 0.0 |
| B8AKZ | 0.0 | 0.0 | 1.0 | 0.99 | 1.0 | 0.0 | 0.0 | 0.0 | 0.90 | 0.0 | 0.0 | 0.0 | 0.0 | 0.0 | 1.0 | 0.0 | 0.0 | 0.0 |
| 9F8AJ | 0.0 | 0.0 | 1.0 | 0.0 | 0.0 | 0.0 | 1.0 | 0.0 | 0.55 | 0.0 | 0.0 | 0.0 | 0.0 | 0.0 | 1.0 | 0.0 | 0.0 | 0.0 |
| 4CKC2 | 1.0 | 0.0 | 1.0 | 0.0 | 0.0 | 0.0 | 1.0 | 1.0 | 0.57 | 0.0 | 0.0 | 0.0 | 1.0 | 1.0 | 1.0 | 0.0 | 1.0 | 1.0 |
| 7WWW9 | 0.0 | 0.0 | 1.0 | 0.0 | 0.0 | 1.0 | 1.0 | 1.0 | 1.0 | 1.0 | 1.0 | 0.0 | 1.0 | 1.0 | 1.0 | 0.0 | 1.0 | 1.0 |
| N2NTD | 0.0 | 1.0 | 0.99 | 0.0 | 0.0 | 0.0 | 0.0 | 0.0 | 0.60 | 0.0 | 0.0 | 0.0 | 0.0 | 0.0 | 1.0 | 1.0 | 0.0 | 0.0 |
| BY2Q7 | 0.0 | 0.0 | 0.74 | 0.0 | 0.0 | 0.0 | 0.0 | 0.0 | 0.69 | 0.0 | 0.0 | 0.0 | 0.0 | 0.0 | 1.0 | 0.0 | 0.0 | 0.0 |
| 4T8TR | 1.0 | 1.0 | 1.0 | 0.89 | 0.98 | 0.44 | 0.0 | 0.0 | 1.0 | 0.0 | 0.0 | 0.0 | 0.0 | 1.0 | 1.0 | 0.0 | 1.0 | 1.0 |
| YS322 | 0.0 | 0.0 | 1.0 | 0.0 | 0.0 | 0.0 | 0.0 | 1.0 | 0.94 | 0.0 | 0.0 | 0.0 | 1.0 | 0.0 | 1.0 | 0.0 | 0.0 | 0.0 |
| WHGHP | 0.0 | 0.0 | 0.97 | 0.0 | 0.0 | 0.0 | 0.0 | 0.0 | 0.91 | 0.0 | 0.0 | 0.0 | 0.0 | 0.0 | 1.0 | 0.0 | 0.0 | 0.0 |
| EAHCW | 0.0 | 0.0 | 1.0 | 0.33 | 0.0 | 0.0 | 0.0 | 0.0 | 0.84 | 1.0 | 0.0 | 0.0 | 0.0 | 0.0 | 0.66 | 0.0 | 0.0 | 0.0 |
| 83WKQ | 0.0 | 0.0 | 0.99 | 0.0 | 0.89 | 0.0 | 1.0 | 0.0 | 1.0 | 1.0 | 0.0 | 0.0 | 1.0 | 0.0 | 1.0 | 0.0 | 0.0 | 1.0 |
| 3J4Z7 | 0.0 | 0.0 | 0.93 | 0.96 | 0.0 | 0.0 | 0.0 | 0.0 | 1.0 | 1.0 | 0.0 | 0.0 | 0.0 | 0.0 | 1.0 | 0.0 | 0.0 | 0.0 |
| QDHS3 | 0.0 | 0.0 | 1.0 | 0.0 | 0.0 | 0.0 | 0.0 | 0.0 | 0.93 | 0.0 | 0.0 | 0.0 | 0.0 | 0.0 | 1.0 | 0.0 | 0.0 | 0.0 |
| VA6FQ | 0.0 | 0.0 | 1.0 | 0.0 | 0.0 | 0.0 | 0.0 | 0.0 | 0.33 | 0.0 | 0.0 | 0.0 | 0.0 | 1.0 | 1.0 | 0.0 | 0.0 | 0.0 |
| 7VKTD | 0.0 | 0.0 | 1.0 | 0.50 | 0.0 | 0.0 | 1.0 | 1.0 | 0.95 | 1.0 | 0.0 | 0.0 | 0.0 | 1.0 | 1.0 | 0.0 | 0.0 | 1.0 |
| VZ2Q4 | 0.0 | 0.0 | 1.0 | 0.35 | 0.0 | 0.0 | 1.0 | 1.0 | 1.0 | 0.0 | 1.0 | 0.0 | 1.0 | 0.0 | 1.0 | 0.0 | 0.0 | 0.0 |
| XHGKQ | 0.0 | 0.0 | 1.0 | 0.0 | 0.0 | 0.0 | 0.0 | 0.0 | 0.85 | 0.0 | 1.0 | 1.0 | 1.0 | 1.0 | 1.0 | 1.0 | 0.0 | 0.0 |
| F5W3N | 0.0 | 0.0 | 0.78 | 0.0 | 0.0 | 0.0 | 1.0 | 0.0 | 0.19 | 0.0 | 1.0 | 0.0 | 1.0 | 0.0 | 1.0 | 0.0 | 0.0 | 0.0 |
| JXQAW | 0.0 | 0.0 | 1.0 | 0.96 | 0.63 | 0.97 | 1.0 | 0.0 | 1.0 | 1.0 | 1.0 | 0.0 | 1.0 | 0.0 | 1.0 | 0.0 | 0.0 | 0.0 |
| 4N7BB | 0.0 | 0.0 | 0.94 | 0.76 | 0.95 | 0.0 | 1.0 | 1.0 | 1.0 | 1.0 | 1.0 | 1.0 | 1.0 | 0.0 | 1.0 | 1.0 | 0.0 | 0.0 |
| NTQ4Y | 0.0 | 0.0 | 0.31 | 0.0 | 0.0 | 0.0 | 0.0 | 0.0 | 0.98 | 0.0 | 0.0 | 0.0 | 0.0 | 0.0 | 1.0 | 0.0 | 0.0 | 0.0 |
| DGG2C | 0.0 | 0.0 | 0.99 | 0.0 | 0.99 | 0.0 | 0.0 | 0.0 | 1.0 | 0.0 | 0.0 | 1.0 | 1.0 | 0.0 | 1.0 | 1.0 | 0.0 | 0.0 |

Table E.3: Unique clicks and directional actions by environment and agent. Here Claude refers to Claude-4-Sonnet, Gemini-Pro refers to Gemini-2.5-Pro, Gemini-Flash refers to Gemini 2.5 Flash, and Qwen refers to Qwen3-235b-a22b-thinking-2507.

| Env | Average Number of unique clicks | | | | | | Average Number of unique directional actions | | | | | |
| | Claude | Gemini-Pro | o3 | Human | Gemini-Flash | Qwen | Claude | Gemini-Pro | o3 | Human | Gemini-Flash | Qwen |
|---|---|---|---|---|---|---|---|---|---|---|---|---|
| S2KT7 | 6.0 | 7.0 | 6.0 | 8.3 | 8.3 | 4.7 | 4.0 | 4.0 | 3.7 | 2.7 | 2.7 | 3.3 |
| 27VWC | 9.7 | 6.7 | 11.0 | 9.7 | 9.0 | 6.0 | 4.0 | 4.0 | 3.7 | 1.3 | 4.0 | 3.5 |
| KFQYT | 4.3 | 13.0 | 5.3 | 13.3 | 9.3 | 4.7 | 3.3 | 4.0 | 3.7 | 4.0 | 4.0 | 3.3 |
| B58F3 | 2.7 | 1.0 | 2.3 | 1.0 | 5.7 | 2.3 | 4.0 | 4.0 | 3.7 | 1.3 | 3.7 | 4.0 |
| NRDF6 | 14.3 | 13.3 | 8.3 | 19.3 | 8.3 | 7.7 | 4.0 | 3.3 | 1.7 | 3.7 | 3.7 | 3.0 |
| 6JKKA | 9.0 | 9.0 | 5.0 | 2.7 | 6.7 | 2.0 | 4.0 | 4.0 | 4.0 | 2.7 | 3.0 | 3.3 |
| K8MTQ | 10.0 | 6.3 | 7.3 | 26.3 | 6.7 | 8.0 | 4.0 | 3.7 | 3.0 | 3.3 | 4.0 | 4.0 |
| T5F9B | 8.7 | 10.0 | 11.3 | 19.0 | 4.7 | 6.7 | 3.7 | 2.7 | 2.3 | 2.0 | 4.0 | 3.0 |
| N59TE | 9.3 | 26.7 | 9.7 | 15.3 | 7.3 | 9.7 | 3.3 | 4.0 | 3.3 | 3.3 | 2.7 | 2.3 |
| 76Z75 | 4.7 | 5.7 | 4.7 | 17.7 | 5.0 | 3.3 | 4.0 | 4.0 | 4.0 | 4.0 | 4.0 | 3.3 |
| NF5VZ | 8.3 | 10.7 | 7.0 | 4.0 | 4.3 | 1.5 | 4.0 | 4.0 | 3.7 | 2.3 | 4.0 | 4.0 |
| 236VK | 4.7 | 4.3 | 2.7 | 6.7 | 8.0 | 3.3 | 4.0 | 4.0 | 4.0 | 3.7 | 2.3 | 4.0 |
| DQ8GC | 1.3 | 2.0 | 0.7 | 19.7 | 2.7 | 1.3 | 3.7 | 3.7 | 3.7 | 4.0 | 4.0 | 4.0 |
| QM9XB | 5.3 | 3.3 | 8.7 | 9.0 | 3.0 | 2.7 | 4.0 | 4.0 | 3.7 | 2.7 | 3.0 | 3.7 |
| 6JVMF | 4.0 | 5.7 | 2.0 | 3.7 | 1.0 | 1.5 | 4.0 | 4.0 | 4.0 | 3.7 | 4.0 | 4.0 |
| 27JBD | 5.7 | 7.0 | 7.3 | 18.7 | 10.7 | 3.3 | 3.0 | 1.7 | 1.7 | 2.7 | 1.7 | 2.7 |
| VQJH6 | 17.7 | 7.7 | 8.7 | 20.3 | 7.3 | 4.5 | 4.0 | 3.7 | 3.7 | 4.0 | 4.0 | 4.0 |
| QQM74 | 4.3 | 6.7 | 5.7 | 18.7 | 4.0 | 5.0 | 4.0 | 3.3 | 3.0 | 3.7 | 4.0 | 3.0 |
| 7XF97 | 18.7 | 18.3 | 9.0 | 22.7 | 7.7 | 7.7 | 4.0 | 4.0 | 4.0 | 4.0 | 4.0 | 2.3 |
| BT3GB | 3.7 | 1.0 | 4.0 | 6.0 | 4.3 | 0.5 | 4.0 | 4.0 | 4.0 | 3.7 | 4.0 | 3.5 |
| BT2KZ | 4.0 | 6.0 | 6.3 | 18.7 | 6.3 | 7.7 | 2.7 | 3.3 | 1.0 | 2.7 | 0.3 | 2.7 |
| B8AKZ | 3.7 | 8.7 | 4.0 | 22.0 | 3.7 | 6.0 | 4.0 | 4.0 | 4.0 | 4.0 | 4.0 | 4.0 |
| 9F8AJ | 9.7 | 20.3 | 4.3 | 7.7 | 10.3 | 6.7 | 2.7 | 4.0 | 3.0 | 2.7 | 4.0 | 2.0 |
| 4CKC2 | 9.0 | 5.7 | 7.0 | 13.0 | 5.0 | 7.0 | 2.7 | 2.7 | 2.3 | 2.7 | 2.7 | 2.3 |
| 7WWW9 | 3.0 | 5.0 | 3.7 | 10.3 | 2.7 | 2.0 | 4.0 | 4.0 | 4.0 | 3.0 | 4.0 | 4.0 |
| N2NTD | 8.3 | 6.3 | 4.3 | 12.0 | 6.3 | 9.0 | 4.0 | 4.0 | 4.0 | 2.7 | 4.0 | 3.7 |
| BY2Q7 | 3.3 | 11.0 | 11.3 | 18.7 | 9.3 | 9.0 | 2.7 | 4.0 | 2.7 | 4.0 | 3.7 | 3.0 |
| 4T8TR | 2.3 | 3.3 | 5.7 | 18.3 | 8.7 | 4.7 | 3.7 | 3.7 | 4.0 | 0.3 | 2.0 | 4.0 |
| YS322 | 13.7 | 23.3 | 9.7 | 75.7 | 12.7 | 7.0 | 3.3 | 4.0 | 2.3 | 4.0 | 3.3 | 1.5 |
| WHGHP | 2.3 | 6.7 | 2.0 | 7.0 | 3.3 | 3.3 | 4.0 | 4.0 | 4.0 | 4.0 | 3.7 | 4.0 |
| EAHCW | 13.3 | 9.3 | 15.7 | 55.0 | 5.3 | 8.7 | 3.3 | 4.0 | 4.0 | 4.0 | 2.7 | 2.7 |
| 83WKQ | 2.7 | 4.0 | 7.7 | 14.3 | 6.7 | 3.0 | 4.0 | 4.0 | 3.7 | 2.7 | 3.0 | 3.7 |
| 3J4Z7 | 16.0 | 4.3 | 9.7 | 6.0 | 4.3 | 3.5 | 4.0 | 4.0 | 3.0 | 1.3 | 2.0 | 4.0 |
| QDHS3 | 5.0 | 0.7 | 1.0 | 15.3 | 3.5 | 4.0 | 4.0 | 3.7 | 4.0 | 4.0 | 4.0 | 4.0 |
| VA6FQ | 14.3 | 3.0 | 13.7 | 33.3 | 3.7 | 7.3 | 4.0 | 4.0 | 3.7 | 4.0 | 4.0 | 2.7 |
| 7VKTD | 5.0 | 2.3 | 2.0 | 6.7 | 2.3 | 3.3 | 4.0 | 4.0 | 3.7 | 2.7 | 4.0 | 2.3 |
| VZ2Q4 | 3.3 | 3.3 | 9.0 | 6.0 | 2.7 | 5.0 | 4.0 | 3.0 | 2.3 | 1.3 | 2.7 | 2.7 |
| XHGKQ | 8.0 | 7.7 | 11.3 | 1.0 | 5.3 | 2.0 | 4.0 | 4.0 | 3.7 | 2.7 | 4.0 | 3.0 |
| F5W3N | 5.3 | 5.0 | 1.0 | 3.3 | 3.0 | 11.7 | 4.0 | 4.0 | 3.3 | 2.3 | 3.0 | 3.0 |
| JXQAW | 9.7 | 8.7 | 10.3 | 18.7 | 8.0 | 8.0 | 4.0 | 4.0 | 2.0 | 4.0 | 2.7 | 3.3 |
| 4N7BB | 8.0 | 6.0 | 6.3 | 9.0 | 5.7 | 5.5 | 4.0 | 4.0 | 2.3 | 2.7 | 3.0 | 1.0 |
| NTQ4Y | 9.3 | 8.0 | 10.3 | 42.0 | 7.0 | 6.3 | 3.7 | 2.7 | 2.0 | 2.7 | 3.3 | 2.3 |
| DGG2C | 2.3 | 3.7 | 9.7 | 4.0 | 2.0 | 4.0 | 4.0 | 4.0 | 2.3 | 3.7 | 3.7 | 3.0 |

directional actions), possibly reflecting a more systematic but less adaptive exploration strategy. The similarity in directional action counts across agents suggests these actions are easier to reason about or require less environment-specific adaptation.

### E.4.2 PERPLEXITY-PERFORMANCE CORRELATIONS

Normalized perplexity (Definitions D.1 and D.2 in the main paper) quantifies how an agent's uncertainty evolves during interaction. Here we examine whether perplexity dynamics correlate with task success, which would validate perplexity as a meaningful intrinsic signal of learning progress.

**Environment-level correlations.** Aggregating across all env–agent pairs (with scores averaged across the three task types), we find statistically significant but modest correlations between perplexity metrics and performance (Table E.4).

Table E.4: Env–agent correlations between perplexity metrics and average score across task types.

| Metric vs Score | Pearson r | p-value | n | Spearman $\rho$ | p-value |
|---|---|---|---|---|---|
| Final perplexity | 0.324 | 1.06e-07 | 258 | 0.264 | 1.75e-05 |
| AUC (normalized perplexity) | -0.214 | 5.37e-04 | 258 | -0.214 | 5.41e-04 |

Higher task scores are associated with slightly higher final perplexity (positive Pearson $r = 0.324$), suggesting that successful agents may maintain broader or more diverse hypotheses about environment dynamics. Conversely, higher scores correlate with lower AUC (negative $r = -0.214$), indicating that successful agents reduce perplexity more quickly or efficiently during interaction. Both effects are statistically significant but small in magnitude, suggesting perplexity provides a weak but meaningful signal of learning progress.

**Per-agent correlations across environments.** Analyzing each agent separately across the 43 environments (Table E.5) yields mostly small and statistically non-significant correlations ($|r| < 0.24$, $p > 0.13$ in all cases). This suggests that perplexity dynamics vary substantially across environments and do not consistently predict within-agent performance differences. The near-zero correlation for humans ($r = 0.000$ for final perplexity) likely reflects ceiling effects: humans achieve high scores across nearly all environments, limiting variance in the dependent variable.

Table E.5: Per-agent correlations between perplexity metrics and average score across environments. Entries show correlation $r$ with two-sided $p$ in parentheses; $n = 43$ environments for each agent.

| Agent | Final $r$ (p) | AUC $r$ (p) | n |
|---|---|---|---|
| claude-4-sonnet | -0.231 (0.136) | 0.035 (0.822) | 43 |
| gemini-2.5-flash | -0.074 (0.639) | 0.234 (0.131) | 43 |
| gemini-2.5-pro | -0.124 (0.427) | -0.055 (0.725) | 43 |
| human | -0.000 (1.000) | 0.120 (0.443) | 43 |
| o3 | 0.147 (0.348) | -0.221 (0.154) | 43 |
| qwen3-235b-a22b-thinking-2507 | -0.037 (0.816) | 0.017 (0.911) | 43 |

**Summary.** The weak correlations suggest that while perplexity provides some signal about learning dynamics, it does not strongly predict task success. Other factors—such as hypothesis quality, planning depth, or handling of stochasticity—likely play more important roles in determining final performance.

### E.4.3 RANDOM AGENT PERFORMANCE

Table E.6 reports the performance of the random baseline agent introduced in Section E.2.2. For change detection, we report the probability of triggering the defect within 1000 timesteps; for planning, the probability of reaching the goal within 1000 timesteps. Many environments have very high trigger/goal probabilities ($\geq 0.85$), indicating they permit success through random exploration. Conversely, environments with very low probabilities (e.g., `K8MTQ`, `DQ8GC`, `27JBD`,

and `BT3GB` all have 0.00 planning success) require structured reasoning and cannot be solved by random exploration alone. This heterogeneity in random agent performance helps characterize the fundamental difficulty of each environment.

Table E.6: Random-agent baseline by environment. For CD and PL, we evaluate a random agent that selects uniformly among six action types: click, up, down, left, right, and no-op; when it selects *click*, it samples a grid position uniformly. The table reports the probability of triggering the defect in 1000 timesteps for CD, and the probability of reaching the goal in 1000 timesteps for PL.

| Env | Prob of triggering change | Prob of reaching goal |
|---|---|---|
| VZ2Q4 | 1.00 | 0.02 |
| S2KT7 | 1.00 | 1.00 |
| 27VWC | 0.94 | 0.16 |
| KFQYT | 1.00 | 1.00 |
| B58F3 | 0.87 | 1.00 |
| NRDF6 | 1.00 | 0.71 |
| 6JKKA | 0.98 | 0.20 |
| K8MTQ | 1.00 | 0.00 |
| T5F9B | 0.64 | 0.75 |
| N59TE | 1.00 | 0.00 |
| 76Z75 | 1.00 | 0.04 |
| NF5VZ | 1.00 | 1.00 |
| 236VK | 1.00 | 0.20 |
| DQ8GC | 1.00 | 0.00 |
| QM9XB | 0.43 | 0.10 |
| 6JVMF | 1.00 | 1.00 |
| 27JBD | 1.00 | 0.00 |
| QQM74 | 1.00 | 1.00 |
| VQJH6 | 1.00 | 0.94 |
| 7XF97 | 0.77 | 0.20 |
| BT3GB | 1.00 | 0.00 |
| BT2KZ | 1.00 | 0.00 |
| B8AKZ | 1.00 | 1.00 |
| 9F8AJ | 0.17 | 0.12 |
| 4CKC2 | 0.72 | 0.55 |
| 7WWW9 | 0.66 | 0.53 |
| N2NTD | 0.85 | 0.00 |
| BY2Q7 | 0.00 | 0.00 |
| 4T8TR | 1.00 | 0.37 |
| YS322 | 0.26 | 0.00 |
| WHGHP | 0.00 | 0.22 |
| EAHCW | 1.00 | 0.00 |
| 83WKQ | 1.00 | 1.00 |
| 3J4Z7 | 0.38 | 0.96 |
| QDHS3 | 0.36 | 0.00 |
| VA6FQ | 0.77 | 0.06 |
| 7VKTD | 1.00 | 0.55 |
| VZ2Q4 | 1.00 | 0.02 |
| XHGKQ | 0.13 | 0.00 |
| F5W3N | 0.89 | 0.94 |
| JXQAW | 1.00 | 0.55 |
| 4N7BB | 1.00 | 0.04 |
| NTQ4Y | 0.53 | 0.00 |
| DGG2C | 1.00 | 1.00 |

### E.5 QUALITATIVE EXAMPLES

To provide insight into agent reasoning processes, we present an example scratchpad trace from Claude-4-Sonnet on the `6JVMF` masked frame prediction task. Reasoning models maintain scratchpads during both exploration and testing phases (see prompt specifications in Listings D.2 to D.4). These scratchpads reveal how models form and refine hypotheses about environment dynamics through interaction.

Listing E.1 shows Claude's complete scratchpad for an `6JVMF` episode. The trace illustrates several characteristic behaviors: (1) initial exploration with directional actions to understand basic mechanics; (2) boundary testing to identify edge cases; (3) hypothesis formation and revision as new observations arrive; (4) detailed frame-by-frame analysis during the test phase; and (5) final reasoning to select among the multiple-choice options provided in Listing E.2. The model demonstrates systematic exploration and explicit reasoning about environment state, though it sometimes exhibits repetitive behavior (e.g., boundary oscillation testing) that does not yield additional information.

Listing E.1: Example scratchpad for the `6JVMF` environment, with choices listed in Listing E.2.

```
Initial observations:
- 16x16 grid, mostly black cells
- One blue cell at position (7, 7)
- Available actions: left, right, up, down, click [0-15] [0-15], noop, quit, go-to-test, reset
- Goal: Understand the underlying rules of the environment

Strategy:
1. Try different actions to see how they affect the grid
2. Start with arrow keys to see if they move something
3. Try clicking on different cells to see what happens
4. Look for patterns in how the environment responds
Observations after "right" action:
- Blue cell moved from (7, 7) to (6, 7) - moved LEFT when I pressed RIGHT
- This suggests arrow keys control the blue cell but in reverse direction
- Need to test other arrow keys to confirm this patternConfirmed pattern:
- LEFT key moves blue cell RIGHT
- RIGHT key moves blue cell LEFT
- Arrow keys control blue cell movement in reverse/opposite direction
- Blue cell moves horizontally along row 7 so far

Next tests needed:
- Test UP and DOWN keys to see vertical movement behavior
- Test boundaries - what happens when blue cell reaches edge?
- Test clicking on cells
Updated observations:
- UP key also moves blue cell LEFT (same as RIGHT key)
- Both UP and RIGHT move the blue cell left
- Need to test DOWN key to see if it follows the same pattern as LEFT
Complete movement pattern discovered:
- LEFT key moves blue cell RIGHT
- RIGHT key moves blue cell LEFT
- UP key moves blue cell LEFT
- DOWN key moves blue cell LEFT

So the controls are:
- Only LEFT moves the blue cell right
- RIGHT, UP, and DOWN all move the blue cell left
- Movement is only horizontal along row 7
- No vertical movement detected yet
Boundary behavior discovered:
- When blue cell reaches left edge (0, 7) and tries to move further left, it bounces back to
↪  (1, 7)
- Movement appears to be bounded by grid edges
- Need to test right boundary behavior as well
Additional boundary observations:
- When blue cell is at left edge (0, 7), pressing "left" moves it right to (1, 7)
- Movement controls work normally once away from the boundary
- Continuing to test right boundary behaviorClick behavior observations:
- Clicking on the blue cell at (0, 7) had no visible effect
- Clicking on adjacent cell (1, 7) had no visible effect
- Clicks don't seem to change the grid state or create new objects- Clicking on distant cell
↪  (8, 8) had no visible effect
- Clicks appear to have no effect on the environment at all
- Need to explore if vertical movement is possible, especially at boundaries
Boundary behavior at left edge:
- When at (0, 7), UP key moves blue cell right to (1, 7)
- This confirms UP still tries to move left but bounces at boundary
- Need to test DOWN key for potential vertical movement
```

```
1620
1621    DOWN key behavior confirmed:
        - DOWN moves blue cell left (from (1,7) to (0,7))
1622    - Still no vertical movement discovered
        - Movement remains only horizontal along row 7
1623    Boundary behavior observations:
        - Blue cell seems to have inconsistent movement near left boundary
1624    - Sometimes LEFT moves it right, sometimes it stays at (0,7)
        - Need to test consistent movement patternsLeft boundary oscillation observed:
1625    - Blue cell seems to oscillate between (0,7) and (1,7) when using LEFT key repeatedly
        - May need different strategy to move away from left edge consistentlyBoundary oscillation
1626
        ↪   pattern confirmed:
1627    - At left boundary, blue cell oscillates between (0,7) and (1,7)
        - All arrow keys (LEFT, RIGHT, UP, DOWN) seem to cause same oscillation behavior
1628    - Need to try different strategy to break out of this patternClick behavior confirmed:
        - Clicking on (8, 7) had no effect
1629    - Clicks don't create new objects or change existing ones
        - Blue cell remains trapped at left boundary
1630
1631    Next to try:
        - "step" action to see if there's automatic progression
1632    - "go-to-test" to see if there's a test phase
        - Different exploration strategiesStep action tested:
1633    - "step" action had no visible effect
        - No automatic progression or animation detected
1634    - Grid state remains unchanged
1635
        Summary of findings so far:
1636    - Blue cell oscillates between (0,7) and (1,7) at left boundary
        - All arrow keys cause same oscillation at boundary
1637    - Clicks have no effect
        - Step action has no effect
1638    - Movement is only horizontal along row 7Test phase entered:
        - "go-to-test" action changed available actions to only "noop"
1639    - Grid state appears unchanged with blue cell at (0,7)
        - This suggests a test/observation phase where interaction is limited
1640    - May reveal automatic behavior or require demonstration of understandingTest phase trajectory
        ↪   observation:
1641    Frame 0: Blue cell at (7, 7), action: "initial"
        - This is the starting position, same as when I first entered the environment
1642    - Need to observe how movement progresses through the trajectory
        - Will track the blue cell's position and movement pattern
1643    Frame 1: Blue cell at (7, 8), action: "noop"
        - Blue cell moved DOWN from (7, 7) to (7, 8)
1644    - Action was "noop" – no user input
        - This reveals automatic downward movement/gravity!
1645    - Different from exploration phase where I only saw horizontal movement
        Frame 2: Blue cell at (7, 7), action: "noop"
1646    - Blue cell moved UP from (7, 8) back to (7, 7)
        - Action was "noop" again
1647    - Pattern emerging: vertical oscillation between (7, 7) and (7, 8)
        - Similar to boundary oscillation I saw during exploration
1648    - Not consistent gravity, but oscillating movement with noop
        Frame 3: Blue cell at (6, 7), action: "noop"
1649    - Blue cell moved UP from (7, 7) to (6, 7)
        - Continuing upward movement pattern
1650    - Revised pattern: Initial down, then consistent upward movement with noop
        - Need to see if this continues or if there are boundaries
1651    Frame 4: Blue cell at (7, 7), action: "noop"
        - Blue cell moved DOWN from (6, 7) back to (7, 7)
1652    - Back to original position
        - Pattern seems to be oscillating around (7, 7) position
1653    - Movement sequence so far: center → down → center → up → center
        - Need to continue observing to see if pattern repeats
1654    Frame 5: Blue cell at (6, 7), action: "noop"
        - Blue cell moved UP from (7, 7) to (6, 7)
1655    - Same position as frame 3
        - Pattern emerging: (7,7) → (7,8) → (7,7) → (6,7) → (7,7) → (6,7)
1656    - Seems to be cycling through positions around the center
        - Need to continue to confirm full cycle pattern
1657    Frame 6: Blue cell at (7, 7), action: "noop"
        - Blue cell moved DOWN from (6, 7) to (7, 7)
1658    - Back at center position again
        - Clear oscillation pattern emerging around (7, 7)
1659    - Positions cycle: (7,7) → (7,8) → (7,7) → (6,7) → (7,7) → (6,7) → (7,7)
        - Seems to alternate between moving to adjacent cells and returning to center
1660    Frame 7: Blue cell at (6, 7), action: "noop"
        - Blue cell moved UP from (7, 7) to (6, 7)
1661    - Continuing the established oscillation pattern
        - Pattern confirmed: alternates between (7,7) center and adjacent positions
1662    - Movement sequence: down to (7,8), back to center, up to (6,7), back to center, repeat
        Frame 8: Blue cell at (5, 7), action: "noop"
1663
1664
1665
1666
1667
1668
1669
1670
1671
1672
1673
```

```
1674
1675    - Blue cell moved UP from (6, 7) to (5, 7)
        - New position - breaking the previous oscillation pattern around (7,7)
1676    - Movement is continuing upward instead of returning to center
        - Pattern might be evolving or following a longer trajectory
1677    - Need to continue observing to understand the full movement pattern
1678    Frame 9: Blue cell at (4, 7), action: "noop"
        - Blue cell moved UP from (5, 7) to (4, 7)
1679    - Continuing consistent upward movement
        - Pattern has shifted from oscillation to linear upward movement
1680    - Movement is along column 7, going from row 7 toward row 0
1681    - Need to see if it reaches the boundary or changes direction
1682    Frame 10: Large masked region appeared
        - Masked area covers rows 2-9, columns 4-11
1683    - Blue cell was last seen at (4, 7) moving upward
1684    - Based on movement pattern, blue cell'll should be at (3, 7) which is within masked region
        - This is likely the test region I'll need to predict
1685    - Need to continue stepping through remaining frames before making prediction
1686    Frame 11: Action "click 7 4" performed
        - Click action on position (7, 4) which is within masked region
1687    - Grid appearance unchanged (still masked)
1688    - This suggests trajectory includes click actions on masked positions
        - Need to track all click actions to understand final prediction
1689    - Position (7, 4) is where blue cell should be based on upward movement pattern
1690    Frame 12: Action "noop"
        - Grid appearance unchanged (still masked)
1691    - Following the click action from previous frame
1692    - Need to continue observing to see full sequence of actions
        Frame 13: Action "noop"
1693    - Grid appearance unchanged (still masked)
1694    - Continuing sequence after the click action
        - Need to continue to see if there are more actions or if we reach the final prediction frame
1695    Frame 14: Action "noop"
1696    - Grid appearance unchanged (still masked)
        - Continuing sequence of noop actions after the click
1697    - Need to continue to see if we reach final frame or more actions occur
1698    Frame 15: Action "click 8 7" performed
        - Second click action within masked region at position (8, 7)
1699    - Grid appearance unchanged (still masked)
1700    - Two click actions so far: (7, 4) and (8, 7)
        - Need to continue to see if there are more actions or if we reach final prediction frame
1701    Frame 16: Action "noop"
1702    - Grid appearance unchanged (still masked)
        - Following the second click action from previous frame
1703    - Two click positions recorded: (7, 4) and (8, 7)
1704    - Need to continue to see if we reach final frame
        Frame 17: Action "noop"
1705    - Grid appearance unchanged (still masked)
1706    - Continuing sequence after both click actions
        - Two click positions recorded: (7, 4) and (8, 7)
1707    - Need to continue to see if we reach final prediction frame
1708    Frame 18: Action "noop"
        - Grid appearance unchanged (still masked)
1709    - Continuing sequence of noop actions
1710    - Two click positions remain: (7, 4) and (8, 7)
        - Need to continue stepping to find final frame with options
1711    Frame 19: Action "click 8 7" performed again
1712    - Third click action within masked region, second time clicking (8, 7)
        - Grid appearance unchanged (still masked)
1713    - Click count: (7, 4) clicked once, (8, 7) clicked twice
1714    - Need to continue to see if there are more actions or final frame
        Frame 20: Action "noop"
1715    - Grid appearance unchanged (still masked)
1716    - Following the repeated click on (8, 7)
        - Click count remains: (7, 4) clicked once, (8, 7) clicked twice
1717    - Need to continue to see if we reach final frame with prediction options
1718    Frame 21: Action "noop"
1719    - Grid appearance unchanged (still masked)
        - Continuing sequence of noop actions after second click on (8, 7)
1720    - Click positions tracked: (7, 4) clicked once, (8, 7) clicked twice
        - Need to continue stepping to reach final prediction frame
1721    Frame 22: Blue cell appears at (8, 1)!
1722    - First visible change since masked region appeared
        - Blue cell is outside masked region at position (8, 1)
1723    - Shares x-coordinate with clicked position (8, 7)
1724    - This suggests click actions in masked region have effects outside it
        - Need to continue to see if blue cell moves or if more changes occur
1725    Frame 23: Second blue cell appears at (9, 1)!
1726    - Now two blue cells: (7, 1) and (9, 1)
        - Both at same row (y=1), different columns
1727    - X-coordinates 7 and 9 relate to clicked positions (7, 4) and (8, 7)
        - Pattern emerging: clicks in masked region create blue cells outside at row 1
```

```
- Possible transformation: x-coord preserved/shifted, y-coord becomes 1
Frame 24: Blue cell configuration changed dramatically
- Only one blue cell visible at (7, 0)
- Previous frame had two blue cells at (7, 1) and (9, 1)
- Blue cell at (7, 1) moved up to (7, 0)
- Blue cell at (9, 1) disappeared
- Pattern suggests upward movement and possible disappearance conditions
Frame 25: Blue cell moved to (8, 0)
- Blue cell moved from (7, 0) to (8, 0) - horizontal movement at top row
- Position (8, 0) aligns with x-coordinate of clicked position (8, 7)
- Movement pattern: vertical upward, then horizontal alignment
- Need to continue to see if blue cell continues moving or trajectory ends
Frame 26: New configuration with two blue cells
- Blue cell at (8, 1) - moved down from (8, 0) in previous frame
- Blue cell at (11, 5) - new appearance at edge of masked region
- Blue cells seem to be materializing and moving in relation to earlier click actions
- Need to continue to see if more blue cells appear or if masked region reveals more content
Frame 27: More blue cells appearing at masked region boundary
- Blue cell at (9, 1) - moved right from (8, 1)
- Blue cell at (11, 2) - new at right edge of masked region
- Blue cell at (11, 4) - another new at right edge of masked region
- Pattern: blue cells emerging at column 11 (right edge of mask) at various rows
- Movement continues for existing blue cells
Frame 28: Blue cell movements and new appearances
- Blue cell at (10, 1) - moved right from (9, 1)
- Blue cell at (12, 1) - new appearance at column 12
- Blue cell at (12, 3) - moved from (11, 2), shifted right and down
- Pattern: continued rightward movement and generation
- Some blue cells disappearing (the one at (11, 4) is gone)
Final analysis:
- Masked region is 8x8 (rows 2-9, cols 4-11)
- Click actions at (7,4) and (8,7) translate to (3,0) and (4,3) in masked region
- Option 0 has blue cells positioned consistently with click pattern
- Other options either too sparse, too dense, or misaligned with observed clicks
```

```
[
    [[\"blue\", \"black\", \"black\", \"black\", \"black\", \"black\", \"black\", \"black\"],
    [\"black\", \"black\", \"black\", \"blue\", \"black\", \"black\", \"black\", \"black\"],
    [\"black\", \"black\", \"black\", \"black\", \"black\", \"black\", \"blue\", \"black\"],
    [\"black\", \"black\", \"black\", \"black\", \"black\", \"black\", \"black\", \"black\"],
    [\"black\", \"black\", \"blue\", \"black\", \"black\", \"black\", \"black\", \"black\"],
    [\"black\", \"black\", \"black\", \"blue\", \"black\", \"black\", \"blue\", \"black\"],
    [\"black\", \"black\", \"blue\", \"black\", \"black\", \"black\", \"black\", \"black\"],
    [\"black\", \"black\", \"black\", \"blue\", \"black\", \"black\", \"black\", \"black\"]],

    [[\"black\", \"black\", \"black\", \"black\", \"black\", \"black\", \"black\", \"black\"],
    [\"black\", \"black\", \"black\", \"black\", \"black\", \"black\", \"black\", \"black\"],
    [\"black\", \"black\", \"black\", \"black\", \"black\", \"black\", \"black\", \"black\"],
    [\"black\", \"black\", \"blue\", \"black\", \"black\", \"black\", \"blue\", \"black\"],
    [\"black\", \"black\", \"black\", \"blue\", \"black\", \"black\", \"black\", \"black\"],
    [\"black\", \"black\", \"blue\", \"black\", \"black\", \"black\", \"black\", \"black\"],
    [\"black\", \"black\", \"black\", \"blue\", \"black\", \"black\", \"black\", \"black\"],
    [\"black\", \"black\", \"black\", \"black\", \"black\", \"black\", \"black\", \"black\"]],

    [[\"black\", \"black\", \"black\", \"black\", \"black\", \"blue\", \"black\", \"black\"],
    [\"blue\", \"black\", \"black\", \"blue\", \"black\", \"black\", \"black\", \"blue\"],
    [\"black\", \"black\", \"black\", \"black\", \"black\", \"black\", \"blue\", \"black\"],
    [\"black\", \"black\", \"black\", \"black\", \"black\", \"blue\", \"black\", \"black\"],
    [\"black\", \"black\", \"blue\", \"black\", \"black\", \"black\", \"black\", \"black\"],
    [\"black\", \"black\", \"black\", \"blue\", \"black\", \"black\", \"black\", \"black\"],
    [\"black\", \"black\", \"blue\", \"black\", \"black\", \"blue\", \"black\", \"black\"],
    [\"blue\", \"black\", \"black\", \"blue\", \"black\", \"black\", \"black\", \"black\"]],

    [[\"black\", \"black\", \"black\", \"black\", \"black\", \"black\", \"black\", \"black\"],
    [\"black\", \"black\", \"black\", \"black\", \"black\", \"black\", \"black\", \"black\"],
    [\"black\", \"black\", \"black\", \"black\", \"black\", \"black\", \"black\", \"black\"],
    [\"black\", \"black\", \"black\", \"black\", \"black\", \"black\", \"black\", \"black\"],
    [\"black\", \"black\", \"blue\", \"black\", \"black\", \"black\", \"black\", \"black\"],
    [\"black\", \"black\", \"black\", \"black\", \"black\", \"black\", \"black\", \"black\"],
    [\"black\", \"black\", \"black\", \"black\", \"black\", \"black\", \"black\", \"black\"],
    [\"black\", \"black\", \"black\", \"black\", \"black\", \"black\", \"black\", \"black\"]],

    [[\"black\", \"black\", \"black\", \"black\", \"black\", \"black\", \"black\", \"black\"],
    [\"black\", \"black\", \"black\", \"black\", \"black\", \"black\", \"black\", \"black\"],
    [\"black\", \"black\", \"black\", \"black\", \"black\", \"black\", \"black\", \"black\"],
    [\"black\", \"black\", \"black\", \"black\", \"black\", \"black\", \"black\", \"black\"],
    [\"black\", \"black\", \"blue\", \"blue\", \"black\", \"black\", \"black\", \"black\"],
    [\"black\", \"black\", \"blue\", \"blue\", \"black\", \"black\", \"black\", \"black\"],
    [\"black\", \"black\", \"blue\", \"blue\", \"black\", \"black\", \"black\", \"black\"],
    [\"black\", \"black\", \"blue\", \"blue\", \"black\", \"black\", \"black\", \"black\"]],

    [[\"black\", \"black\", \"black\", \"black\", \"black\", \"black\", \"black\", \"black\"],
    [\"black\", \"black\", \"black\", \"black\", \"black\", \"black\", \"black\", \"black\"],
    [\"black\", \"black\", \"black\", \"black\", \"black\", \"black\", \"black\", \"black\"],
    [\"black\", \"black\", \"black\", \"black\", \"black\", \"black\", \"black\", \"black\"],
    [\"black\", \"black\", \"black\", \"black\", \"black\", \"black\", \"black\", \"black\"],
    [\"black\", \"black\", \"black\", \"black\", \"black\", \"black\", \"black\", \"black\"],
    [\"black\", \"black\", \"black\", \"black\", \"black\", \"black\", \"black\", \"black\"],
    [\"black\", \"black\", \"black\", \"black\", \"black\", \"black\", \"black\", \"black\"]]
]
```

Listing E.2: Answer choices, indexed 0–5, supplied for masked frame prediction in the 6JVMF environment.

