# OpenReview forum: "Benchmarking World-Model Learning"
_ICLR.cc/2026/Conference — Submitted to ICLR 2026_

### Official Review · Reviewer_L3rY · 2025-10-29

**Soundness:** 3
**Presentation:** 3
**Contribution:** 3
**Rating:** 6
**Confidence:** 4

**Summary:**

This paper introduces WorldTest, a novel evaluation protocol that bifurcates agent assessment into a reward-free "interaction phase" for model building and a "test phase" to evaluate generalization on derived, goal-oriented tasks. The authors implement this in AutumnBench, a benchmark with 43 grid-world environments and 129 tasks. Large-scale experiments with 517 humans and three AI models reveal that humans significantly outperform AI, highlighting fundamental deficits in the models' exploration strategies and hypothesis-testing abilities. The work provides a valuable framework and empirical toolkit for a more rigorous evaluation of world models.

**Strengths:**

- The WorldTest protocol's separation of learning and testing is a key contribution, shifting evaluation from task-specific performance to a deeper understanding of world dynamics. Its representation-agnostic and behavior-based nature ensures broad applicability.
- AutumnBench serves as a rich and extensible implementation, featuring 43 diverse environments and 129 tasks across prediction, planning, and counterfactual reasoning. The underlying DSL facilitates future community expansion.
- The large-scale human study provides a robust baseline, and the analysis moves beyond simple metrics to reveal AI's deeper shortcomings in exploration and metacognition, evidenced by analyses of "reset" usage and action perplexity.
- The paper is well-written, with a clear and logical structure that makes the contributions easy to understand.

**Weaknesses:**

- The empirical evidence is confined to 2D grid-worlds, which may limit the generalizability of the conclusions to more complex physical domains, despite the paper's broader claims.
- The use of a GUI for humans versus a text interface for AI is a potential confound. This difference may partly explain behavioral discrepancies, such as the varied use of the "reset" action, and warrants further discussion.
- Attributing model failures to metacognitive deficits is insightful but would be strengthened by more direct evidence. Ablation studies could help disentangle these deficits from artifacts of prompt design or limitations in pre-training data.

**Questions:**

- Recent work has explored RL fine-tuning LLMs for decision-making. Have you considered evaluating RL fine-tuned models on your benchmark to see if this addresses some of the observed limitations? This would complement the current zero-shot results.
- Could you elaborate on the specific challenges you anticipate when instantiating the WorldTest framework in a more complex domain, such as a 3D physics simulation?
- For the AI models, is there a correlation between the quality of their exploration in the interaction phase (e.g., action perplexity dynamics) and their final performance in the test phase? Such an analysis could offer clues for improving exploration strategies.

---

> ### Author Response · Authors · 2025-11-27
>
> We thank the reviewer for their assessment of WorldTest's contributions and the empirical study. We have addressed the concerns raised in the Weaknesses section. Followed by this, we discuss the individual posed questions below.

---

> ### Author Response · Authors · 2025-11-27
>
> > Discussion on generalizability (or limitations) of 2D grid-world domain.
>
> We thank the reviewer for their assessment of WorldTest's contributions and the empirical study. While we agree that 3D or higher-fidelity instantiations of WorldTest are valuable and would enable joint testing of perception and dynamics learning, we believe 2D grid-based environments remain highly effective for isolating and measuring world-model learning capabilities. The abstraction to 2D grids is a methodological choice that has proven effective in influential prior work, such as:
> - **ARC-AGI [1]**: Uses 2D colored grids to test abstract reasoning, has proven challenging even for frontier models, and has directly led to advances in model architecture and design.
> - **Minigrid/BabyAI [2]**: 2D grid environments have been widely used to study language grounding, hierarchical planning, and sample efficiency in RL.
>
> These examples demonstrate that 2D abstraction does not diminish difficulty or impact. Rather, it isolates reasoning challenges from perception.
>
> **AutumnBench's specific advantages.** Our grid-based instantiation offers:
> 1. **Extensibility:** The Autumn DSL enables rapid authoring of new environments (most <50 lines of code).
> 2. **Human evaluation:** Browser-based rendering enabled large-scale crowdsourced human studies (with 517 participants), which would be prohibitively difficult with 3D environments
> 3. **Controlled evaluation:** Abstract grids eliminate confounds from complex visual perception (like lighting, occlusion) and continuous control, allowing more isolated measurement of world-model learning.
>
> Despite 2D abstraction, AutumnBench remains highly challenging. We view 3D extensions as valuable future work that would complement AutumnBench and add perceptual and control challenges.
>
> **References**
>
> [1] Chollet, F., et al. ARC-AGI-2: A new challenge for frontier AI reasoning systems. arXiv preprint arXiv:2505.11831, 2025.
>
> [2] Chevalier-Boisvert, M., et al. BabyAI: A platform to study the sample efficiency of grounded language learning. arXiv preprint arXiv:1810.08272, 2019.

---

> > ### Author Response · Authors · 2025-11-27
> >
> > > Discussion on the use of a GUI for humans versus a text interface for AI
> >
> > We agree with the reviewer that it is important to fairly compare humans and AI models. We took several steps to ensure the text interface provides AI agents with equivalent, and in some ways superior, information to the graphical interface:
> >
> > The text-based interface provides complete, unambiguous descriptions of:
> > 1. The current grid state (exact cell colors and positions)
> > 2. All available actions at each timestep, including reset, directional keys, click, and task-phase transitions
> > 3. The full interaction history
> > 4. Task-specific information (such as mask regions, answer choices, goal specifications)
> >
> > To confirm that agents perceived and understood the available actions, we logged their complete scratchpads, reasoning traces, and intermediate actions during both interaction and test phases. Analysis of these traces, as detailed in **Appendix D.2.2**, demonstrates that agents:
> > - Correctly list available actions in their reasoning
> > - Form hypotheses about environment dynamics
> > - Execute actions deliberately based on those hypotheses
> >
> > **Example: Claude-4-Sonnet's Scratchpad on Snake-like Environment**
> >
> > Below is an excerpt demonstrating task comprehension despite systematic reasoning errors. The model explicitly lists: _"Available actions: left, right, up, down, click, noop, quit, go-to-test, reset"_ and proceeds to test each action systematically.
> >
> > ```
> > Initial observations:
> > - 16x16 grid, mostly black
> > - Green cells at (7,7) and (7,8) - appears to be a horizontal pair
> > - Pink cell at (8,10) - single cell
> > - Need to test different actions to understand the rules
> > - Available actions: left, right, up, down, click, noop, quit, go-to-test, reset
> >
> > [Agent systematically tests various actions]
> >
> > Action taken: noop
> > Result: CHANGE DETECTED! The green cells have moved!
> > - This suggests the environment has physics/gravity or automatic movement
> >
> > [Agent continues exploration, forming incorrect "gravity" hypothesis]
> >
> > Action taken: noop (eighth time)
> > Result: The green object hit the bottom!
> > Key discovery: Green object stops falling when it hits the bottom boundary.
> > This confirms gravity-like physics with collision detection at grid boundaries.
> > ```
> >
> > The model then attributes the rightward movement to the click rather than the earlier right keypress, failing to revise its gravity hypothesis despite contradictory evidence.
> >
> > ```
> > Action taken: noop
> > Result: CHANGE DETECTED! The green cells have moved!
> > - Green cells moved from (7,7),(7,8) to (7,9),(8,9) - moved down and right
> > - This suggests the environment has physics/gravity or automatic movement
> >
> > Key observation: The green object has fallen down one row and the bottom part
> > moved right one column. This looks like gravity or falling behavior!
> >
> > [Multiple noop actions follow with continued downward movement]
> >
> > Action taken: noop (eighth time)
> > Result: The green object hit the bottom!
> > Key discovery: Green object stops falling when it hits the bottom boundary.
> > This confirms gravity-like physics with collision detection at grid boundaries.
> >
> > [Later, after reset and exploration]
> >
> > Action taken: right
> > Result: No visible change - green object remained at (1,9),(2,9)
> > Observation: Arrow key "right" also did not affect the falling green object.
> >
> > Action taken: click 1 9
> > Result: MAJOR CHANGE! The green object has transformed!
> > - After clicking on (1,9), it has become a horizontal 2-cell object at (2,9),(2,10)
> > - The object moved down one row and rotated from vertical to horizontal
> > ```
> > Despite this awareness of all actions, there are behavioral differences between humans and reasoning models.
> >
> > **Reset action specifically.** The "reset" action is explicitly listed in the action space provided to models at every timestep during the interaction phase. Human participants also receive the same/similar instructions about the reset button in the tutorial. The behavioral difference, therefore, reflects a genuine capability gap in metacognitive exploration strategies rather than an artifact of the interface.

---

> > > ### Author Response · Authors · 2025-11-27
> > >
> > > **Prompt robustness.**
> > >
> > > As detailed in **Appendix E.3** in our revision, to ensure that poor performance is not an artifact of suboptimal prompting, we tested three alternative prompt formulations on a subset of problems (across 8 environments and all 3 task types, totaling 24 problems):
> > >
> > > 1. **new_system_prompt:** More explicit and detailed task instructions
> > > 2. **new_system_prompt_refined:** Enhanced version of 1 using Claude's Prompt Improver tool
> > > 3. **prompt_with_tutorial:** Includes worked examples from the human tutorial
> > >
> > > Performance varied only modestly across prompts (typically Δ < 0.1) and across different environments (after ±):
> > >
> > > | Model           | new_system_prompt | new_system_prompt_refined | prompt_with_tutorial | original        |
> > > | --------------- | ----------------- | ------------------------- | -------------------- | --------------- |
> > > | claude-4-sonnet | 0.1605 ± 0.3389   | 0.0630 ± 0.2142           | 0.1418 ± 0.3442      | 0.2609 ± 0.4490 |
> > > | gemini-2.5-pro  | 0.1880 ± 0.3853   | 0.1917 ± 0.3890           | 0.1667 ± 0.3807      | 0.1667 ± 0.3807 |
> > > | o3              | 0.1539 ± 0.3537   | 0.1764 ± 0.3834           | 0.2214 ± 0.3982      | 0.1769 ± 0.3846 |
> > >
> > > This confirms that the interface modality and task descriptions are not the primary bottleneck.

---

> > > > ### Author Response · Authors · 2025-11-27
> > > >
> > > > > Discussion on disentangling the deficits from artifacts of prompt design
> > > >
> > > > We agree that disentangling prompt artifacts from fundamental capability gaps is essential. To address this, we conducted prompt robustness experiments reported in Appendix E.3 and briefly described above.
> > > >
> > > > Several findings point to fundamental capability limitations rather than prompt artifacts:
> > > >
> > > > **1. Behavioral Patterns:** Analysis of agent scratchpads reveals that agents correctly perceive available actions and form explicit hypotheses, yet still fail to:
> > > >
> > > > - Revise hypotheses when confronted with contradictory evidence
> > > > - Design systematic experiments to test alternative explanations
> > > > - Use resets strategically for hypothesis testing
> > > >
> > > > **2. Qualitative Reasoning Failures:** Even when agents verbalize correct reasoning steps, they fail to integrate them correctly. For example, in a snake environment where the green object (the snake object) moves continuously in its current direction, Claude-4-Sonnet correctly observes and reports individual state transitions, but incorrectly attributes causality. It infers that "gravity" causes downward movement rather than recognizing the latent directional state was set by a previous arrow key action. It later fails to update understanding despite further contradictory evidence when observing rightward movement after choosing the right-arrow action.
> > > >
> > > >
> > > > **3. Robustness to Prompt Improvements:** State-of-the-art prompt engineering techniques (including Claude's Prompt Improver feature) yield no consistent improvements, suggesting the bottleneck lies in the models' reasoning capabilities rather than task understanding.
> > > >
> > > > **Conclusion**
> > > >
> > > > These findings collectively suggest that poor exploration, hypothesis testing, and belief updating reflect genuine metacognitive limitations in current models. While different prompting did make the performance vary, these variantions are minimal.
> > > >
> > > > For full reproducibility, we release complete interaction logs, prompts, and agent traces at https://doi.org/10.5281/zenodo.17728515.

---

> > > > > ### Author Response · Authors · 2025-11-27
> > > > >
> > > > > > Discussion on RL fine-tuning LLMs for decision-making
> > > > >
> > > > > We agree that RL fine-tuning is a promising direction and have made preliminary attempts. However, applying RL fine-tuning to AutumnBench faces fundamental challenges distinct from typical decision-making benchmarks: **(1) reward-free exploration phase:** WorldTest's interaction phase is intentionally reward-free. Agents must explore without explicit task objectives. Standard RL fine-tuning requires reward signals, so adapting it to AutumnBench necessitates either designing intrinsic rewards that encourage hypothesis-driven exploration, which is non-trivial, especially under stochastic dynamics, or pre-training on synthetic exploration curricula, which requires a lot of data, which leads to **(2) data scarcity:** effective RL fine-tuning typically requires large, diverse training corpora. AutumnBench currently offers 43 environments with 129 tasks, sufficient for evaluation but modest for stable RL training.
> > > > >
> > > > > Despite these challenges, we believe RL fine-tuning remains a promising direction. We hope that AutumnBench will spur the development of novel RL algorithms that enable hypothesis-driven exploration and robust world-model learning in LLM agents.
> > > > >
> > > > > > Could you elaborate on the specific challenges you anticipate when instantiating the WorldTest framework in a more complex domain, such as a 3D physics simulation?
> > > > >
> > > > > Thank you for this question. We view 3D instantiations of WorldTest as valuable future work that would enable joint testing of perception and dynamics learning. Some challenges we anticipate are:
> > > > >
> > > > > **1. Designing Modification Functions**
> > > > >
> > > > > WorldTest's modification function $\tau$ transforms the interaction environment $\mathcal{M}$ into a challenge environment $\mathcal{M}'$:
> > > > >
> > > > > $$
> > > > > (\mathcal{M}', R, H) = \tau(\mathcal{M}, \xi)
> > > > > $$
> > > > > While WorldTest does not strictly require programmatic access to environment dynamics, such access greatly simplifies the design of modification functions $\tau$ and the generation of challenge environments.
> > > > > In AutumnBench, $\tau$ can directly inspect and modify dynamics:
> > > > > - Environments are specified in <50 lines of declarative code (Autumn DSL)
> > > > > - Change detection challenges can be generated by mutating specific dynamics rules easily by changing few lines in the original program
> > > > >
> > > > > In 3D physics engines (MuJoCo, PyBullet, Isaac Gym):
> > > > > - Dynamics are implicit in complex numerical solvers (constraint solvers, integrators, collision detectors)
> > > > > - We cannot easily inspect what "rules" govern transitions
> > > > > - Modifications must work through low-level parameters (friction, mass, damping) rather than high-level behavioral rules
> > > > > - Generating controlled dynamics variations (e.g., "gravity now affects only red objects") requires manual engineering of each variation
> > > > >
> > > > > **2: Defining Tractable Queries in Continuous Domains**
> > > > >
> > > > > In AutumnBench, queries have discrete, well-defined answers, but in 3D continuous physics:
> > > > > - States and trajectories are continuous and high-dimensional
> > > > > - Prediction queries require specifying a tolerance for "correct" predictions
> > > > > - Planning queries require solving continuous control problems, confounding world-model evaluation with control ability
> > > > > - Change detection queries are harder to formalize when dynamics are implicit
> > > > >
> > > > > We would need to carefully design queries that test world-model understanding without requiring agents to solve hard continuous control or high-precision prediction problems. This is possible but requires more careful task design than in discrete domains.
> > > > >
> > > > > **3: Scalability of Human Evaluation**
> > > > >
> > > > > WorldTest allows comparing human and AI performance on identical challenges. AutumnBench runs efficiently in the browser, enabling large-scale crowdsourced evaluation (517 participants).
> > > > >
> > > > > 3D physics simulations require:
> > > > > - Specific operating systems and hardware (especially for GPU-accelerated rendering)
> > > > > - Software installation (game engines, physics libraries, graphics drivers)
> > > > > - Higher bandwidth for visual streaming or local rendering
> > > > >
> > > > > These requirements increase human study costs by an order of magnitude and limit sample sizes, making it harder to establish reliable human baselines.
> > > > >
> > > > > **4: Perception-Reasoning Confounds**
> > > > >
> > > > > AutumnBench uses abstract 2D grids to isolate world-model learning from perceptual challenges. The state is represented symbolically and fully observable within local regions.
> > > > >
> > > > > In 3D domains, failures could stem from:
> > > > > 1. Visual perception (depth estimation, object recognition, occlusion handling)
> > > > > 2. Motor control (dexterous manipulation in continuous action spaces)
> > > > > 3. World-model learning (understanding dynamics, planning, counterfactual reasoning)
> > > > >
> > > > > Poor performance on a challenge could reflect any combination of these factors. Disentangling them requires additional control conditions and experiments, whereas 2D grids provide a cleaner testbed for reasoning abilities.

---

> ### Author Response · Authors · 2025-11-27
>
> > Discussion on the correlation between the quality of the agent's exploration in the interaction phase and their final performance
>
> Thank you for the suggestion. We appreciate the opportunity to present this analysis, which we detailed in **Appendix E.4.2** using normalized perplexity metrics as proxies for exploration quality. We write down our findings briefly below:
>
> **Environment-Level Analysis**
>
> Aggregating across all environment-agent pairs (43 environments × multiple agents), we find modest correlations:
>
> - **Final perplexity vs. score:** Pearson r = 0.324, p = 1.06×10⁻⁷
> - **Area under curve (perplexity) vs. score:** Pearson r = -0.214, p = 5.37×10⁻⁴
>
> Higher task scores are associated with slightly *higher* final perplexity and *lower* area under the curve.
>
> **Per-Agent Analysis Across Environments**
>
> When analyzing each agent separately across the 43 environments, correlations are weak (|r| < 0.24, p > 0.13 in all cases):
>
> | Agent           | Final Perplexity vs. Score | Area Under Curve vs. Score |
> | --------------- | -------------------------- | -------------------------- |
> | Claude-4-Sonnet | r = -0.23, p = 0.13        | r = 0.04, p = 0.82         |
> | o3              | r = 0.15, p = 0.34         | r = -0.22, p = 0.15        |
> | Gemini-2.5-Pro  | r = -0.13, p = 0.40        | r = 0.11, p = 0.48         |
> | Humans          | r = 0.00, p = 0.99         | r = 0.12, p = 0.45         |
>
> Additional notes:
> - **For humans:** Near-zero correlation reflects ceiling effects. Humans achieve high scores across nearly all environments, limiting variance.
> - **For AI models:** Weak correlations suggest that while perplexity provides some signal about exploration dynamics, it does not strongly predict task success.
>
> These findings indicate that normalized perplexity metrics may not fully or exclusively serve as proxies for exploration quality; other factors that are difficult to quantify, such as hypothesis formation, coverage of the state space of the base environment, and inference under uncertainty, likely play crucial roles in task success.

---

### Official Review · Reviewer_TiEe · 2025-10-31

**Soundness:** 3
**Presentation:** 3
**Contribution:** 2
**Rating:** 2
**Confidence:** 2

**Summary:**

The paper proposes testing world models beyond the environment of interaction. Traditional model-based RL is only interested in the learned policy given the learned model.
This inherently makes sense to me, given that it is hard to define casually which elements of the environment should be directly learned to guide the learning of a good policy ‘in imagination’.
This benchmark allows for environment interaction during the training phase to learn a world model, but no reward or task is available. It creates a very simple grid-like world inspired by the ARC challenge and Minigrid environments.
 After the test phase, the agent is tasked with solving a task that requires a good world model — for example, a task directly related to reconstructing observations at the end of an episode, rendered as a classification objective (this is chosen to simplify the output space for LLMs).Action planning and change detection from the nominal dynamics.
I don’t fully comprehend how this benchmark is useful for development and provides new interpretability aspects compared to existing benchmarks - it seems to be specifically applicable to LLMs.
The proposed Change Detection and Masked Frame Prediction tasks align with the typical reconstruction loss used in evaluating world models, and the planning corresponds to task success.

**Strengths:**

- large benchmark
- well presented

**Weaknesses:**

- not 100% clear how these benchmarks can guide the design of new algorithms
- interrelation to existing model-based evaluations e.g., with continuous action space is missing
- missing novel insight based on the benchmark or mechanistic interpretability

**Questions:**

Q1: How can this benchmark help in developing new algorithms?

Q2: How is it, in principle, different from evaluating task success and reconstruction error in typical world-modeling studies?

Q3: Often, we are not interested in learning the full dynamics but rather the task-relevant dynamics when considering world-model learning. Any comments on why we enforce learning the full dynamics?

Q4: Can this benchmark support any non-LLM-based methods?

Q5: How well are open-source models performing?

Q6: All LLMs tested seem to perform similarly — why might that be, and do you have any thoughts on this?

---

> ### Author Response · Authors · 2025-11-27
>
> We thank the reviewer for the feedback on the presentation and for raising important questions about applicability beyond LLMs. In the revision, we have added an additional baseline in Appendix E.2 (and described in response to Q4), the Autumn-simulator agent, which is a non-LLM baseline to demonstrate the use case of our framework and benchmark beyond LLMs.

---

> ### Author Response · Authors · 2025-11-27
>
> > Q1: Discussion on how WorldTest and AutumnBench help in developing new methods
>
> Thank you for raising this important point about how our framework and benchmark will guide the development of new models and algorithms.
>
> **WorldTest** enables the development and evaluation of world-model learning agents that rely on intrinsic motivation [1] rather than explicit reward signals. Our experiments demonstrate that existing state-of-the-art AI agents fall far short of human-level performance on such reward-free exploration and world-model learning. It additionally enables us to ask **queries** about the environment, which had two key benefits: (1) it makes WorldTest applicable to any agent architecture: LLMs, learned world models (e.g., Dreamer-v3), program synthesis methods, hybrid approaches, and humans, and (2) queries are strictly more general than evaluating agents by attaching a reward-function to the environment.
>
> For example, consider an agent exploring a dungeon composed of interconnected rooms. Through local exploration, the agent experiences transitions between adjacent rooms but never receives a complete map. A query about the world-model that would indicate correctly learning the world-model is: "Are the blue room and green room connected by any path?" This question requires understanding the graph structure of the dungeon's transition function, a property that cannot be determined from any single trajectory but only from integrating knowledge across multiple explorations.
>
> To our knowledge, no existing benchmark or framework can ask such a query about the world model without constraining its outputs. WorldTest does this using a transformation function $\tau$ that consumes the full base environment $M$ and produces a derived challenge environment $M'$ with explicit objectives that depend on $M$'s structural properties, and thus encodes the query within the challenge POMDP. In the dungeon example, one way to pose such a query could be to show the agent and initial state and 4 options, each with different coloured rooms, and ask it to pick all the reachable states from the given initial state.
>
> **Assessment across diverse agents and tasks**
>
> We instantiate WorldTest as AutumnBench with 43 grid-world environments and three challenge families, each testing a core world-modeling capability that has been deemed important in prior work, through three task-types: masked frame prediction (MFP), change detection (CD), and planning.
>
> These three families yield 129 tasks total, enabling a comprehensive assessment of different aspects of world-model learning. We evaluate humans, LLMs, and a simulator agent on AutumnBench and describe the results below.
>
> **Revealed gaps and algorithmic opportunities**
>
> Our evaluation of 517 humans and frontier reasoning models on AutumnBench reveals specific failure modes that point directly to algorithmic improvements needed beyond scaling compute or training data:
>
> 1. Flexible belief updating under uncertainty. In masked frame prediction tasks, models frequently maintain initial hypotheses even when test-phase observations clearly contradict their learned rules. This rigidity indicates that current architectures lack mechanisms to monitor their own uncertainty, recognize when learned models are inadequate, and trigger appropriate belief revision. The analysis is described in more detail in Section 5 of the paper, with environment-specific breakdowns in Appendix E and qualitative traces showing model reasoning patterns (Appendix E.5).
>
> 2. Limitations beyond predicting the next frame accurately. We evaluate an Autumn-simulator agent with access to ground-truth environment dynamics. Despite this privileged information, it underperforms humans across all three task types, particularly in planning with long horizons and reasoning under stochasticity. This gap proves that pixel-level reconstruction, the dominant training objective for world models, misses structural information needed for answering queries about the environment.
>
> 3. Limitations that go beyond scaling computational resources. Increased computational cost improves performance in only 58% of environments. Change detection benefits least, with only 33% of environments showing improvement. This indicates fundamental reasoning limitations that cannot be overcome by scaling resources alone, i.e., the bottleneck is algorithmic, not computational.
>
> Through AutumnBench, we reveal gaps between humans and AI models in exploring environments and modeling complex transitions in the absence of external rewards.
>
> **References**
>
> [1] An Information-Theoretic Perspective on Intrinsic Motivation in Reinforcement Learning: A Survey. Aubret, Arthur and Matignon, Laetitia and Hassas, Salima. arXiv. https://arxiv.org/abs/2209.08890.

---

> > ### Author Response · Authors · 2025-11-27
> >
> > > Q2: Discussion on if/how MFP and CD are different from reconstruction error and how planning is different from task success
> >
> > We agree that further clarification on the differences between the tasks in AutumnBench and typical reconstruction error will improve the clarity and positioning of the paper. While our tasks assess similar capabilities as traditional benchmarks, there are key differences that make them more challenging:
> >
> > **Masked Frame Prediction** requires conditioning on non-masked trajectory portions in stochastic environments. This is a challenging inference problem even with simulator access. Our Autumn-simulator agent baseline (Appendix E.2) with ground-truth simulator access still falls short of human performance, indicating the task tests inference capabilities beyond typical reconstruction-loss evaluation.
> >
> > **Change Detection** requires detecting changes in dynamics amid stochastic variation in the environments, not simply applying reconstruction loss to individual trajectories. The challenge is distinguishing true dynamics-changes from natural stochastic variation. Humans substantially outperform the Autumn-simulator agent, suggesting they either perform better inference over trajectory distributions or maintain more robust world-model representations supporting counterfactual reasoning.
> >
> > **Planning** does test task success, but with a critical constraint: agents must succeed with only a single attempt since they cannot reset to retry. This requires forward simulation and the ability to recognize and avoid dead states (states from which the goal is unreachable) before committing to actions.
> >
> > We chose these task families because they reflect capabilities emphasized in the world-modeling literature. However, our results with the Autumn-simulator agent (described in Appendix E.2) show that possessing these capabilities in isolation does not yield human-level performance.

---

> > > ### Author Response · Authors · 2025-11-27
> > >
> > > > Q3: Discussion on why we enforce agents to learn the full dynamics of the environment
> > >
> > > Thank you for the question. We agree that the agent should focus on learning the dynamics of the world that are important to solving tasks or answering questions about that world. We inform agents of the task type (masked frame prediction, change detection, or planning) before the interaction phase, allowing them to focus on task-relevant aspects. Our analysis shows that humans adapt their interaction length based on task type, suggesting they successfully prioritize learning relevant information. For example, planning tasks require understanding the model enough to avoid stale states, and consequently, humans spend less time interacting during planning tasks than during other tasks.
> > >
> > > In addition to that, we manually design all test-phase tasks to ensure they test understanding of abstract rules (e.g., what happens at collisions, border behavior, which objects have agency) rather than memorization or pixel-perfect recall. This is why even our Autumn-simulator baseline, which has perfect next-frame prediction, still underperforms humans. The tasks require reasoning about dynamics at a conceptual level beyond raw prediction.
> > >
> > > This design choice has important implications for building better world-model learning agents: it suggests that progress in world-model learning agents requires moving beyond pixel-level reconstruction objectives toward methods that explicitly capture structural properties and causal relationships in the dynamics.

---

> > > > ### Author Response · Authors · 2025-11-27
> > > >
> > > > > Q4: Discussion on non-LLM baseline
> > > >
> > > > Thank you, we agree that adding non-LLM-based baselines would allow a more comprehensive analysis of existing methods' performance.
> > > >
> > > > To concretely demonstrate that AutumnBench applies beyond LLMs, we added a non-LLM baseline in our revision (Appendix E.2): the **Autumn-simulator agent**. This agent has direct access to the ground-truth Autumn interpreter and the base environment program, making it comparable to next-frame prediction models like Dreamer-v3, and providing an approximation to the upper bound on what next-frame prediction alone can achieve. It uses task-specific solvers that are based on reconstruction error for MFP and change detection and breadth-first search for planning:
> > > > - **Masked frame prediction**: Samples 10,000 trajectories from the interpreter and selects the most frequent outcome after filtering for consistency with unmasked frames.
> > > > - **Change detection**: Compares observed frames to expected next‑frame predictions for 10,000 sample trajectories and flags the first discrepancy.
> > > > - **Planning**: Performs breadth‑first search over symbolic interpreter states from the start state, with depth and breadth limits of 10 on 10,000 samples.
> > > >
> > > > Despite having perfect next‑frame predictions, this baseline still underperforms humans on all three tasks, revealing the following challenges for such approaches:
> > > >
> > > > - **Masked frame prediction**: Exact inference requires reasoning over multiple stochastic trajectories consistent with partial observations. This is a challenging inference problem even with ground-truth dynamics.
> > > > - **Change detection**: Distinguishing true dynamics changes from stochastic variation requires comparing distributions over multiple trajectories rather than single next‑frame predictions.
> > > > - **Planning**: BFS lacks heuristic guidance and expands exponentially under stochasticity and long horizons, often reaching the branching limit (depth/breadth of 10) before finding solutions. Even with perfect dynamics knowledge, the search becomes intractable.
> > > >
> > > > This demonstrates that AutumnBench tests meaningful world-model capabilities that challenge even agents with perfect predictive models, validating its utility for developing general-purpose world-modeling systems beyond current LLMs.

---

> > > > > ### Author Response · Authors · 2025-11-27
> > > > >
> > > > > > Q5: Discussion on how open-source models perform
> > > > >
> > > > > In the original submission, we evaluated the open-source model Qwen3-235B-A22B-Thinking-2507 but only included it in Figure 3 (cost vs. score per problem across environments). In the revised version, we now report Qwen3's full results across all analyses in Figures 4-6. The expanded analysis reveals that Qwen3 performs comparably to proprietary models overall, but shows notable weaknesses on specific task types:
> > > > >
> > > > >
> > > > > - Masked Frame Prediction: Similarly, poor performance indicates challenges with inference over stochastic trajectories and reasoning about partially observed sequences.
> > > > > - Change Detection: Qwen3 performs worst among all evaluated models, suggesting particular difficulty with detecting changes in the dynamics amid stochastic variation.
> > > > > - Planning: Performance is similar to Gemini-2.5-Pro, comparable to mid-tier proprietary models.
> > > > >
> > > > > Overall, Qwen3 exhibits the same fundamental deficits as proprietary models: poor hypothesis-driven exploration, difficulty handling stochasticity, and limited counterfactual reasoning. This suggests that while open-source models are approaching proprietary models in planning capabilities, they lag further behind on tasks requiring robust probabilistic reasoning and belief updating.

---

> > > > > > ### Author Response · Authors · 2025-11-27
> > > > > >
> > > > > > > Q6: Discussion on differences in performance of LLMs
> > > > > >
> > > > > > As a reference, we would like to mention that at release, the performance of all models on SWE-Bench [2] was also quite similar. We suspect this is because the training pipeline of models does not incorporate tasks that have a similar structure to that of AutumnBench, resulting in similar performance across models. That said, we do observe behavioral differences in the models, which are discussed in Section 4:
> > > > > >
> > > > > > **1. Exploration strategies (Figure 5 and Appendix E.4.1):** Models exhibit distinct action distributions during the interaction phase:
> > > > > > - Claude-4-Sonnet shows the highest action perplexity and more diverse exploration patterns
> > > > > > - o3 demonstrates more focused, lower-perplexity behavior, suggesting more directed hypothesis testing
> > > > > > - Gemini models fall between these extremes
> > > > > >
> > > > > > **2. Perplexity over time (Figure 6):** Models differ in how their uncertainty around actions they take evolves:
> > > > > > - Some models (e.g., Claude) maintain high perplexity throughout, suggesting trying different actions constantly
> > > > > > - Others (e.g., o3) converge to lower perplexity faster, potentially indicating premature commitment to incorrect hypotheses
> > > > > > - As detailed in Appendix E.4.2, the correlation between perplexity metrics and performance varies across models as well
> > > > > >
> > > > > > **3. Task-specific strengths:** Per-environment analysis (Appendix E.2) shows models excel on different environments and task types, suggesting different inductive biases.
> > > > > >
> > > > > > These differences suggest that while current models share fundamental limitations in metacognitive reasoning, their underlying architectures and training procedures create distinct behavioral profiles.
> > > > > >
> > > > > > **References**
> > > > > >
> > > > > > [2] SWE-bench: Can Language Models Resolve Real-world Github Issues? Carlos Jimenez, John Yang, Alexander Wettig, Shunyu Yao, Kexin Pei, Ofir Press, Karthik R Narasimhan. ICLR 2024.

---

### Official Review · Reviewer_piDb · 2025-10-31

**Soundness:** 3
**Presentation:** 2
**Contribution:** 3
**Rating:** 4
**Confidence:** 5

**Summary:**

This work focuses on evaluating the ability of models - LLMs in particular - to develop a world model of an environment. The contribution is two fold, first introducing a general purpose framework for evaluating a world model (World Test) then a suite of gridworld tasks that implement the world test framework (Autumn Bench). Results are then presented across human participants and reasoning models, showing that human out-perform LLMs, and that LLM performance can't always be improved by having more compute time.

**Strengths:**

The paper is well scoped to a conference paper and relatively well written. The authors clearly contextualise their approach within the existent literature on evaluating world models. Additionally the framework for world model evaluation seems well defined and described.

**Weaknesses:**

There are two weaknesses worth highlighting, first the overall presentation and polish of the paper, second the methodology for evaluating LLMs.

The paper needs to be proofread, there are a number of typos throughout (in particular the same sentence appears to be repeated verbatim at line 341 and 343). Additionally all empirical results are reported in plots none of which have titles or axis labels. This is particularly confusing in figure 6 where plots share a y axis despite showing different quantities. I would also note the discussion section introduces a collection of concepts - like metacognition at line 460 - which have not previously been introduced or unpacked.

In comparing LLM to Human Performance it seems to rely at least in part on the quality of the prompt describing the task. By comparison the human user interface is well designed - have the authors tried a variety of prompting strategies to allow them to confidently say their findings represent the best these models can do?

**Questions:**

To what degree is the poor performance of the LLMs a result of the task formulation in the prompt?

Given that these are grid world environments, can you provide a clearer notion of what this method can and cannot evaluate compared with say a 3d simulated environment?

---

> ### Author Response · Authors · 2025-11-27
>
> Thank you for catching the issues with the clarity of the paper. In response to the changes, we have
> - Fixed all typos, including the repeated sentence at lines 341 and 343
> - Added titles and axis labels to all plots
> - Clarified Figure 6 by using separate y-axes for different quantities
> - Introduced and defined the concept of metacognition earlier in the paper.
>
> Additionally, thank you for the insightful suggestions. We have revised the paper accordingly and added these discussion points below:

---

> > ### Author Response · Authors · 2025-11-27
> >
> > > 1. To what degree is the poor performance of the LLMs a result of the task formulation in the prompt?
> >
> > Thank you, we agree that analyzing the root cause of the poor performance of these LLMs would allow us to
> > have more insights into revealing these agents' weaknesses to construct better agents, and also to improve the
> > clarity of the paper.
> >
> > **Prompt design**
> >
> > We believe it is very important to provide fair settings for both humans and agents.
> > Thus, we have made sure that in the prompt, the agent receives all the following information:
> > 1. The evaluation structure. This is encoded in the prompt as follows for the masked frame prediction:
> > > Welcome, you are now in the interactive phase, where you can interact with the grid using the available actions.
> > > During the interactive phase your goal is to act in the environment to understand the underlying rules of the environment.
> > > Understand the environment and the dynamics of the environment well.
> > and
> > > After the interactive phase, you will be asked to use this knowledge about
> > the environment to answer some questions about it.\n\n# Task Description:\nIn the
> > test phase, you will step through frames from a trajectory in this same
> > environment you interacted with (use the `step` action to step through the
> > trajectory).
> > >  Towards the end of the trajectory, parts of the grid will be masked (where the masked locations are marked as `mask`)
> > and you will be given a set of choices to fill in the masked region at the final timestep. You need to choose option that fits the masked region at the final
> > timestep.
> >
> > 2. The schema of the observation:
> > > Here is the initial state of the grid:
> > > [[\"black\", \"black\", \"red\", ...], ...]
> > 3. The action space.
> > > The following actions are available at this step:
> > > left,
> > > right,
> > > up,
> > > down,
> > > click [0-9] [0-9],
> > > noop,
> > > quit,
> > > go-to-test
> > > reset
> >
> > Above are some examples from the masked frame prediction task. We leave the full version of these exact prompts in the
> > Appendix D.2, Appendix E.1, from Listing D.2 to D.4 for each task.
> > Furthermore, to ensure that the agent received and incorporated the information provided in the prompt,
> > and whether the agent's performance is due to a different prompting style, we conducted two complementary analyses in our revision: (1) we examined the agent's scratchpads to verify task comprehension, and (2) we tested alternative task formulations in the prompts.
> >
> > We discuss more details on these two analyses in our following comments.

---

> > > ### Author Response · Authors · 2025-11-27
> > >
> > > **Analysis 1: Qualitative analysis on whether the agent incorporated task information in the reasoning trace**
> > >
> > > While we incorporated the complete description of the tasks for both the interaction and evaluation phases in Appendix D.2 and Appendix E.1, we agree that it would be interesting to verify that models perceived and used this information. Thus, we logged their complete scratchpads, thoughts, and all intermediate actions during both interaction and test phases.
> > >
> > > We log the full traces for this analysis on Zenodo: https://zenodo.org/records/17728515. To make our discussion self-contained, we quote an excerpt from Claude-4-Sonnet's reasoning trace on a snake-like environment. We can see from these excerpts that the agents managed to comprehend the task but made systematic reasoning errors:
> > >
> > > 1. The agent comprehends the environment, including which cells are at which position, and which action led to which effects. Here, the agent sees the Green object (snake) and the Pink object (food), and the agent also managed to recognize movements. Yet, the model misattributes the snake's movement: it infers "gravity" rather than recognizing that a prior down action set the latent motion direction of the snake object.
> > >
> > > ```
> > > Initial observations:
> > > - 16x16 grid, mostly black
> > > - Green cells at (7,7) and (7,8) - appears to be a horizontal pair
> > > - Pink cell at (8,10) - single cell
> > > - Need to test different actions to understand the rules
> > > - Available actions: left, right, up, down, click, noop, quit, go-to-test, resetAction taken: right
> > > Result: No visible change to the grid - all objects remained in same positions
> > > - Green cells still at (7,7) and (7,8)
> > > - Pink cell still at (8,10)
> > >
> > > Hypothesis: Arrow keys might not directly move objects, or there might be a selected object conceptAction taken: click 7 7
> > > Result: No visible change to the grid - objects remain in same positions
> > > - Green cells still at (7,7) and (7,8)
> > > - Pink cell still at (8,10)
> > >
> > > [some additional hypotheses and actions]
> > >
> > > Summary so far:
> > > - Tried all 4 arrow keys (left, right, up, down) - no visible effects
> > > - Tried clicking on both green and pink objects - no visible effects
> > > - Objects seem static or there might be hidden mechanics
> > >
> > > Next approaches to try:
> > > - Maybe try "noop" to see if time/physics progresses automatically
> > > - Try clicking on empty black cells
> > > - Try multiple actions in sequence
> > > - Consider that the environment might be more complex than direct object manipulationAction taken: noop
> > > Result: CHANGE DETECTED! The green cells have moved!
> > > - Green cells moved from (7,7),(7,8) to (7,9),(8,9) - moved down and right
> > > - Pink cell moved from (8,10) to (8,10) - stayed in same position
> > > - This suggests the environment has physics/gravity or automatic movement
> > >
> > > Key observation: The green object (which appears to be a 2-cell object) has fallen down one row and the bottom part moved right one column. This looks like gravity or falling behavior!Action taken: noop (second time)
> > > Result: The green object continued to fall!
> > > - Green cells moved from (7,9),(8,9) to (8,9),(9,9) - fell down one more row
> > > - Pink cell stayed at (8,10) - still not moving
> > > - The green object appears to be a 2-cell vertical object that falls due to gravity
> > > - Pink object appears to be stable/not affected by gravity
> > >
> > > Pattern observed: Green object is falling downward each turn when noop is used. This confirms gravity-like physics in the environment.Action taken: noop (third time)
> > > Result: The green object continued to fall!
> > > - Green cells moved from (8,9),(9,9) to (9,9),(10,9) - fell down one more row
> > > - Pink cell stayed at (8,10) - still not moving
> > > - The green object is now at rows 9-10, column 9
> > >
> > > [some more observations with `noop` action]
> > >
> > > Pattern continues: Green object falls steadily downward with each noop action. Should hit bottom soon.Action taken: noop (sixth time)
> > > Result: The green object continued to fall!
> > > - Green cells moved from (11,9),(12,9) to (12,9),(13,9) - fell down one more row
> > > - Pink cell stayed at (8,10) - still not moving
> > > - The green object is now at rows 12-13, column 9
> > > - Very close to the bottom now (only 2-3 more rows until bottom edge at row 15)
> > >
> > > CRITICAL DISCOVERY: When the green object hits the bottom, it splits and one part teleports to the top of the same column! This is a major physics rule - objects that hit the bottom boundary undergo some kind of splitting/teleportation behavior.Action taken: noop (tenth time)
> > > Result: The green cells have merged and started falling again!
> > > - The two separate green cells at (0,9) and (15,9) have merged into a 2-cell vertical object at (0,9) and (1,9)
> > > - The bottom green cell disappeared from (15,9)
> > > - Pink cell stayed at (8,10) - still not moving
> > > - The green object is now falling again from the top!
> > > ```

---

> > > > ### Author Response · Authors · 2025-11-27
> > > >
> > > > 2. Agent recognizes different actions but later attributes a rightward movement to a click, rather than recognizing its previous right action. It fails to revise its earlier hypothesis about gravity moving the snake in light of new evidence and to reason abstractly about its own actions and the resulting observations.
> > > >
> > > > ```
> > > > Observation: Arrow keys don't seem to directly move the falling objects. The green object is still in the same position where it was after the last noop.Action taken: right
> > > > Result: No visible change - green object remained at (1,9),(2,9)
> > > > - Green cells still at rows 1-2, column 9
> > > > - Pink cell still at (8,10)
> > > > - Arrow key "right" also did not affect the falling green object
> > > >
> > > > Observation: Both left and right arrow keys don't move the falling green object. This suggests arrow keys either don't control objects directly, or there's a different interaction mechanism needed.Action taken: click 1 9
> > > > Result: MAJOR CHANGE! The green object has transformed!
> > > > - The green object was at (1,9),(2,9) as a vertical 2-cell object
> > > > - After clicking on (1,9), it has become a horizontal 2-cell object at (2,9),(2,10)
> > > > - The object moved down one row and rotated from vertical to horizontal
> > > > - Pink cell still at (8,10) - unchanged
> > > > ```
> > > >
> > > > 3. The agent incorporate understanding on the task with its prior reasoning, for example, in the masked frame prediction tasks, the scratchpad shows that, despite correctly mapping the options to the mask, the model answers incorrectly because it learned the wrong world model: it fails to represent that the snake consumes food upon collision, leading to a confident, yet wrong choice.
> > > >
> > > > ```
> > > > TRAJECTORY PREDICTION UPDATE:
> > > > - Green object vertical physics: Likely falling downward
> > > > - Frame 23 → Frame 24: Vertical at (9,9),(10,9) → likely moved to (10,9),(11,9) or (8,9),(9,9)
> > > > - Frame 24 → Frame 25: Continued vertical movement
> > > > - Pink object: Static at (8,10) - no change expected
> > > >
> > > > MASK ANALYSIS:
> > > > - The mask covers exactly where both objects should be
> > > > - This is building toward the final test
> > > > - I need to predict the final configuration under the mask
> > > > - Based on vertical physics, green object is likely moving vertically
> > > > - Pink object should be unchangedFINAL FRAME 25 ANALYSIS - CHOICE TIME:
> > > >
> > > > The trajectory has ended with "is_finished": true and I'm presented with 6 options to fill the masked region.
> > > >
> > > > MASKED REGION ANALYSIS:
> > > > - Mask covers rows 7-10, columns 7-10 (4x4 square)
> > > > - But the options show 3x4 patterns, suggesting the mask represents rows 7-9, columns 7-10
> > > >
> > > > ...
> > > >
> > > > Frame 24-25: Objects hidden under mask, but physics should continue
> > > > - Green object: Vertical, likely continued downward movement
> > > > - Pink object: Static, should remain at (8,10)
> > > >
> > > > PHYSICS PREDICTION:
> > > > Based on established vertical physics:
> > > > - Green object in Frame 23: (9,9),(10,9)
> > > > - Green object in Frame 24: Likely moved down to (10,9),(11,9)
> > > > - Green object in Frame 25: Likely moved down to (11,9),(12,9)
> > > >
> > > > BUT WAIT - the mask only covers rows 7-10, and the options show 3x4 patterns.
> > > > This suggests the mask represents rows 7-9, columns 7-10.
> > > >
> > > > REVISED PREDICTION:
> > > > If mask is rows 7-9, columns 7-10:
> > > > - Green object last seen at (9,9),(10,9) in Frame 23
> > > > - Row 9 is at the bottom of the mask
> > > > - Row 10 is outside the mask
> > > > - So green object likely moved down but bottom part went outside mask
> > > >
> > > > Looking at the options:
> > > > - Each option is 3 rows × 4 columns
> > > > - This matches rows 7-9, columns 7-10 of the mask
> > > > - Need to predict where green and pink objects are within this region
> > > >
> > > > POSITION MAPPING:
> > > > - Pink object at (8,10) maps to row 1, column 3 in the 3x4 options
> > > > - Green object at (9,9) maps to row 2, column 2 in the 3x4 options
> > > > - If green moved down, (10,9) would be row 3, column 2 - but that's outside the 3x4 mask
> > > >
> > > > FINAL PREDICTION:
> > > > Based on Frame 23 positions:
> > > > - Pink at (8,10) → row 1, column 3 in options
> > > > - Green at (9,9) → row 2, column 2 in options
> > > > - If green moved down to (10,9), that would be outside the mask
> > > >
> > > > Looking at options:
> > > > Option 2: "black black black pink\nblack black green pink\nblack black black black"
> > > > - Has pink at row 0, col 3 (matches (8,10))
> > > > - Has green at row 1, col 2 (close to (9,9))
> > > > - But has pink in two places...
> > > >
> > > > Option 3: "black black green black\nblack black green pink\nblack black black black"
> > > > - Has green at row 0, col 2 and row 1, col 2 (vertical green object)
> > > > - Has pink at row 1, col 3 (matches (8,10))
> > > > - This matches a vertical green object spanning two rows!
> > > >
> > > > Option 3 looks most consistent with:
> > > > - Vertical green object at column 2 (maps to original column 9)
> > > > - Pink object at row 1, col 3 (maps to original (8,10))
> > > > ```
> > > >
> > > > **Summary**: The traces demonstrate that models comprehend task descriptions and available actions, but frequently err due to learning incorrect world models. These are failures of reasoning rather than misunderstandings of the task. These limitations persist beyond prompt phrasing alone.

---

> > > > > ### Author Response · Authors · 2025-11-27
> > > > >
> > > > > **Analysis 2: Prompt robustness experiments**
> > > > >
> > > > > Finally, to further verify that poor performance is not an artifact of suboptimal prompting, we evaluated a subset of problems (8 environments across 3 task types, resulting in 24 problems) under three alternative system prompts:
> > > > >
> > > > > 1. **new_system_prompt**: A more descriptive task specification
> > > > > 2. **prompt_with_tutorial**: A prompt including the full human tutorial example
> > > > > 3. **new_system_prompt_refined**: A refined variant of (1) produced using Claude's Prompt Improver tool with example traces ([link](https://docs.anthropic.com/en/docs/build-with-claude/prompt-engineering/prompt-improver)).
> > > > >
> > > > > Full experimental details, exact prompts, and complete interaction logs are provided in **Appendix E.3** and released as an anonymized dataset at https://doi.org/10.5281/zenodo.17728515.
> > > > > For convenience, we reproduce the results table here:
> > > > >
> > > > > | Model           | new_system_prompt | new_system_prompt_refined | prompt_with_tutorial | original        |
> > > > > | --------------- | ----------------- | ------------------------- | -------------------- | --------------- |
> > > > > | claude-4-sonnet | 0.1605 ± 0.3389   | 0.0630 ± 0.2142           | 0.1418 ± 0.3442      | 0.2609 ± 0.4490 |
> > > > > | gemini-2.5-pro  | 0.1880 ± 0.3853   | 0.1917 ± 0.3890           | 0.1667 ± 0.3807      | 0.1667 ± 0.3807 |
> > > > > | o3              | 0.1539 ± 0.3537   | 0.1764 ± 0.3834           | 0.2214 ± 0.3982      | 0.1769 ± 0.3846 |
> > > > >
> > > > > Across models, alternative prompts produced only modest changes (typically Δ < 0.1) and did not close the gap to human performance, indicating that poor performance reflects reasoning limitations rather than prompt artifacts.

---

> > > > > > ### Author Response · Authors · 2025-11-27
> > > > > >
> > > > > > > Given that these are grid world environments, can you provide a clearer notion of what this method can and cannot evaluate compared with say a 3d simulated environment?
> > > > > >
> > > > > > Thank you, we agree that it would be clarifying to discuss the positioning of AutumnBench to other 3D-simulated simulations. Building AutumnBench in 2D grid-based environments currently offers key advantages:
> > > > > >
> > > > > > 1. They let us reduce complexity and isolate tests to what we care about (i.e., learning new dynamics rather than solving complex perception problems and motor control), making it easier to separate prior knowledge of agents from learning new rules.
> > > > > > 2. They enable large-scale human data collection (517 participants across 129 tasks) and allow comparisons with AI models at a manageable cost. This provides statistical power to detect real differences in learning, which is useful to later build AI models with similar capabilities.
> > > > > >
> > > > > > While 2D grid-worlds have clear advantages, we would need to instantiate WorldTest in 3D environments for robotics and embodied AI applications. A 3D instantiation would introduce complementary evaluation dimensions:
> > > > > >
> > > > > > 1. They enable the assessment of continuous physics and spatial reasoning that better reflect real-world dynamics.
> > > > > > 2. They introduce more complicated visual perception challenges, such as object recognition and handling occlusion, that are necessary for an agent to eventually perform well in the real world.

---

### Official Review · Reviewer_wsyV · 2025-11-01

**Soundness:** 3
**Presentation:** 3
**Contribution:** 2
**Rating:** 6
**Confidence:** 4

**Summary:**

This paper challenges the current view of world model evaluation by proposing a new inference time protocol called “WorldTest”. WorldTest contains two phases that a pretrained reasoning model or a human subject undergoes: (1) Goal & reward free interaction phase and (2) Test phase. During the interaction phase, the agent explores the “rules” of the environment with no explicit reward function or goal state. During the test phase, the agent must solve a specific objective on an environment derived from the base environment. The same process is followed to evaluate world model capabilities in humans. The paper reports that humans vastly outperform reasoning agents due to their superior metacognitive abilities like experiment design, uncertainty quantification and flexible belief updates and suggests improving these abilities in current reasoning models.

**Strengths:**

1. The goal of this paper to tackle evaluation of world model capabilities is important and well timed. There has been speculation among researchers regarding world model capabilities in reasoning models. The final results expose this gap for LLM reasoning agents.
2. The paper presents a simple and intuitive idea to use a reward free exploration phase and a goal oriented test phase for world model evaluation.
3. Experiments and results are well presented and easy to follow. If the method is truly novel and first of its kind, it has the potential to become a standard world model benchmark for reasoning and agent based LLMs in the near future.
4. Use of “reset” action in humans as a hypothesis testing tool is an insightful conjecture - something that did not cross my mind immediately when reading the paper.

**Weaknesses:**

Also adding Questions here along with Weaknesses:
1. While I really liked the presented idea, I am not sure how novel the contributions are when compared to DiscoveryWorld [1] and URLB [2]. I find the paper lacking in justifying the the utility of their implemented WorldTest method against numerous similar testbeds available. For example, URLB presents very similar ideas with the only difference being that during test phase, the agent is tested on the same environment. A simple variation on the environments used in URLB like changing the robot/walker parameters (size, gripper, etc) would yield a WorldTest like scenario.
2. Do the authors claim “AutumnBench” as a part of the contribution? If so, is there a specific reason for creating and using “AutumnBench” instead of other existing tasks like BabyAI [3] or ARC-AGI2 [4]. I can imagine a minigrid BabyAI environment could be easily tuned in such a way that there is no inherent goal during the exploration phase and the already existing difficulty levels could provide variations to the existing task for the “Test” phase (masked prediction, planning, etc).
3. I found the categorizatoin of world model evaluations on Page 1 and 2 rather vague and confusing (non-interactive, representation-based, gym-like, unsupervised RL). It does not seem to serve any purpose other than introducing a few related works. Perhaps move them to a related works section and contrast the paper’s approach against the past works more clearly? This would benefit future readers in assessing why they would want to use WorldTest instead of other benchmarks.
4. As a follow up to the related works, The authors consider ARC-AGI [5] as a non-interactive environment. However, it seems like (a) an internal rule based system/proto-world model based on shown examples (to answer - how did we arrive from initial to final state?) is required to solve the test problem and (b) several interactions must be made to select grid size and place colored tiles on the grid during test. Could AutumnBench be a more difficult version of ARC-AGI (similar to ARC-2, ARC-3)? If not, please clarify where exactly the added complexity and novelty lies for AutumnBench.
5.  Why were traditional world models from RL literature like Dreamer-v3, Transformer WMs (IRIS), etc not compared or discussed in this paper? To adapt this benchmark to traditional WMs, it seems like a simple conversion of current observation grids to integer vectors instead of string vectors and enumerated actions to another output vector for a policy/planning method. Rewards could be removed during the WM learning/exploration phase. How would such a model be evaluated and how would they perform?
6. How are the environments changed during the test phase for Masked Frame Prediction and Planning tasks? I understand that the Change Detection task inherently requires some perturbation to the dynamics. For the other tasks, are there specific rule based systems that consistently alter the environments? Would the authors provide examples for clarity, if any?
7. While the authors promptly present a few abstract conjectures on why reasoning models failed to match the scores of humans, it would have been great to add a discussion on more concrete steps to improve current reasoning models. How would one go about adding the metacognitive abilities like “better uncertainty quantification” and “felxible belief updating”? I would appreciate a few sentences on this topic during the discussion phase.
8. The authors mention several times - "Due to cost constraints, we evaluated each model’s performance based on a single trajectory completion per problem". To help improve the reproducibility of this work, would the authors be open to disclosing and/or discussing the cost of evaluating the three models used in the paper?
9. **Not part of my evaluation**: I am curious what the authors think about evaluating their testbed on an RL based finetuned reasoning model. Several works have shown that RL based finetuning on a more specific set of behaviors yield superior performance [6] [7]. Similarly, what would they comment on program synthesis methods like [8] or human cognition inspired models like [9]. I would like to hear the authors’ take on how each method would score against WorldTest and harder benchmarks.

**Questions:**

Asked in the weakness section. Please refer above.

I lean towards a weak accept since I think such a dataset and evaluation protocol is essential to uncover the flaws in today's reasoning agents that are widely being used for coding and other tasks. These tasks require vaguely similar meta-cognitive components that today's reasoning agents severly lack. However, I'd recommend the authors to further develop the work towards (1) comparison with more types of world models seen in RL and (2) concrete discussion on improving world model capabilities of current reasoning models.

References

*[1] Peter Jansen, et. al. Discoveryworld: A virtual environment for developing and evaluating automated scientific discovery agents. Advances in Neural Information Processing Systems, 2024.*

*[2] Michael Laskin, et. al. Urlb: Unsupervised reinforcement learning benchmark. Advances in
Neural Information Processing Systems, 2021*

*[3] Chevalier-Boisvert, M., D’Eramo, C., Willems, L., Beaudoin, P., Pascanu, R., & Courville, A. (2019). BabyAI: A platform to study the sample efficiency of grounded language learning. arXiv preprint arXiv:1810.08272.*

*[4] Chollet, F., Knoop, M., Kamradt, G., Landers, B., & Pinkard, H. (2025). ARC-AGI-2: A new challenge for frontier AI reasoning systems. arXiv preprint arXiv:2505.11831*

*[5] François Chollet. On the measure of intelligence. arXiv preprint arXiv:1911.01547, 2019*

*[6] Zhai, S, et. al. (2024). Fine-tuning large vision-language models as decision-making agents via reinforcement learning. arXiv:2405.10292*

*[7] Chen, J., et.al. (2025). G1: Bootstrapping perception and reasoning abilities of vision-language models via reinforcement learning. arXiv preprint arXiv:2505.13426*

*[8] https://substack.com/home/post/p-172998849*

*[9] Wang, G., et.al. (2025). Hierarchical Reasoning Model [Preprint]. arXiv. [https://arxiv.org/abs/2506.21734]*

---

> ### Author Response · Authors · 2025-11-27
>
> We thank the reviewer for their comprehensive assessment and helpful suggestions. We discuss each comment below.
>
> **Comment**:
> > 1.  Discussion on positioning WorldTest and comparisons to URLB and DiscoveryWorld
>
> We agree that clarifying the distinction between WorldTest and other frameworks like DiscoveryWorld and URLB will strengthen the position of the paper. In response, we have made this clarification in Section 1 (Introduction) in our revision and discuss the details below.
>
> Specifically, the distinction lies in WorldTest's ability to pose flexible _queries_ about an environment, and test it in an automated manner, which is a more general approach than attaching a fixed reward function to trajectories (i.e., sequence of actions, observations). We describe in more detail with a motivating example below.
>
> Consider an interaction environment $\mathcal{M}$ that is a dungeon with many interconnected rooms. During interaction, the agent observes local transitions through its actions: moving between adjacent rooms, opening doors, and collecting items. Thus, the agent receives no global information about the dungeon's structure.
>
> In both DiscoveryWorld's reward-based settings and URLB, an agent might navigate successfully to various goals (rooms) during interaction, performing well on trajectory-based metrics, yet still hold incorrect beliefs about the environment's structure.
>
> Suppose we want to evaluate whether the agent has learned the true structure of this dungeon, a natural question is:
>
> **"Are the blue room and the green room connected by any path in the true dungeon layout?"**
>
> This asks about a global property of the environment's transition dynamics $T$: whether two states are connected in the transition graph. Answering requires understanding the environment's structure beyond what any single trajectory reveals: for example, a single trajectory might take the agent from the starting room to the blue room, then to several other rooms, but never to the green room. From this trajectory alone, we cannot determine whether the blue and green rooms are connected.
>
> Errors in learning the structure wouldn't necessarily hurt performance on seen tasks, but would prevent the agent from being able to answer broader queries about the interaction environment:
>
> - "What if this door were locked?"
> - "If action A were unavailable, could we still reach state X?"
> - "What is the shortest path between two locations?"
> - "Which regions are unreachable from the starting state?"
>
> Finally, while DiscoveryWorld also poses queries to the agents by having them describe environment dynamics in natural language, the evaluation of these queries is either manual (by using humans), or costly and unreliable (with LLM judges).

---

> > ### Author Response · Authors · 2025-11-27
> >
> > To complement the discussion above, we provide a more formal comparison between URLB and WorldTest below.
> >
> > URLB evaluates agents via downstream objectives of the form:
> >
> > $$\mathbb{E}\_{\tau \sim (\mathcal{M},\pi)}[f\_{\mathrm{obj}}(t)],$$
> >
> > where $t$ is a trajectory sampled from policy $\pi$ in environment $\mathcal{M}$, and $f_{\mathrm{obj}}$ operates solely on this trajectory. The evaluation asks: "Given this single rollout, how well did the agent perform?"
> >
> > **The Limitation**
> >
> > Such objectives cannot query properties of $\mathcal{M}$ that are not revealed through individual trajectories. In our dungeon example, a single trajectory might take the agent from the starting room to the blue room, then to several other rooms, but never to the green room. From this trajectory alone, we cannot determine whether the blue and green rooms are connected. We would need to either:
> > 1. Exhaustively sample trajectories until we find a path (intractable), or
> > 2. Directly query the agent about connectivity (impossible in URLB's framework).
> >
> > URLB provides no mechanism to reward an agent for correctly answering questions whose truth value depends on the entire environment dynamics, not just the features of a single trajectory.
> >
> > **Relationship to WorldTest**
> >
> > WorldTest strictly subsumes URLB. URLB corresponds to the special case where the modification function is the identity:
> >
> > $$
> > \tau_{\mathrm{URLB}}(\mathcal{M}, \xi) = (\mathcal{M}, R_{\mathrm{new}}, H_{\mathrm{new}}),
> > $$
> >
> > That is, $\tau$ leaves the interaction environment unchanged ($\mathcal{M}' = \mathcal{M}$) and introduces a new reward function $R_{\mathrm{new}}$ defined over trajectories in that same environment. In contrast, WorldTest's expanded scope enables evaluation of world model understanding that was impossible in prior frameworks.
> >
> > **How WorldTest Overcomes This Limitation**
> >
> > Instead of evaluating the agent in the same environment with a new reward function, we transform the interaction environment into a derived challenge environment that encodes our query into its structure.
> >
> > The transformation is specified by a modification function $\tau$:
> >
> > $$(\mathcal{M}', R, H) = \tau(\mathcal{M}, \xi),$$
> >
> > where:
> > - $\mathcal{M}$ is the original interaction environment
> > - $\xi$ represents auxiliary randomness for the transformation
> > - $\mathcal{M}'$ is the derived challenge environment (may have completely different dynamics)
> > - $R$ is the reward function for the challenge
> > - $H$ is the episode horizon
> >
> > $\tau$ has access to the full specification of $\mathcal{M}$, including aspects of the dynamics never observed during interaction. This allows $\tau$ to embed environment-level properties, rather than trajectory-level ones, of $\mathcal{M}$ into the structure of $\mathcal{M}'$.
> >
> > **Applying This to Our Example**
> >
> > For the connectivity question, the modification function $\tau$:
> >
> > 1. Inspects $\mathcal{M}$'s complete transition graph $T$,
> > 2. Determines whether the blue and green rooms are connected using graph search,
> > 3. Produces a one-step bandit environment $\mathcal{M}'$ where the agent chooses between two actions: "connected" or "disconnected."
> >
> > The reward function is:
> >
> > $$
> > R(a) =
> > \begin{cases}
> > 1, & \text{if } a \text{ correctly matches the true connectivity in } \mathcal{M}, \\\\
> > 0, & \text{otherwise}.
> > \end{cases}
> > $$
> >
> > The evaluation measures the agent's knowledge of a property of $\mathcal{M}$ that was encoded in its transition dynamics, not the agent's performance on any interaction-phase trajectory in $\mathcal{M}$. The agent must have built an internal world model during interaction that captures this structural property. The challenge $\mathcal{M}'$ tests this knowledge through behavioral performance; no natural-language description or human evaluation required.
> >
> > Broadly, WorldTest permits arbitrary transformations $\tau$ that can:
> > - Modify the state space, action space, and transition dynamics
> > - Create challenge environments structurally unrelated to the interaction environment
> > - Embed latent properties of $\mathcal{M}$ (including counterfactual properties) into $\mathcal{M}'$
> > - Test knowledge that cannot be expressed as a function of any single trajectory

---

> ### Author Response · Authors · 2025-11-27
>
> > 2. Whether AutumnBench is part of the paper's contribution, and discussion on why we need AutumnBench instead of reusing or repurposing BabyAI and ARG-AGI2.
>
> Thank you, we confirm that we claimed AutumnBench as part of our contribution in both Section 1 (Introduction) and Section 6 (Conclusion). Moreover, we agree that providing a more detailed discussion on AutumnBench as a contribution would further strengthen the positioning of our paper. We discuss these points in detail below.
>
> **The contribution of AutumnBench to our framework**
>
> In both of the mentioned sections, we highlighted that AutumnBench serves as the first concrete instantiation of the WorldTest framework. Our goal with AutumnBench is twofold: (1) to show that it is possible to implement WorldTest's more general querying framework as a concrete benchmark, and (2) to show that, as a benchmark, it reveals weaknesses and area for improvements for current state-of-the-art AI agents.
>
> The current set of tasks is not exhaustive, but provides a foundation for systematic world-model evaluation that the community can build upon.
>
> **Using AutumnBench instead of repurposing BabyAI or ARC-AGI2**
>
> We agree with the reviewer that one could construct a WorldTest-style benchmark using existing RL environments such as Minigrid or BabyAI by defining appropriate modification functions $\tau$, or by adding an interaction component to ARC-AGI2. Our choice to base the instantiation of WorldTest in the Autumn DSL was primarily practical. Specifically, to illustrate the utility of WorldTest, we require modelling multiple types of queries in an efficient and extensible manner. In our instantiation, the types of queries are masked frame prediction (MFP), change detection (CD), and planning. Furthermore, as an individual benchmark, we require the deployment of these environments on the web for the experiments with human participants.
>
> AutumnBench is built on the Autumn DSL, which enables concise specification of causal structure and makes it straightforward to implement tasks involving counterfactual reasoning, such as the counterfactual change detection task. The DSL also executes efficiently in the browser, which is crucial for collecting human performance data at scale. In addition, AutumnBench’s abstract 2D grid puzzles are intentionally lightweight, making them well-suited for crowdsourced human evaluation.
>
> In contrast, deploying a WorldTest-style benchmark built on Minigrid for human participants would impose higher technical requirements (e.g., specific operating systems, hardware configurations, installation steps).

---

> ### Author Response · Authors · 2025-11-27
>
> > 3. Moving the categorization of world model evaluation to the related works and providing a clearer discussion connecting and contrasting these related works to WorldTest.
>
> We agree that improving the description and motivation of WorldTest will strengthen the positioning and clarity of the paper. In the revision, we have moved the related work into its own section (Section 2) to more clearly contrast WorldTest against existing approaches. We have also added a motivating example to Section 1 (Introduction) that illustrates how WorldTest enables evaluations impossible under trajectory-based frameworks like URLB that is inline with what we have discussed above.
>
> In summary, the four categories differ in how directly they test knowledge of environment-level properties of a learned world‑model:
>
> 1. **Non‑interactive benchmarks** (e.g., ARC-AGI) target rule learning from static examples but provide no exploration phase. Agents receive curated examples and cannot actively gather information through interaction.
>
> 2. **Representation‑based approaches** probe world‑models via queries but impose restrictive formats: agents must respond in prescribed ways (e.g., text descriptions, symbolic predicates), and evaluation relies on format‑specific proxy metrics.
>
> 3. **Gym-like benchmarks** do not constrain world-model representations but do not directly evaluate world-model learning. Agents simply optimize explicit task objectives, potentially memorizing task-specific policies without building transferable models.
>
> 4. **URLB** (unsupervised RL benchmarks) is closest to our approach but remains restrictive: evaluation objectives must be functions of trajectories in the same environment, preventing measurement of properties that depend on the full dynamics structure rather than individual sampled paths.

---

> ### Author Response · Authors · 2025-11-27
>
> > 4. Clarification on why we consider ARC-AGI non-interactive and comparing AutumnBench to ARC-AGI2 and ARC-AGI3. The commonalities in terms of requirements for internal/hidden rule understanding.
>
> Thank you, we find it interesting to discuss the interactive aspect of ARC-AGI and how it is related to AutumnBench. First, we acknowledge that it is true for both ARC-AGI and AutumnBench that the agent has to understand some internal environment rules to solve the evaluation. Furthermore, there exists ARC-AGI interfaces that allow agents to step-by-step filling in the evaluation grid in an interactive manner.
>
> However, the purpose of interaction is different. In ARC-AGI, the agent receives a fixed set of curated examples and has no opportunity to explore or interact with the environment to learn the rules. This also implies that the agent either learn the rule upfront from the examples, or fails the test.
>
> In WorldTest and consequently AutumnBench, by contrast, the agent must actively explore during the interaction phase to gather the observations to build its own "rule". The data it learns from is determined by its own actions, not by predefined examples.
>
> In summary, the difference is the important emphasis of WorldTest on the interaction during the world model learning and evaluation. This difference is why we characterize ARC-AGI as non-interactive: it tests rule inference from passive data rather than the ability to actively acquire information. Furthermore, even when we can model ARC-AGI’s test phase to include interaction, WorldTest differs in an important way: the test environment in WorldTest is a dynamical environment whose behavior may directly depend on the dynamics of the base environment in the interaction phase.

---

> ### Author Response · Authors · 2025-11-27
>
> > 6. Discussion on building traditional world models from RL literature like Dreamer-v3, Transformer WMs (IRIS), and their performance
>
> We agree that testing with model-based reinforcement learning baselines would provide further insights into what is needed to solve AutumnBench and WorldTest in general. In the revised version, we now discuss how these conventional world-model agents could be evaluated using AutumnBench in **Appendix E.1** and include details of a non‑LLM baseline with simulator access in **Appendix E.2**. To create a more self-contained discussion, we briefly demonstrate these points below.
>
> **Using traditional next-state prediction agents on AutumnBench** A learned world model of the form $\hat{p}(x_{t+1}\mid x_{1:t}, a_{1:t})$ can be evaluated on the three tasks as follows:
> - **Masked Frame Prediction (MFP):** Sample multiple rollouts from the model given the initial state and action sequence, condition on unmasked observations, and select the answer choice whose final frame appears most frequently across samples.
> - **Change Detection (CD):** Flag the first timestep where the observed frame's negative log‑likelihood under the predicted distribution exceeds a threshold (e.g., scaled by the model's entropy).
> - **Planning (PL):** Use model‑based search (e.g., MCTS or MPC) to find action sequences that reach the goal state within the horizon.
>
> Dreamer‑style agents could leverage value‑guided search over learned dynamics; however, all three tasks remain challenging without well‑calibrated predictive likelihoods. While building and tuning a full Dreamer‑style agent is feasible in principle, it requires non‑trivial engineering to align interfaces and objectives of WorldTest.
>
> As an approximation to the upper bound for the performance of traditional world models, we added a non‑LLM baseline in our revision in Appendix E.2: the **Autumn‑simulator agent** with direct access to the Autumn interpreter. This agent uses task-specific solvers:
> - **MFP**: Samples 10,000 trajectories from the ground-truth program and selects the most frequent outcome after filtering for consistency with unmasked frames.
> - **CD**: Compares observed frames to expected next‑frame predictions for 10,000 sample trajectories and flags the first discrepancy.
> - **PL**: Performs breadth‑first search with depth and breadth limits of 10, searched using 10,000 samples.
>
> Despite having the gruond-truth simulator, this baseline still underperforms humans on all three tasks, revealing the following challenges for similar next-frame prediction approaches:
>
> - **MFP**: Exact inference requires reasoning over multiple stochastic trajectories consistent with partial observations. This is a challenging inference problem even with ground-truth dynamics.
> - **CD**: Distinguishing true dynamics changes from stochastic variation requires comparing distributions over multiple trajectories rather than single next‑frame predictions.
> - **PL**: BFS lacks heuristic guidance and expands exponentially under stochasticity and long horizons, often reaching the branching limit (depth/breadth of 10) before finding solutions. Even with perfect dynamics knowledge, the search becomes intractable.
>
> These results suggest that possessing accurate next-frame prediction in isolation is insufficient to reach human-level performance on our tasks. Building a robust Dreamer‑v3 baseline with learned (rather than ground-truth) models remains valuable future work, though we expect similar fundamental challenges to persist.

---

> ### Author Response · Authors · 2025-11-27
>
> > 6. How are the environments changed during the test phase, the three tasks, and whether there are specific rule-based systems that consistently alter the environments
>
> Thank you for the clarifying question. The test phase encodes queries about world understanding by modifying the base POMDP in task-specific ways. The test phase evaluates the agent's world understanding by encoding three queries about the environment as modified POMDPs. We manually and carefully design each query to test particular aspects of the agent's knowledge.
>
> The modifications vary by task type in the following ways:
>
> **Masked Frame Prediction**: The query is transformed into a partially observed trajectory with strategically masked frames. We carefully design both the masking patterns and answer choices to encode the question. The action space is replaced with trajectory navigation (step, rewind), allowing review but preventing new interactions. The observation space includes the masked trajectory plus six answer choices. For stochastic environments, we design masking patterns and choices so exactly one option remains consistent with all possible realizations given the visible frames. This forces agents to reason about what must have occurred rather than conducting new experiments.
>
> **Planning Tasks**: The query is designed as a goal to achieve a target configuration. We carefully design these goals to test whether agents can mentally simulate outcomes and identify paths that avoid dead states (states from which the goal is unreachable). We remove the reset action to prevent trial-and-error exploration, forcing single-attempt planning. The observation space is expanded to include the goal specification (target subgrid configuration and highlight mask). This formulation parallels URLB's downstream-task evaluation with goal-conditioned objectives.
>
> Our reproduction package provides comprehensive examples, including:
> 1. Base Autumn programs (original interaction phase environments)
> 2. Change detection modifications (modified programs with condition statements indicating exact changes)
> 3. Masked frame prediction tasks (complete masked trajectories, six answer choices, correct answer)
> 4. Planning tasks (initial states and goal specifications with highlighted regions)
> 5. The Autumn interpreter for interactive simulation
>
> As a concrete illustration, consider a snake-like environment (XHGKQ):
> - **Original dynamics**: A green snake moves continuously in its current direction (arrow key controlled). When colliding with pink food, the food disappears, it grows, and new food appears randomly. Here, a good query to ask is: ``What happens when the snake "eats" food?''
> - **Change detection**: We modify the collision handler so food disappears without snake growth. The defect time occurs after the first collision.
> - **Masked frame prediction**: We show a trajectory where collision frames are masked and provide six options with different outcomes (growth vs. no growth, food presence vs. absence, snake position variations).
> - **Planning**: Given an initial position, the agent must plan directional moves to position the snake's head at a target location while growing by 3 pixels.

---

> > ### Author Response · Authors · 2025-11-27
> >
> > > 7. While the authors promptly present a few abstract conjectures on why reasoning models failed to match the scores of humans, it would have been great to add a discussion on more concrete steps to improve current reasoning models. How would one go about adding the metacognitive abilities like “better uncertainty quantification” and “flexible belief updating”? I would appreciate a few sentences on this topic during the discussion phase.
> >
> > Thank you, we agree that having more concrete evidence on why the reasoning model fails would provide clearer directions on improving the model.
> > In our revision, we have added a qualitative analysis of the agent's reasoning trace, and we log the full traces for this analysis on Zenodo: https://zenodo.org/records/17728515.
> >
> > Our inspection of these traces reveals that the agent often fails in deriving the correct hypothesis on the world model and often gets stuck on their original belief. To make our discussion self-contained, we quote an excerpt from Claude-4-Sonnet's reasoning trace on a snake-like environment.
> >
> > The agent logs the available actions and forms hypotheses, reasons about the environment, and tests the hypotheses with actions. Yet, the model misattributes the snake's movement: it infers "gravity" rather than recognizing that a prior down action set the latent motion direction of the snake object.
> >
> > ```
> > Key observation: The green object (which appears to be a 2-cell object) has fallen down one row and the bottom part moved right one column. This looks like gravity or falling behavior!Action taken: noop (second time)
> > Result: The green object continued to fall!
> > - Green cells moved from (7,9),(8,9) to (8,9),(9,9) - fell down one more row
> > - Pink cell stayed at (8,10) - still not moving
> > - The green object appears to be a 2-cell vertical object that falls due to gravity
> > - Pink object appears to be stable/not affected by gravity
> > ```
> >
> > In later step, despite correctly mapping the options to the mask, the model answers incorrectly because it learned the wrong world model: it fails to represent that the snake consumes food upon collision, leading to a confident, yet wrong choice.
> >
> > ```
> > REVISED PREDICTION:
> > If mask is rows 7-9, columns 7-10:
> > - Green object last seen at (9,9),(10,9) in Frame 23
> > - Row 9 is at the bottom of the mask
> > - Row 10 is outside the mask
> > - So green object likely moved down but bottom part went outside mask
> >
> > Looking at the options:
> > - Each option is 3 rows × 4 columns
> > - This matches rows 7-9, columns 7-10 of the mask
> > - Need to predict where green and pink objects are within this region
> >
> > POSITION MAPPING:
> > - Pink object at (8,10) maps to row 1, column 3 in the 3x4 options
> > - Green object at (9,9) maps to row 2, column 2 in the 3x4 options
> > - If green moved down, (10,9) would be row 3, column 2 - but that's outside the 3x4 mask
> >
> > FINAL PREDICTION:
> > Based on Frame 23 positions:
> > - Pink at (8,10) → row 1, column 3 in options
> > - Green at (9,9) → row 2, column 2 in options
> > - If green moved down to (10,9), that would be outside the mask
> >
> > Looking at options:
> > Option 2: "black black black pink\nblack black green pink\nblack black black black"
> > - Has pink at row 0, col 3 (matches (8,10))
> > - Has green at row 1, col 2 (close to (9,9))
> > - But has pink in two places...
> >
> > Option 3: "black black green black\nblack black green pink\nblack black black black"
> > - Has green at row 0, col 2 and row 1, col 2 (vertical green object)
> > - Has pink at row 1, col 3 (matches (8,10))
> > - This matches a vertical green object spanning two rows!
> >
> > Option 3 looks most consistent with:
> > - Vertical green object at column 2 (maps to original column 9)
> > - Pink object at row 1, col 3 (maps to original (8,10))
> > ```
> > We believe either a stronger model that would allow the "right prediction" in the first place, or a stronger feedback signal through tool use, and its meta-cognitive abilities, with more flexible belief update mechanisms, would be able to avoid being overly confident in its prediction.

---

> > > ### Author Response · Authors · 2025-11-27
> > >
> > > > 8. Discussion on the cost of evaluating the three models used in the paper
> > >
> > > We report aggregate evaluation costs per model in **Figure 4** and provide details in Appendix D. Across 129 problems, o3 averaged about \\$12 per problem. Average per‑problem costs for Claude‑4‑Sonnet and Gemini‑2.5‑Pro were approximately \\$1.6 and \\$2, respectively. The total cost across all three models on all AutumnBench problems was about \\$2,064. The human study cost approximately \\$6,400 in total.
> > > Exact token counts are not always available for proprietary APIs, so we report dollar costs recorded at run time.

---

> ### Author Response · Authors · 2025-11-27
>
> > 9. Evaluating AutumnBench on RL-based finetuned reasoning models, program synthesis methods, or human cognition-inspired models like HRM.
>
> We believe RL fine‑tuning is promising and have made preliminary attempts. In practice, two challenges limited progress: (i) **data scarcity**: Autumn currently offers 43 environments and 129 tasks, whereas stable RL fine-tuning typically requires large, diverse training curricula with hundreds or thousands of examples; and (ii) **objective design**: WorldTest's interaction phase is intentionally reward‑free, so shaping intrinsic rewards that encourage hypothesis‑driven exploration is non‑trivial, especially under stochastic dynamics where distinguishing informative from random observations is difficult.
>
> We also explored GEPA‑style prompt search [2] as an alternative to RL fine-tuning, but defining an intrinsic‑motivation score that is robust across environments with stochastic transitions proved difficult.
>
> **Program synthesis approaches** are appealing, but we believe they must be coupled with abstract reasoning over the synthesized program to succeed on our tasks. Notably, our Autumn-simulator agent (Appendix E.2), which has direct access to the exact ground-truth program, still underperforms humans, especially on planning. This suggests that possessing the program alone is insufficient. Agents must reason about it effectively to plan under long horizons with stochastic branching.
>
> **Cognition‑inspired models** such as HRM [1] are currently designed for static reasoning tasks. They would need extensions to represent and update time‑varying dynamics, handle stochastic environments, and perform active exploration to be evaluated on AutumnBench.
>
>
> **References**
>
> [1] Wang, G., et.al. (2025). Hierarchical Reasoning Model [Preprint]. arXiv. [https://arxiv.org/abs/2506.21734]
>
> [2] Agrawal et al. (2025). GEPA: Reflective Prompt Evolution Can Outperform Reinforcement Learning. arXiv. [https://arxiv.org/abs/2507.19457]

---

### Author Response · Authors · 2025-11-13
**Thank you for your reviews**

Thank you for your thoughtful reviews and valuable feedback. We appreciate your recognition of the paper's contributions as well as your detailed comments and questions. We are currently addressing all points raised, including proofreading fixes and the suggested supplementary experiments. We will submit our full responses shortly.

---

> ### Author Response · Authors · 2025-11-27
> **Summary of Changes Made in the Revision**
>
> We thank the reviewers for their constructive feedback. We have made the following revisions to address the comments:
>
> - **Improved motivation and structure**: We reorganized the introduction to better motivate the WorldTest framework and moved Related Works into a dedicated section (Section 2) for improved clarity and flow.
> - **Added a new baseline evaluation**: We added an Autumn-simulator agent baseline that has direct access to ground-truth environment dynamics (Appendix E.2.1), establishing an approximation to the upper bound on what perfect world knowledge can achieve.
> - **Traditional world-model evaluation**: We added a detailed methodology in Appendix E.1 for how next-state prediction models (e.g., Dreamer-v3 style approaches) can be evaluated on AutumnBench tasks.
> - **Included an open-source model**: We included results from Qwen3-235b-a22b-thinking-2507, an open-source reasoning model, across all figures and tables to broaden the scope of our evaluation.
> - **Made minor improvement to the manuscript**: We fixed typos, improved plot clarity, and removed repetitions throughout the manuscript.
>
> **Added Additional Analyses:**
> - **Prompt robustness study**: We evaluated three alternative prompt formulations on a subset of problems, showing that while optimal prompts vary by model, performance differences remain modest (Appendix E.3.2, Table E.1).
> - **Perplexity-performance correlation**: We added statistical analysis examining relationships between exploration perplexity metrics and task success, revealing weak but significant correlations at the environment-agent level (Appendix E.4.2, Tables E.4-E.5).

---

### Meta-Review · Area_Chair_T1dV · 2026-01-03

**Summary:**

The reviewers generally find the paper's idea timely and potentially valuable, particularly as an evaluation framework for probing the reasoning limitations of LLMs through interactive environments. However, concerns were raised about the benchmark's novelty and positioning relative to existing world-model evaluations, its limited generalizability beyond 2D grid-world settings, and the sensitivity of the results to prompt design. While the rebuttal addressed many clarification questions and improved the presentation and analysis, key issues remain regarding the scope of the benchmark and the strength of the claims it supports. Taken together, these unresolved concerns lead me to lean slightly toward rejection, despite recognizing the paper's merits and potential.

**Reviewer Concerns:**

### Addressed

**Positioning and novelty relative to existing benchmarks**. The authors clarified how WorldTest/AutumnBench differs from prior benchmarks and provided additional discussion and results to better situate their contribution.

**Presentation quality and clarity of methodology**. The rebuttal and revisions addressed multiple presentation issues and clarified aspects of the evaluation protocol, improving the overall readability and transparency of the paper.

**Human–LLM performance gap**. Additional analyses were provided to examine whether the observed performance gap could be attributed primarily to prompt design, strengthening the empirical basis for the reported findings.

### Outstanding

**Limited generalizability**. While 2D grid worlds enable controlled experimentation and affordable human evaluation, the benchmark remains largely limited to LLMs operating in simple environments. As a result, framing the contribution as a general "world model" benchmark appears somewhat overstated.

**Sensitivity to prompt design**. Although alternative prompts preserve the overall gap between LLM and human performance, the substantial changes in relative rankings among LLMs raise concerns about the benchmark's reliability for measuring improvements or fine-grained model differences.

**Reviewer Scores:**

**Reviewer wsyV (initial score: 6)**. The reviewer raised concerns about the novelty and positioning of WorldTest/AutumnBench relative to existing benchmarks, the lack of comparison with traditional RL world models, and the need for clearer justification of the benchmark's utility and implications for improving reasoning models. In the rebuttal, the authors clarified these points, addressed the reviewer's questions, and included additional results. Therefore, I would expect the reviewer to maintain their positive score.

**Reviewer piDb (initial score: 4)**. The reviewer raised concerns regarding the paper's presentation quality and the clarity of the evaluation methodology for LLMs, in particular questioning whether the reported poor LLM performance could be attributed to prompt design rather than inherent model limitations. In response, the authors revised the manuscript to address the presentation issues and conducted two complementary analyses to assess the impact of prompt formulation. While alternative prompts do not close the performance gap between LLMs and humans, they do affect the relative ranking among LLMs. Given these clarifications and results, I would expect the reviewer to maintain their initial score.

**Reviewer TiEe (initial scores: 2)**. The reviewer questioned the benchmark's practical impact and novelty, in particular how it meaningfully differs from existing world-model evaluations, whether it yields new algorithmic or mechanistic insights, and how broadly it applies. The authors addressed some, but not all, of these concerns. For example, regarding Q4, it remains unclear how the Autumn-Simulator agent described in Section E.2 aligns with the spirit of Dreamer-v3 or how it operates in detail. Given the reviewer's initially low score, I would expect them to maintain their score or possibly increase it slightly, while still retaining an overall negative assessment.

**Reviewer L3rY (initial scores: 6)**. The reviewer expressed concerns about the limited generalizability beyond 2D grid worlds, potential confounds from differing human–AI interfaces, and the strength of the metacognitive claims, while asking questions regarding evaluating RL-fine-tuned models, extending to 3D domains, and analyzing exploration–performance correlations. In the rebuttal, the authors addressed these concerns with detailed clarifications and additional results. Therefore, I would expect the reviewer to maintain their positive rating.

---

### Decision · Program_Chairs · 2026-01-26

Reject